# A theory of cortical map formation in the visual brain

Sohrab Najafian [1,5], Erin Koch[1,2,5], Kai Lun Teh [3,4], Jianzhong Jin[1], Hamed Rahimi-Nasrabadi [1], Qasim Zaidi [1], Jens Kremkow [3,4] & Jose-Manuel Alonso [1✉]

The cerebral cortex receives multiple afferents from the thalamus that segregate by stimulus modality forming cortical maps for each sense. In vision, the primary visual cortex maps the multiple dimensions of the visual stimulus in patterns that vary across species for reasons unknown. Here we introduce a general theory of cortical map formation, which proposes that map diversity emerges from species variations in the thalamic afferent density sampling sensory space. In the theory, increasing afferent sampling density enlarges the cortical domains representing the same visual point, allowing the segregation of afferents and cortical targets by multiple stimulus dimensions. We illustrate the theory with an afferent-density model that accurately replicates the maps of different species through afferent segregation followed by thalamocortical convergence pruned by visual experience. Because thalamo-cortical pathways use similar mechanisms for axon segregation and pruning, the theory may extend to other sensory areas of the mammalian brain.

[1] Department of Biological and Visual Sciences, SUNY College of Optometry, New York, NY 10036, United States. [2] Division of Biology and Biological Engineering, Caltech, Pasadena, CA 91125, United States. [3] Neuroscience Research Center, Charité-Universitätsmedizin Berlin, Charitéplatz 1, 10117 Berlin, Germany. [4] Bernstein Center for Computational Neuroscience Berlin, Philippstraße 13, 10115 Berlin, Germany. [5] These authors contributed equally: Sohrab Najafian, Erin Koch. ✉email: jalonso@sunyopt.edu

Brains segregate neuronal circuits by function. Neurons responding to visual stimuli are located in different sensory areas than those responding to touch or sound and, within each sensory area, neurons responding to different stimulus dimensions are located in different regions forming sensory maps. In mammals, the map of primary visual cortex has a systematic representation of stimulus spatial location, making nearby neurons responsive to nearby stimuli and neurons that are far apart responsive to stimuli that are also far apart. In mammals with high visual acuity and binocular vision, the primary visual cortex maps a more complex combination of stimulus dimensions that include inter-related gradients for spatial position, eye input, light-dark polarity, orientation, and spatial resolution[1–5]. Surprisingly, although visual cortical maps are very diverse across species, the mapping of stimulus orientation can be strikingly similar in different mammalian orders such as primates, carnivores and scandentia[1,3,6]. This puzzling balance between map similarity and diversity has been the topic of intensive research over the past decades and inspired a large number of computational models[7–16]. However, while previous models were very successful at simulating general map patterns for some stimulus dimensions, they were challenged by the use of limited biological constraints and the complexity of simulating inter-related topographies for a large number of stimulus dimensions that include spatial position, eye dominance, light-dark polarity, orientation, spatial resolution, stimulus selectivity, and receptive field structure.

The image pattern of a map for stimulus orientation can be simply simulated by filtering noise. However, simulating an image pattern is very different from simulating a cortical map just as simulating a grid pattern is very different from simulating a map of a city street grid. Accurate geographical maps require reproducing multiple inter-related geographical features to fully describe a spatial location (e.g. the New York Public Library at the west-south corner of 42nd street and 5th avenue in New York City). Similarly, accurate simulations of visual cortical maps require reproducing multiple stimulus dimensions and its relations. A major challenge in visual neuroscience is to understand the functional organization of these multi-dimensional visual cortical maps and their diversity across species. Here we introduce a general theory of cortical map formation, whose main proposal is that cortical map diversity originates from differences in the afferent sampling density of sensory space across species. The theory proposes that, as sampling density increases and more afferents are available to sample each point of visual space, the visual cortex increases the cortical area representing each point and segregates afferents and cortical neurons by other stimulus dimensions that are not just spatial position. We test and support the theory with a computational afferent-density model that accurately replicates a large body of experimental data including new measurements that we obtained to test specific theory predictions.

## Results

Thalamic afferents sample visual space very differently across species. Humans and macaques sample each visual point very densely through a large number of afferents with overlapping receptive fields that need to be accommodated in a large cortical region. By comparison, mice sample visual space more sparsely through a much smaller number of afferents that can be accommodated in a much smaller cortical region. As the number of thalamic afferents increases across species, the visual cortex becomes larger[17,18] allowing humans and macaques to have much larger cortical areas per visual point than mice. The larger number of afferents also allows humans and macaques to have afferents with smaller receptive fields and perceive smaller visual details than mice.

**Theory of cortical map formation.** The close relation between afferent sampling density and cortical size is the main pillar of our theory. The theory proposes that, as afferent sampling density increases, the cortex becomes larger and segregates afferents by other stimulus dimensions that are not just spatial position[17]. This main proposition is best described with a cartoon of brain regions receiving a limited number of thalamic afferents (Fig. 1a, b). In this cartoon, a brain region receiving 32 afferents to sample eight positions of visual space (Fig. 1a) only needs a small area of 32 cortical pixels to accommodate one afferent per cortical pixel (Fig. 1b, left). Because there are four types of afferents (ON and OFF from contralateral and ipsilateral eyes), this small 32-pixel cortex can only use one afferent of each type to sample each position of visual space. Therefore, the 32-pixel cortex can only segregate afferents by spatial position (Fig. 1b, left). By comparison, a brain region receiving 512 afferents needs a larger area of 128 cortical pixels to sample just two spatial positions (Fig. 1b, right). Because this larger brain region samples each spatial position with 64 afferents, it can segregate the afferents by position, eye input (contralateral or ipsilateral) and ON-OFF polarity (light or dark). This segregation replicates the pronounced tendency of brains to sort neurons with different response properties in different areas, layers or clusters. It also maximizes the response synchrony of neighboring neurons to stimuli, helping them drive their common cortical targets more effectively[19] while minimizing axon wiring[20,21] (Fig. 1c). The primary visual cortex receives input from many different types of afferents. However, for simplicity, the theory considers only the four afferent types that are best preserved across species, which are those that signal the onset of light (ON) or dark (OFF) stimuli projected on the contralateral and/or ipsilateral retinas.

The theory assumes that afferents are sorted following rules of like-to-like connectivity that have been demonstrated in the thalamocortical pathway of different species and sensory systems[22–27]. The sorting makes the four types of afferents to become organized in an eye-polarity grid[5,28] (Fig. 1d) of OFF and ON afferents from contralateral and ipsilateral eyes (OFF-contra, OFF-ipsi, ON-contra, ON-ipsi), and makes cortical receptive fields dominated by ON or OFF visual responses driven by either eye (Fig. 1e). The theory puts forward several predictions, some of which already have experimental support. For example, it predicts that the size of the cortex should increase in proportion to the number of afferents that it receives, which has been demonstrated in different species[17,18]. It also predicts that retinotopy should change slower across the contra-ipsi border than ON-OFF border of the eye-polarity grid (Fig. 1e), a prediction that is consistent with experimental measurements[28–30]. The theory also puts forward predictions that have not been tested in experiments. For example, it predicts that gradients for ON-OFF response balance, orientation selectivity, spatial frequency selectivity, and spatial resolution should be all correlated because changes in ON-OFF response balance affect cortical stimulus selectivity. That is, as the ON-OFF response balance increases, the dominant subregion of the cortical receptive field should become more suppressed by the stronger flank (Fig. 1f, top), increasing orientation selectivity (Fig. 1f, bottom left), spatial frequency bandpass (Fig. 1f, bottom right, see changes in low spatial frequency cutoff), and spatial resolution (Fig. 1f, bottom right, see changes in high spatial frequency cutoff). In the following sections of the paper, we describe a computational model that simulates the cortical maps predicted by the theory, compare model simulations with experimental measurements, and summarize the main theory propositions at the end.

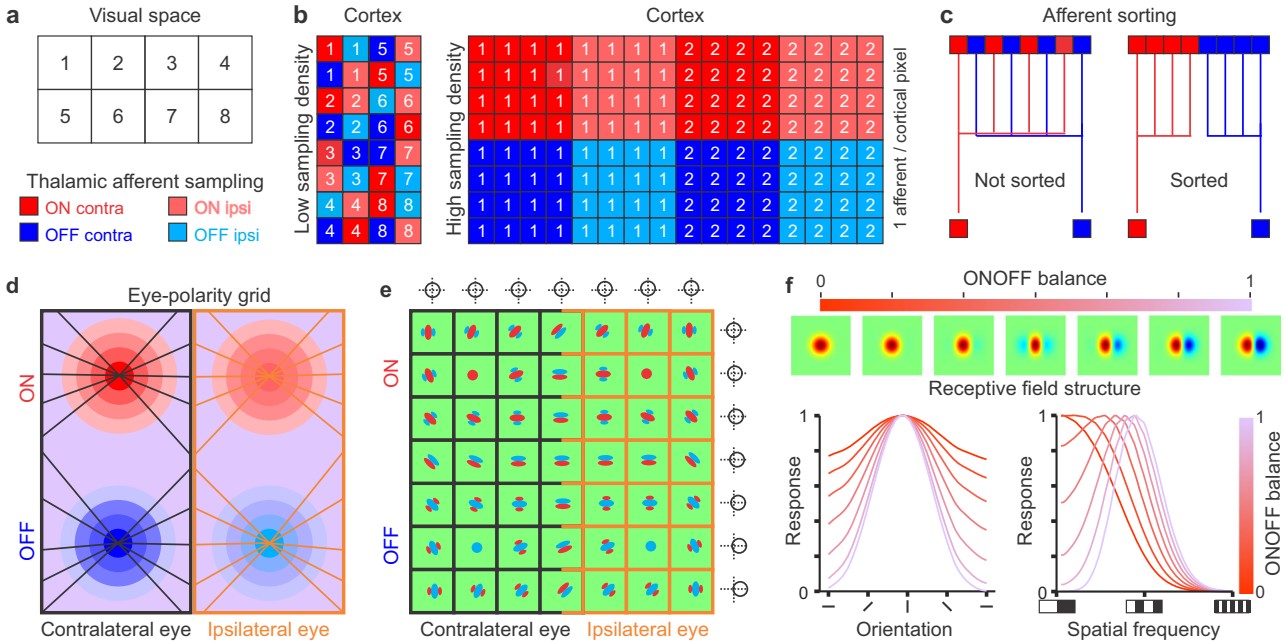

**Fig. 1 Theory of cortical map formation. a** Cartoon illustrating 8 adjacent positions in visual space that can be discriminated as separate by visual cortex (e.g. two stimuli are perceived as separate only if they fall in different squares). The theory assumes that each square position is sampled by four different types of thalamic afferents that signal the onset of light (ON: red) or dark stimuli (OFF: blue) projected on the contralateral (contra: high contrast) or ipsilateral retina (ipsi: low contrast). The size of the squares represents the visual resolution (receptive field size) of the thalamic afferents. **b** Cartoon illustrating a small cortex sampling each spatial position with four afferents (left) and a larger cortex sampling each spatial position with 64 afferents (right). **c** Cartoon illustrating the afferent sorting. **d** The theory assumes that the visual cortex sorts afferents with overlapping receptive fields by eye input and contrast polarity, forming an eye-polarity grid with separate regions for ON-contra, OFF-contra, ON-ipsi, and OFF-ipsi afferents that are linked by iso-orientation lines. **e** This afferent sorting makes cortical receptive fields become dominated by eye input and ONOFF polarity. Because afferents with different eye input are better matched in retinotopy than afferents with different polarity, retinotopy changes slower with cortical distance across the eye than ONOFF polarity border (circles and crosslines at panel outer borders illustrate changes in receptive field position). **f** The theory predicts that increases in ONOFF response balance are associated with increases in stimulus selectivity. The top panel shows an increase in ONOFF response balance from ON dominated to equal ON and OFF subregion strength. The bottom panel illustrates how the increase in ONOFF response balance increases orientation selectivity (left), bandpass tuning and spatial resolution (right).

**Main stages of the computational model**. The computational model has three main stages: retinal development, development of the cortical subplate, and development of the visual cortex. In the first stage, the model simulates mosaics of ON and OFF retinal ganglion cells from two eyes projecting, through the thalamus, to one brain hemisphere, ipsilateral to one eye and contralateral to the other (Fig. 2a). The model uses these mosaics to simulate the afferent sampling density, defined as the number of thalamic afferents with overlapping receptive fields (Fig. 2b). Future versions of the model could increase the afferent sampling density by simulating retino-geniculo-cortical divergence/convergence[31,32] and/or including additional mosaics of retinal ganglion cells[33]. However, for simplicity, all simulations reported in the paper use only one set of ON-OFF mosaics for each eye.

The second stage of the model simulates the sorting of thalamic afferents in the cortical subplate, an embryonic brain structure that receives the afferents days or weeks before the cerebral cortex is formed[34,35] and that plays an important role in the development of cortical maps[36–38]. The model sorts the thalamic afferents first by retinotopy, then by eye input, and then by ON-OFF polarity (Fig. 2c), replicating a developmental sequence experimentally demonstrated in the ferret lateral geniculate nucleus[5,39], which is likely to be preserved across species (see methods for details).

At the third and last developmental stage, the afferents grow into an immature visual cortex and start spreading their axonal arbors. The axon arbor spread provides each cortical region with convergent input from multiple thalamic afferents while preserving the dominance by eye input and ON-OFF polarity from the afferent sorting (Fig. 2d, e). The thalamocortical convergence of ON and OFF afferents makes each cortical region orientation selective, generating a primordial orientation map (Fig. 2f). At the end of development, visual experience optimizes the primordial map by maximizing the coverage and binocular match of stimulus orientation (Fig. 2g). In the next section of the paper, we describe each stage of our afferent-density model and the associated theory predictions in greater detail (see methods for a mathematical description of the model and use our graphical user interface[40] to perform customized stimulations).

**Model stage 1. Retinal development**. The theory predicts that the organization of a visual cortical map should be determined by the retinal position of ON and OFF ganglion cells sampling visual space with the two eyes, as proposed by previous retinal models[8,12,15,41,42]. However, unlike previous retinal models, the theory does not require (and does not rely on) a Moire interference of ON and OFF retinal arrays to generate orientation maps. The theory predicts that nearly all ON-OFF mosaic geometries should be able to generate an orientation map as long as the afferent sampling density is high. Conversely, it predicts that orientation maps will not form if the afferent sampling density is low, regardless of the ON-OFF mosaic geometry. Several factors can make the afferent sampling density low, including a small eye, a small number of retinal ganglion cells, a small visual thalamus, and/or a small number of retinal ganglion cells projecting to the visual thalamus.

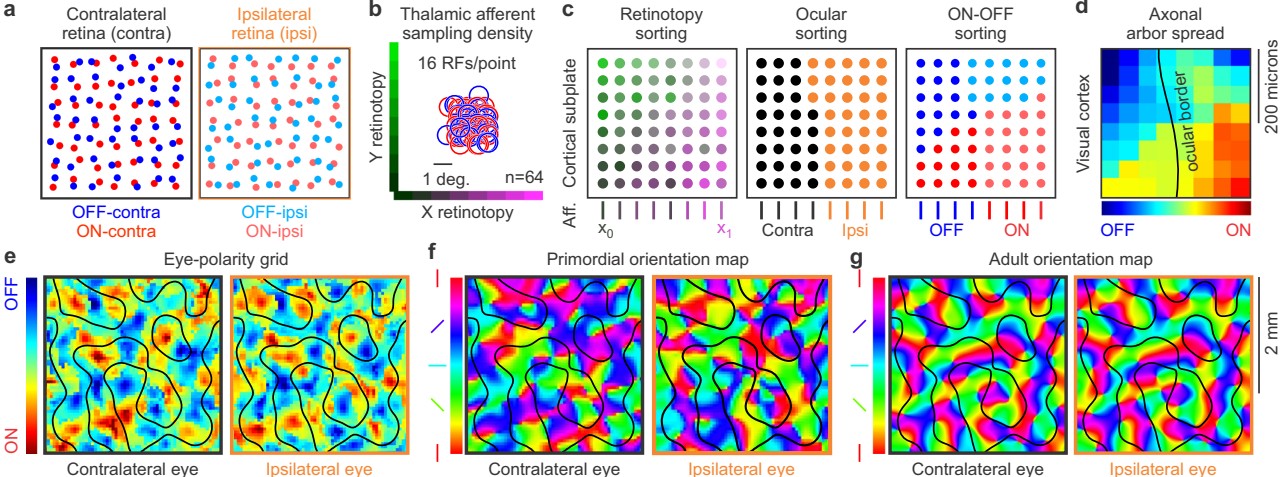

**Fig. 2 Main stages of the computational model. a** Retinal development. Model simulations of ON and OFF retinal ganglion cells (red and blue circles) from the contralateral and ipsilateral eyes (high and low contrast). **b** Receptive fields (RF) of 64 thalamic afferents receiving input from the retinas illustrated in a and thalamic afferent sampling density (16 receptive fields sampling the same visual point). **c** Model simulations of the afferent sorting in the cortical subplate. The 64 afferents (Aff.) illustrated in b are sorted first by retinotopy (left), then by eye input (middle) and then by ONOFF polarity (right). **d** Model simulations of the eye-polarity grid after the afferent axon arbors spread and combine in each cortical pixel (red: ON dominated, blue: OFF dominated, black line: border between regions dominated by contralateral and ipsilateral eyes). **e** Model simulations of the eye-polarity grid in a larger cortical patch. **f** Primordial orientation maps for the contralateral (left) and ipsilateral eyes (right) resulting from thalamocortical convergence. **g** Adult orientation map after the primordial map is optimized by visual experience.

To simulate these theory predictions, the afferent-density model can generate a large variety of retinas with different sizes, sampling densities, retinotopic distances between ON and OFF afferents, and asymmetries in retinal ganglion cell density across retinotopic axes and/or cell types (Fig. 3a). It can also simulate ON-OFF retinal mosaics of different species by finely adjusting the retinal-distance distributions of each pair combination of ON-OFF cell types (Supplementary Fig 1, see methods for details). To simulate a retina, the model makes grids of retinal ganglion cells for each type with a fixed cell density (Fig. 3a, left). Then, it randomizes the location of each cell with a jitter value multiplied by the cell separation (Fig. 3a, middle). Then, it applies an ON-OFF interaction factor that prevents two cells from occupying the same retinal position (Fig. 3a, right). All subsequent stages of the model, from the afferent sorting to the synaptic competition are fully determined by the ON and OFF retinotopy from the two eyes. Therefore, two model simulations that use exactly the same retinas (and thalamic afferents) will perform exactly the same afferent sorting (by retinotopy, eye input and ONOFF polarity), and generate exactly the same visual cortical maps (Fig. 3b).

Our afferent-density model can generate cortical orientation maps with any retina that, through visual thalamus, provides the cortex with a large number of afferents with overlapping receptive fields. For example, the theory predicts that an animal with an ON-OFF retinal mosaic of a mouse diverging extensively in a large visual thalamus of a cat will have a cortical orientation map because the large thalamus provides the cortex with high afferent sampling density. Conversely, an animal with an ON-OFF retinal mosaic of a cat but a small visual thalamus of a mouse will not have a cortical orientation map because the small thalamus provides low afferent sampling density (Fig. 3c). Although these chimera animals may not exist in nature, the retinal/thalamic cell-density ratios vary greatly across species. For example, rabbits have a larger number of retinal ganglion cells than cats[17] but a much smaller visual thalamus and, as the theory predicts, rabbits do not have cortical orientation maps. The theory also predicts that animals sampling the binocular field more densely with one

eye than the other will only form cortical orientation maps with the eye that has sufficiently large afferent sampling density (the dominant eye). Only at later stages of development, when the maps are binocularly matched by visual experience (see below), cortical neurons driven by the non-dominant eye will form weak orientation clusters to match the orientation preferences of the dominant eye (Fig. 3d). This prediction is consistent with experimental measures in tree shrews, an animal that samples the binocular field more densely with the contralateral than ipsilateral eye[43]. All published orientation maps in tree shrews were obtained through stimulation of the contralateral eye; the maps cannot be measured through stimulation of the ipsilateral eye[3,44]. Notice that the afferent sampling density of visual space from our theory is very different from the cortical sampling of ON-OFF retinal geometry previously proposed[45] and leads to opposite predictions (i.e. the sampling of ON-OFF retinal geometry predicts that orientation maps in tree shrews should be weaker for the dominant than non-dominant eyes because the cortical/retinal ratio is lower for the dominant eye).

**Model stage 2. Development of the cortical subplate**. The theory predicts that species sampling visual space with a large number of thalamic afferents will have large cortical regions representing each visual point and many afferents with overlapping receptive fields. In turn, the large number of afferents with overlapping receptive fields should make the afferents segregate in cortex by other stimulus dimensions besides spatial position. For simplicity, the model simulates the sorting of afferents in the cortical subplate (i.e. before axon arbor spread); however, the sorting could also occur in visual cortex through molecular gradients and/or synaptic competition. The most important requirement of both theory and model is that afferents are sorted by spatial position (retinotopy), eye input and ON-OFF polarity, regardless of the sorting mechanism. The model performs the retinotopic sorting by assigning nearby positions of the cortical subplate to afferents with nearby retinotopy. It performs the ocular and ON-OFF sorting by convolving a difference-of-

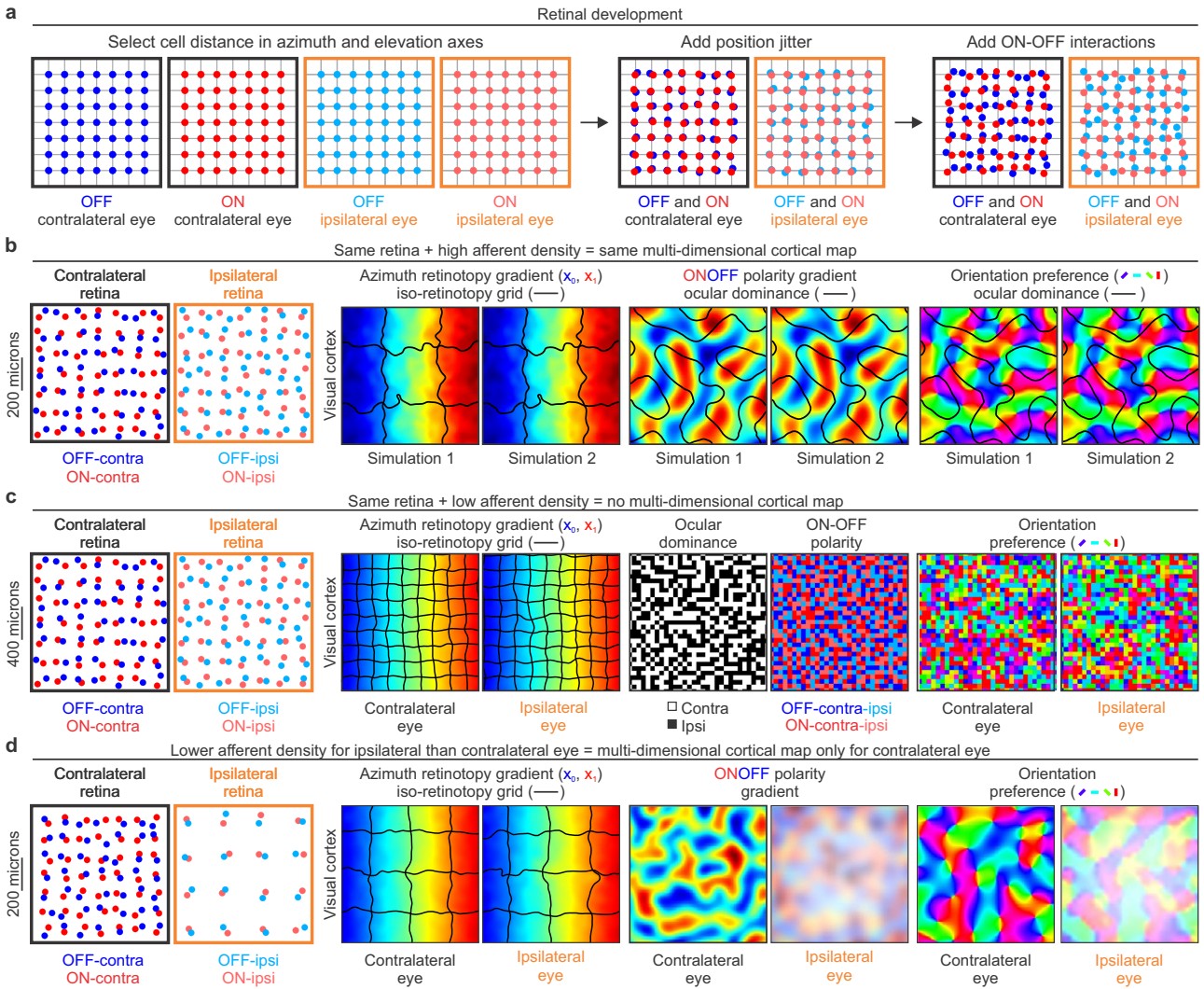

**Fig. 3 First model stage: retinal development. a** Mosaics of ON and OFF retinal ganglion cells from two eyes generated by the model. The model places cells of each type in a grid (left), then randomly jitters their retinal positions (middle), and applies cell interactions that prevent ON and OFF cells from occupying the same position in each eye. **b** If afferent sampling density is high, identical ONOFF retinal mosaics generate identical visual cortical maps as shown by two simulations (1 and 2) illustrated in each pair of panels. From left to right, simulated retinas for the two eyes and repeated simulated maps for retinotopy, ONOFF, and orientation preference of stimuli presented in the contralateral eye. The colors illustrate cortical gradients of azimuth retinotopy, ONOFF dominance and orientation preference. The black lines illustrate borders of iso-retinotopic cortical patches in the retinotopic map (regions containing cortical cells with overlapping receptive fields), and the ocular dominance borders in the ONOFF and orientation maps. **c** The same ONOFF retinal mosaics used in the simulations above (**b**) do not generate orientation maps if the spacing of retinal cells is increased to make the afferent sampling density lower (notice different retinal scale in **b** and **c**, 200 versus 400 microns). From left to right, each pair of panels show the two retinas, retinotopic maps for contralateral and ipsilateral eyes, afferents in the cortical subplate, and orientation maps. Notice that the size of the iso-retinotopic patch is smaller in (**c**) than (**b**) because the afferent sampling density is also lower. **d** If the afferent sampling density is lower for the ipsilateral than the contralateral eye, orientation maps emerge from afferent sorting only for the contralateral eye. From left to right, each pair of panels show the two retinas, retinotopic maps, ONOFF maps and orientation maps for the two eyes. Notice that the size of the iso-retinotopic patch is large for both eyes because there is a large number of afferents from the contralateral eye. The afferents from the ipsilateral eye are fewer in number and should have larger axon arbors because they have less synaptic competition.

Gaussians function (afferent sorting filter) with a cortical subplate made of random binary values. The afferent sorting filter provides a simple approach to adjust the shape and size of the afferent clusters and iso-retinotopic domains based on visual sampling[29] (see Supplementary Fig 2). However, the model can use any other sorting filter that segregates afferents by retinotopy, eye input and ON-OFF polarity as long as the sorting maximizes the coverage of all stimulus dimensions per visual point.

The only role of the afferent sorting filter is to find the best local movement of an afferent to join afferents of the same type (Fig. 4a). During the ocular sorting, afferents move to regions

with the same eye input and retinotopy (Fig. 4b) and, during the ONOFF sorting, they move to regions with the same retinotopy, eye input and ON-OFF polarity (Fig. 4c, left). This simple algorithm is very effective at sorting afferents by type (Fig. 4c, left). Moreover, because the sorted afferents have similar retinotopy, the retinotopic map is preserved (Fig. 4c, right).

The theory predicts that visual retinotopic maps will not form if thalamic afferents are not sorted by retinotopy (Fig. 4d), ocular dominance maps will not form if afferents are not sorted by eye input (Fig. 4e), and ON-OFF maps will not form if afferents are not sorted by ON-OFF polarity (Fig. 4f). In addition, the theory

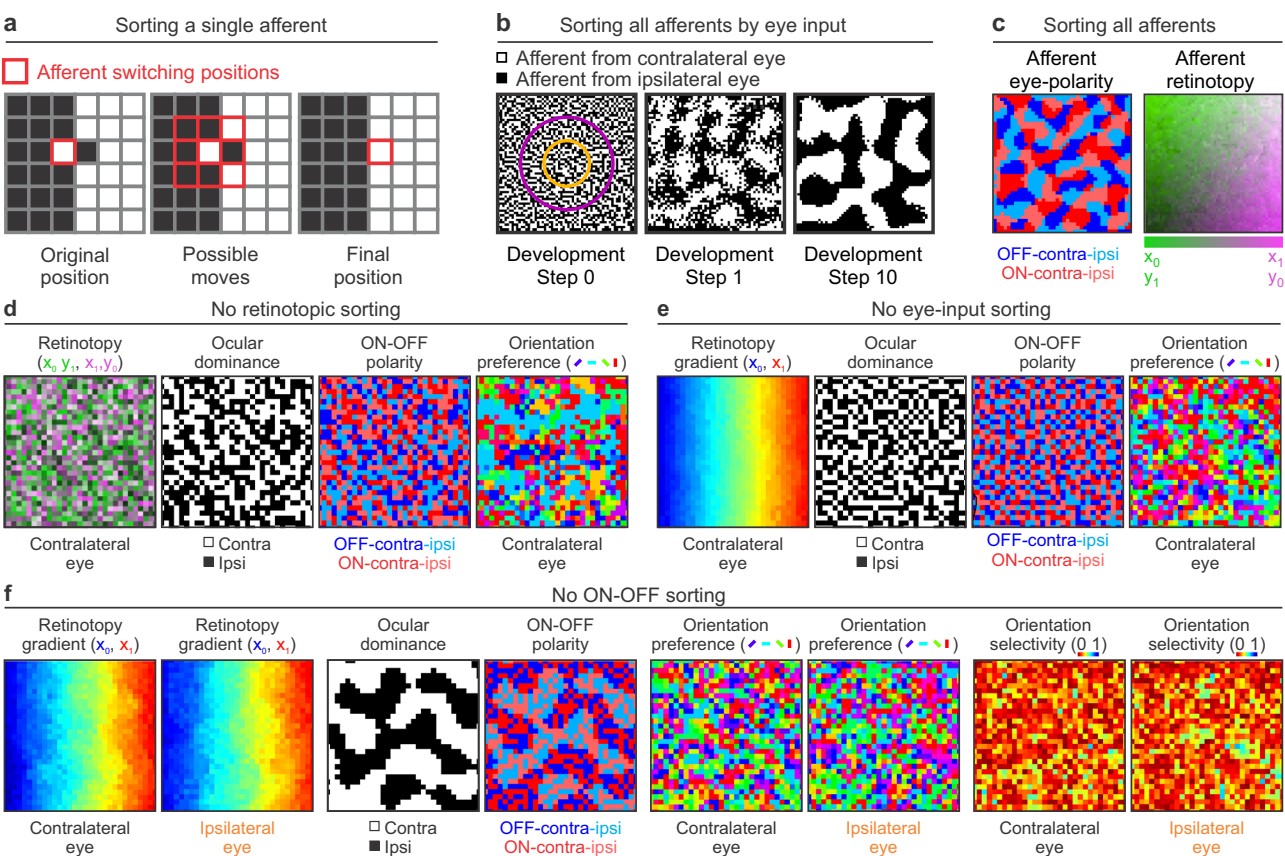

**Fig. 4 Second model stage. Afferent sorting in the cortical subplate. a** The model sorts afferents by retinotopy, eye input and ONOFF polarity. The sorting by eye input is done by performing a convolution between an afferent sorting filter and a binary randomized cortical subplate that assigns values of 1 to afferents from the contralateral eye (black pixels) and -1 to afferents from the ipsilateral eye (white pixels). The model selects an afferent (left, red outline) and computes the convolution at 9 neighboring positions (middle). Then, it selects the position that gives the highest convolution value (right), which is the position where the afferent is surrounded by the largest number of afferents of the same type. **b** Sorting of afferents by eye input in a larger cortical plate. The afferent sorting is performed in 10 developmental steps based on the convolutions with the afferent sorting filter (concentric purple and orange circles on the left). The afferents are randomly organized at step 0 (left), start segregating at step 1 (middle) and the segregation is complete at step 10 (right). **c** The same sorting method is used for ONOFF polarity. At the end of the sorting process, afferents are segregated by eye input and ONOFF polarity (left) in addition to retinotopy (right). **d** Random afferent sorting by retinotopy causes random mapping of all stimulus dimensions in simulated visual cortex. From left to right, mapping of retinotopy, ocular dominance, ONOFF polarity and orientation preference for the contralateral eye. **e** Random afferent sorting by eye input causes random mapping of all stimulus dimensions except retinotopy in simulated cortex. From left to right, normal cortical retinotopy and random mapping for ocular dominance, ONOFF polarity and orientation preference. **f** Random afferent sorting for ONOFF polarity causes random cortical mapping of all stimulus dimensions except retinotopy and eye input. From left to right, pairs of panels showing cortical retinotopy for both eyes, afferent segregation by eye input but not ONOFF polarity, and random mapping of orientation preference and selectivity for both eyes.

predicts that any disruption of afferent sorting will affect the maps for stimulus dimensions originating from thalamocortical convergence such as orientation and spatial frequency (Fig. 4d–f). In the model, random mapping of retinotopy, eye input and/or ON-OFF polarity leads to random mapping of stimulus orientation, even if the afferent sampling density is high. Without afferent sorting, cortical neurons with different orientation preference (e.g. vertical or horizontal) and orientation selectivity (e.g. narrow or broad bandwidth) become neighbors (Fig. 4f).

**Model stage 3. Development of visual cortex.** After the afferent sorting is complete, the model simulates the spread of the afferent axonal arbors in visual cortex. The axonal arbor is modeled as a Gaussian function centered at the main axon trunk of each afferent. It has a maximum synaptic weight of one at its center and a minimum of zero at its edges (Fig. 5a, radius: 10 cortical pixels). The spread of the axonal arbors provides each cortical pixel of 50 × 50 microns with convergent input from multiple afferents. Therefore, simulations within each cortical pixel can be

interpreted as averages of multiple neighboring neurons contained within a 50 × 50 microns of cortex that can be measured with multiunit activity[28,46]. The convergent thalamocortical input includes the main thalamic afferent that reached the cortical pixel first (during the development of the cortical subplate) and other afferents with diverse properties that reached the cortical pixel later through axon arbor spread.

The spread of the axonal arbors makes multiple thalamic afferents to converge at the same cortical pixel and compete for cortical space. To simulate the synaptic competition, the model calculates the dominant receptive field of each cortical pixel (Fig. 5b, left) as the weighted receptive field average of all thalamic afferents with the ON-OFF polarity of the afferent that reached the cortical pixel first, before axon arbor spread. Then, the model changes the synaptic weights of the afferents with opposite polarity that have receptive fields overlapping the dominant receptive field. The synaptic weights of these non-dominant afferents become zero if their receptive fields fully overlap the dominant receptive field (Fig. 5b middle and right, central red dots) but can approach the maximum value of one if

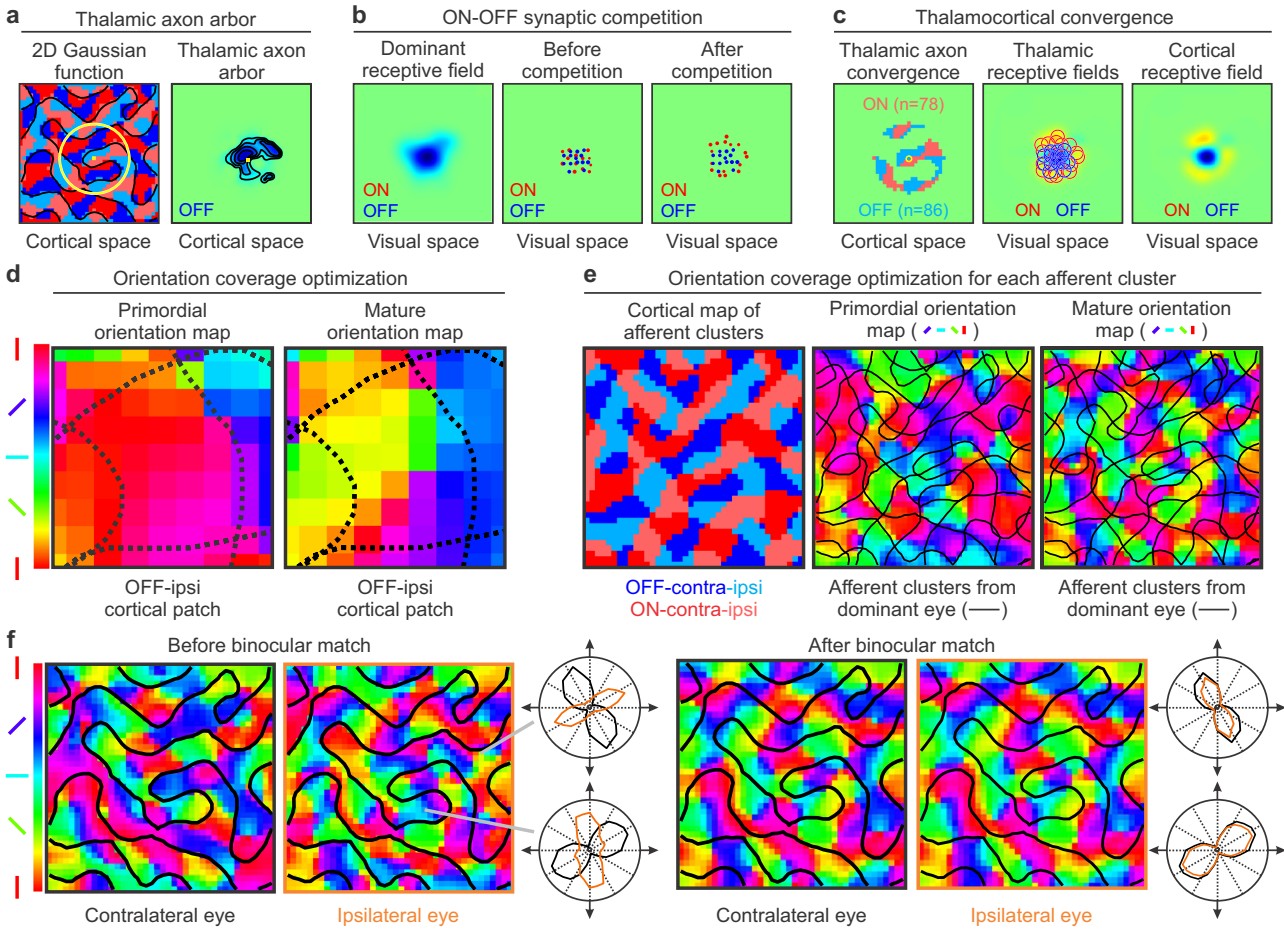

**Fig. 5 Third model stage. Development of visual cortex. a** After the afferent sorting is complete, the model simulates the spread of each afferent axon arbor with a Gaussian function (left, yellow circle, standard deviation ~3 cortical pixels, radius: 10 cortical pixels). The synaptic weight of the axon arbor is maximum at the center of the Gaussian (small yellow pixel) and zero at the Gaussian borders. The final size of the axon arbor (right) is also shaped by synaptic competition. **b** The synaptic competition changes the synaptic weights of the afferents with non-dominant polarity converging at each cortical pixel. The synaptic change is based on the overlap between the afferent receptive field and the dominant receptive field of the cortical pixel (left). The synaptic weight decreases when the overlap is large and increases when it is small, as illustrated in the figure for afferents with synaptic strength >50% of maximum (middle and right). The dot location illustrates the receptive field position in visual space and the dot size illustrates the synaptic strength. **c** Afferents converging at the same cortical pixel segregate in cortical space (left, cortical position of main axon trunks), and visual space (afferent receptive fields). The cortical receptive field is calculated as the weighted receptive field average of the afferents (right). **d** The model optimizes the orientation coverage at each afferent cluster, as illustrated for an example OFF-ipsi afferent cluster (left, demarcated by dotted lines). This afferent cluster is strongly biased towards horizontal orientations (red pixels) in the primordial map but its orientation coverage becomes more uniform in the mature map (right, notice the greater diversity in pixel colors). **e** The model maximizes the orientation coverage for each afferent cluster (left) to transform the primordial orientation map (middle) into a mature map (right, afferent clusters demarcated by black lines). **f** The model also maximizes the binocular match of the orientation map by adjusting the synaptic weights of the afferents from the non-dominant eye (left) to match those of the dominant eye in the mature orientation map (right).

the receptive field overlap is minimal (Fig. 5b right, red dots at the edges, notice that only synaptic weights >50% of maximum are shown in middle and right panels). At the end of the synaptic competition, the multiple afferents converging at the same cortical pixel are segregated by ON-OFF polarity in both cortical space (Fig. 5c, left) and visual space (Fig. 5c, middle).

The model calculates the cortical receptive field of each cortical pixel as the weighted receptive-field average of all the converging afferents from each eye (Fig. 5c, right). Future versions of the model could also simulate receptive fields of single cortical neurons by taking subsets of thalamic afferents but, for simplicity, all reported simulations in this paper are average receptive fields of neuronal populations. The model extracts the orientation preference from the cortical receptive field of each cortical pixel to generate a primordial orientation map. This primordial map

simulates the map of naïve brains not exposed to visual experience[47] and does not yet have a complete representation of stimulus orientation for each afferent cluster. For example, an OFF-ipsi afferent cluster in the primordial map may be strongly biased towards vertical orientations and fail to sample horizontal orientations (Fig. 5d, left, area within dotted lines), causing a visual scotoma for horizontal dark stimuli from the ipsilateral eye.

The theory assumes that visual experience prevents the emergence of these scotomas by optimizing the orientation coverage of each afferent cluster. The model performs this optimization by convolving the cortical patch from each afferent cluster with twelve orientation-coverage filters to generate twelve possible orientation-map patches and assigns the orientation patch with best orientation coverage to the mature orientation map (Fig. 5d, right, area within dotted lines). This optimization is

performed for all afferent clusters (Fig. 5e, left) to transform the primordial map (Fig. 5e, middle) into a mature orientation map that becomes periodic (Fig. 5e, right). The optimization of orientation coverage can be interpreted as the result of afferent co-activation by stimuli with multiple orientations during visual experience (Supplementary Fig 3). At the end of cortical development, the model optimizes the binocular orientation match, a process that is known to require visual experience in both rodents[48,49] and carnivores[50]. The binocular match is simulated by adjusting the synaptic weights of afferents from the non-dominant eye to match the orientation preference of the dominant eye (Fig. 5f, see methods for more detail). Notice that this optimization is modulated by visual experience but is fully determined by the shape and size of the afferent clusters emerging from afferent sorting. That is, if visual experience is normal (i.e. it provides a homogeneous distribution of orientations from both eyes), two animals with identical retinas will generate an identical sorting of afferents for retinotopy, eye input, and ON-OFF polarity, identical ON and OFF afferent clusters, and identical cortical maps.

**Mature visual cortical maps.** At the end of cortical development, the model generates mature cortical maps for multiple stimulus dimensions for the contralateral eye, ipsilateral eye or both eyes (Fig. 6a). It also generates anatomical maps of afferents converging at the same cortical pixel in cortical space (Fig. 6b, c, first panel from the left) and visual space (Fig. 6b, c, second panel from the left) together with the binocular cortical receptive fields and orientation tuning emerging from afferent convergence (Fig. 6b, c, third and fourth panels starting from the left, see also Supplementary Fig 4a, b). The model also generates receptive fields and response properties measured across linear tracks of visual cortex (Fig. 6d), closely replicating experimental measurements[28]. As in the experimental measures, the simulations generate diverse cortical receptive field structures (e.g. ON dominated, OFF dominated, ON-OFF balanced), gradients of orientation preference (e.g. fast fractures or slow linear zones), gradients of orientation tuning (e.g. broad or narrow, see also Supplementary Fig 4c), and orientation pinwheels that are OFF dominated, ON dominated or ON-OFF balanced[28]. This diversity emerges in the model from variations in ON-OFF response balance across the cortex. Regions strongly dominated by ON or OFF afferents generate broad orientation tuning and rapid changes in orientation preference (e.g. section from 0 to 0.1 mm in Fig. 6d) whereas regions with greater ON-OFF response balance generate sharper orientation tuning and slower changes in orientation preference (e.g. section from 0.5 to 0.6 mm in Fig. 6d).

The orientation maps generated by the model have similar power spectrums and hexagonal periodic geometry[8] to orientation maps measured in different species (Supplementary Fig 4d). The model also simulates changes in cortical map architecture caused by monocular deprivation or orientation biases in visual experience (Supplementary Fig 4e). It also replicates experimental measurements of multielectrode recordings crossing ON and OFF domains along linear tracks that are parallel or orthogonal to ocular dominance bands[28] (Supplementary Fig 5). The model also replicates the smooth changes in retinotopy across cortical distance[4,51] and the retinotopic distortions at neighboring cortical regions[28,46,52] (Fig. 6e, f, movies Fig 6e, f). The smooth retinotopy changes result from the afferent sorting by retinotopy and the local distortions from the sorting by other stimulus dimensions (e.g. afferents with the same retinotopy are separated in cortex if they have different eye input and/or ON-OFF polarity).

**Model-data comparison of species variations.** An important prediction of the theory is that cortical orientation maps from individual animals should be accurately reconstructed by using as input the retinotopy of ON and OFF ganglion cells from the two eyes. Unfortunately, there are no current measurements of cortical orientation maps and retinal ganglion cells obtained in the same animal, and all retinal measures are from one eye only. In absence of these data, the theory also predicts that differences in orientation maps across species should be also replicated using ocular dominance or retinotopy maps as input[29] (although individual-animal maps cannot be replicated without ONOFF retinotopy). To test this prediction, we simulated the orientation maps of three different species (macaque, cat and tree shrew)[1,3,6] using as inputs published ocular-dominance maps from the same individual animal (Fig. 7a, left, middle), or retinotopic maps from the same species (Fig. 7a, right, tree shrews do not have ocular dominance maps).

When using individual ocular dominance maps as inputs, the model made a binary black-and-white version of the ocular dominance map to simulate the cortical subplate. Then, it assigned a random binary value of ON-OFF polarity to each pixel of the cortical subplate while preserving the segregation by eye input. The randomized ON-OFF values were then sorted with the afferent sorting filter estimated from the ocular dominance segregation (see[29] for details) to generate ON and OFF afferent clusters for the contralateral and ipsilateral eyes. At the last simulation stage, the model optimized the orientation coverage of each afferent cluster to generate the orientation maps (Fig. 7c, left and middle).

When using the tree-shrew retinotopic map as input, the simulations were more challenging because the available retinotopic maps are coarse and from different individual animals than the published orientation maps[3]. To overcome these limitations, the model generated a cortical subplate with the same shape as the published orientation map. Then, it divided the cortical subplate into 12 large retinotopic sectors separated by three iso-retinotopic azimuth lines and three iso-retinotopic elevation lines. The position of the iso-retinotopic lines was based on the species retinotopic map and the global pattern of the orientation map from the individual animal. The model assumes that, to maximize orientation coverage, the iso-orientation domains need to fit within the iso-retinotopic domains. Therefore, the orientation domains become elongated in elongated iso-retinotopic domains and round in round iso-retinotopic domains. Because the cortical map of tree shrew is strongly OFF-centric and dominated by afferents from the contralateral eye[53], these simulations can be interpreted as a maximization of orientation coverage in cortical clusters dominated by OFF afferents from the contralateral eye.

The model was able to closely reproduce the species differences in orientation map geometry (compare Fig. 7a, c). It also reproduced the tendency for iso-orientation lines to run orthogonal to ocular dominance borders in macaques and cats and the tendency for same-sign pinwheels (e.g. both rotating clockwise) to be farther apart than different-sign pinwheels (e.g. one rotating clockwise and the other counterclockwise). It also reproduced the differences in average pinwheel distance and orientation periodicity across species (compare Fig. 7b and d). We quantified the model accuracy at reproducing map differences between species by measuring the model error for different map properties (Supplementary Table 1). The model error was calculated as the difference between each experimentally-measured and model-simulated map for each species and map property. For comparison, a control error was calculated as the difference between experimentally-measured maps from two different species. The model error was significantly lower than the control error for all map properties. Moreover, the average model error across properties and species was four times lower than the

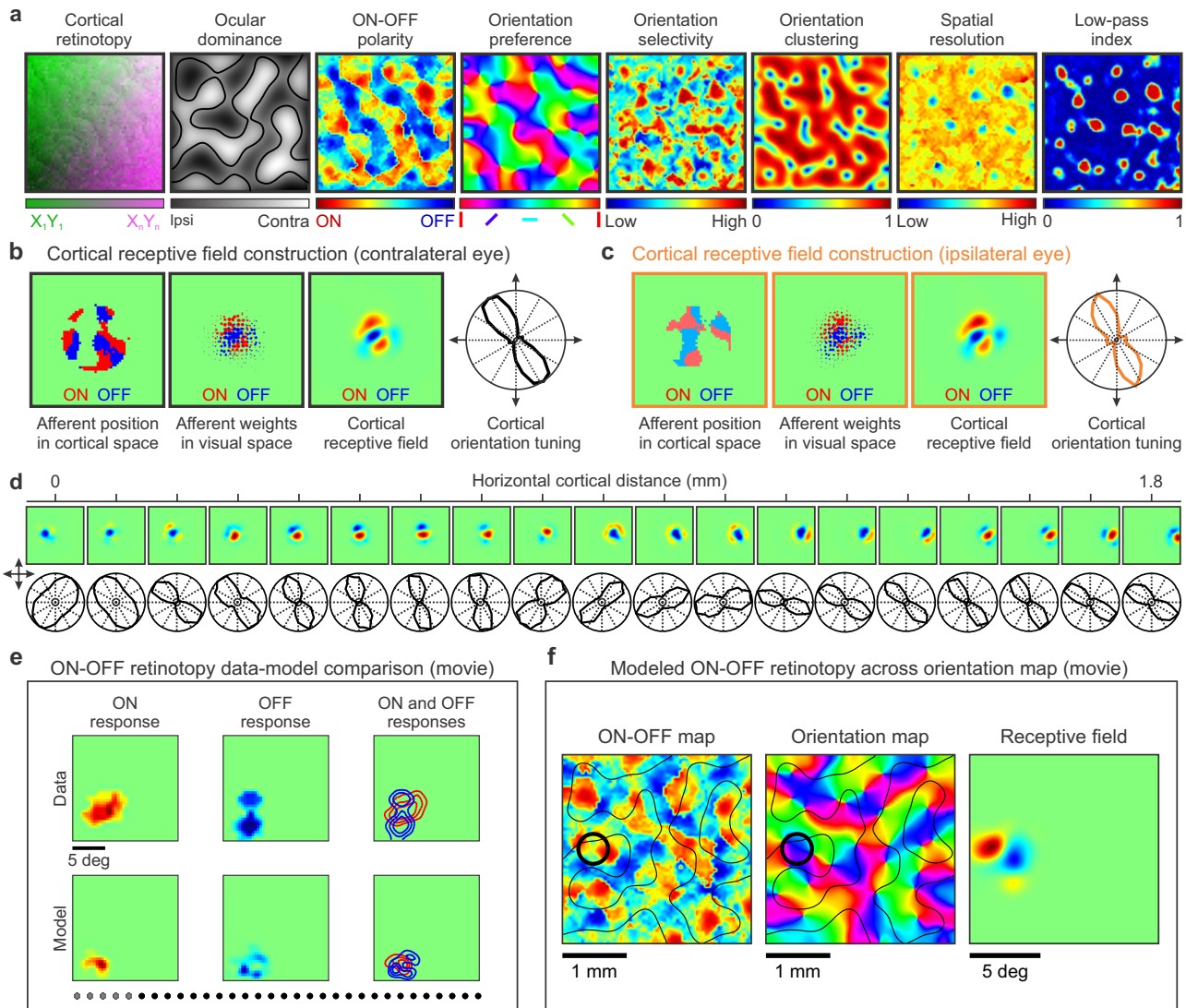

**Fig. 6 Model outputs. Visual maps for multiple stimulus dimensions, cortical receptive fields and neuronal response properties. a** The model generates maps for multiple stimulus dimensions illustrated, from left to right, for retinotopy, ocular dominance, ONOFF polarity, orientation preference, orientation selectivity, orientation clustering, spatial resolution (highest spatial frequency generating half-maximum response), and low-pass index (response at lowest spatial frequency divided by maximum response). **b** The model output also provides the position in cortical space of the main afferent axons converging at the same cortical pixel (left square panel), the afferent receptive field positions and synaptic weights in visual space (middle square panel, dot position and size), and the cortical receptive field and orientation tuning resulting from afferent convergence (for the contralateral eye). **c** Same as (**b**) for the ipsilateral eye (notice the precise binocular match of orientation preference). **d** It also generates cortical receptive fields (top) and orientation tuning (bottom) for each pixel of the visual map. As in experimental measures[28], the receptive fields change with cortical distance in position (retinotopy), ONOFF dominance, orientation preference, and subregion structure (top). The orientation preference and orientation selectivity also changes systematically with cortical distance (bottom). **e** Model-data comparison of ON-OFF retinotopy changes with cortical distance (see movie Fig. 6e). The gray dots at the bottom illustrate the cortical pixels included in the receptive field average (average along a cortical distance of 400 microns, distance between pixels 100 microns). Notice that ONOFF retinotopy systematically changes along a diagonal line in visual space. However, the diagonal line is not straight; ON and OFF retinotopy move around and intersect with each other, as expected from ONOFF afferent sorting. **f** Model simulations illustrating the relation between ONOFF dominance (left), orientation preference (middle), and receptive field structure (right, see movie Fig. 6f).

average control error. Based on these results, we conclude that the simulated cortical maps reproduce the differences across species of experimentally-measured maps.

**Combined visual cortical topography for multiple stimulus dimensions**. The theory also predicts that the map gradients for orientation selectivity, low-pass filtering, spatial resolution and orientation local homogeneity should be all highly correlated because they all originate from changes in ON-OFF response

balance (Fig. 1d–f). Unfortunately, there are just a few studies that measured visual cortical topography for more than two stimulus dimensions[6,54–56]. Therefore, to increase the biological constraints of the model, we performed multielectrode recordings from horizontal tracks of cat visual cortex and used these recordings to measure systematic changes in spatial position, ocular dominance, ON-OFF receptive-field structure, orientation preference, orientation selectivity, homogeneity of orientation preference, spatial-frequency resolution (highest spatial frequency generating half of maximum response, S50), and spatial-frequency low-pass filtering.

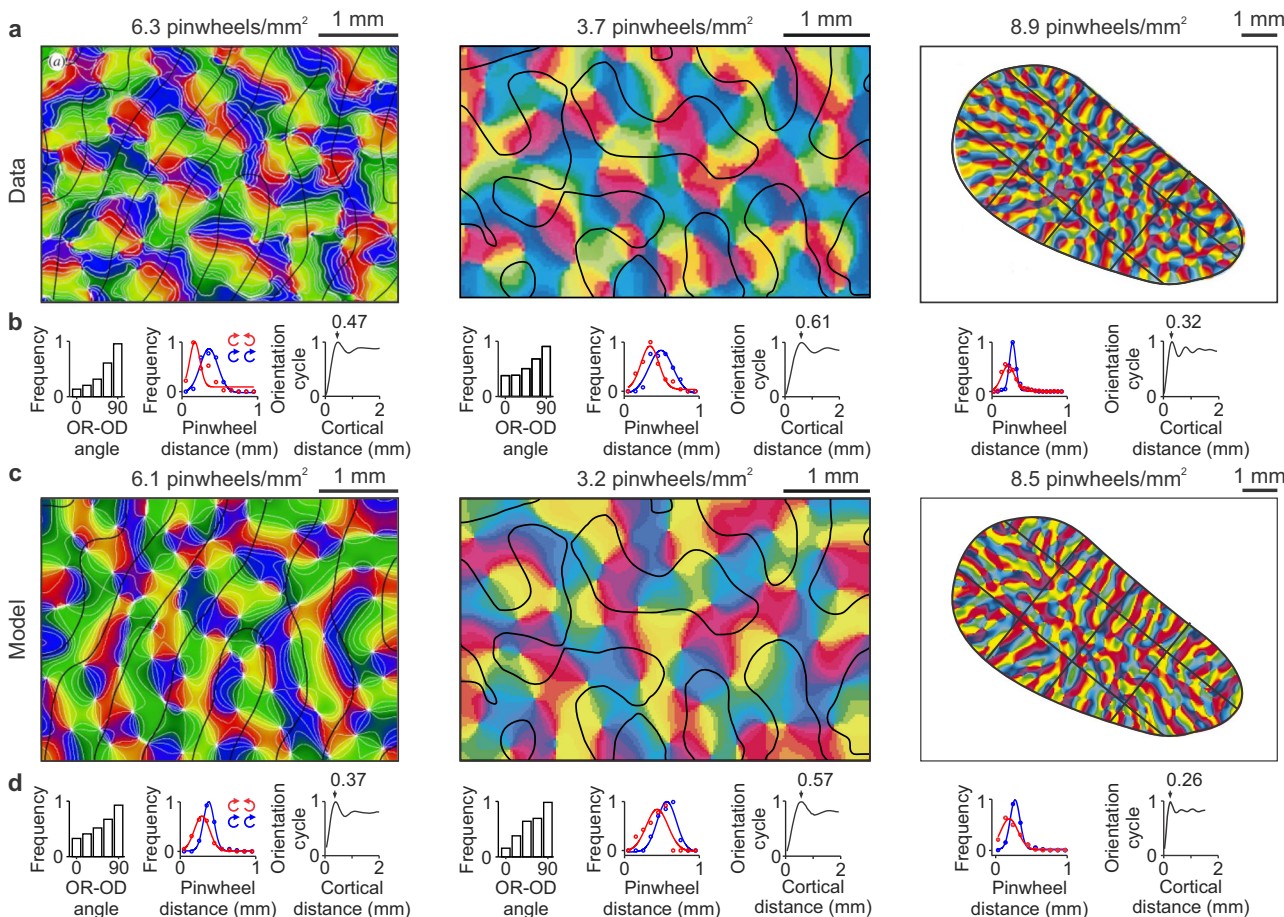

**Fig. 7 Model simulations of cortical orientation maps in three different species. a** Maps of stimulus orientation and pinwheel density experimentally measured in the visual cortex of a macaque (left), a cat (middle) and a tree shrew (right). Reproduced with permission from Horton and Adams, 2005 (original data from Blasdel, 1992), Hubener et al., 1997, and Bosking et al., 1997. The black lines illustrate ocular dominance borders in the macaque and cat visual cortex and iso-retinotopic lines in the tree shrew. **b** Statistics of the three orientation maps from (**a**) showing, for each animal, from left to right, distribution of angles between iso-orientation lines and ocular-dominance borders (in macaque and cat, tree shrews do not have ocular dominance borders), distance between pinwheels with same (blue) or opposite (red) rotations, and orientation map periodicity. **c–d** Model simulations of the experimental data from (**a–b**) (using the same color map as the published data). Source data are provided as a Source Data file.

The theory predicts that, as ON and OFF responses become balanced in strength, the ON-OFF spatial antagonism should increase (Fig. 1f). In turn, the increased ON-OFF spatial antagonism should reduce the range of stimulus orientations driving a response (enhance orientation selectivity), shrink the width of the receptive-field-subregions (enhance spatial resolution), and reduce the cortical responses to low spatial frequencies (enhance spatial frequency selectivity)[5].

Our experimental measurements confirm these predictions (Fig. 8). In horizontal recording tracks performed with multi-electrode arrays in cat visual cortex, cortical spatial resolution systematically increased with cortical distance over a few hundred microns (Fig. 8a). Moreover, the increases in spatial resolution were strongly associated with increases in orientation selectivity (measured as a reduction in circular variance) and orientation-preference homogeneity (Fig. 8a). Such relations could be demonstrated in horizontal tracks running parallel to an ocular dominance band (Fig. 8a) and in tracks crossing the ocular dominance border (Fig. 8c). Moreover, in tracks crossing the ocular dominance border, there was a strong association between response increases to low spatial frequencies and increases in eye dominance measured as an increase in ocular dominance index, ODI (Fig. 8c). Finally, as predicted by the theory, we also found a correlation between orientation tuning and ON-OFF response balance (Fig. 8e). The

model replicated all these experimental measurements (Fig. 8b, d, f) and the correlations among gradients for different stimulus dimensions measured within single horizontal tracks in individual animals (Fig. 9a–d) and in multiple recording tracks from different animals pooled together (Fig. 9e, f). The model also replicated the sign of the correlations, data scatter, the distribution of slopes from different combinations of stimulus dimensions (Fig. 9g–j) and the orthogonal relations between map gradients for different stimulus dimensions (Fig. 9k, see more examples in Supplementary Fig 6). Replicating the data scatter for spatial resolution was particularly challenging and deserves to be described in some more detail. Our measurements in cat visual cortex demonstrate that spatial resolution (SF50) can change by almost 1 cpd within just 400 microns of horizontal cortical distance (Fig. 9a). However, without ON-OFF synaptic competition, the largest changes simulated by the model were four times smaller (0.25 cpd instead of 1 cpd). Therefore, the synaptic competition between ON and OFF afferents is probably important to make the ON-OFF response balance more effective at reducing the width of the receptive-field-subregions and increase the range of spatial resolution within an orientation domain.

To further quantify the model performance, we simulated 1000 recording tracks and counted those that replicated the correlations measured in our experiments. For statistical comparison, we

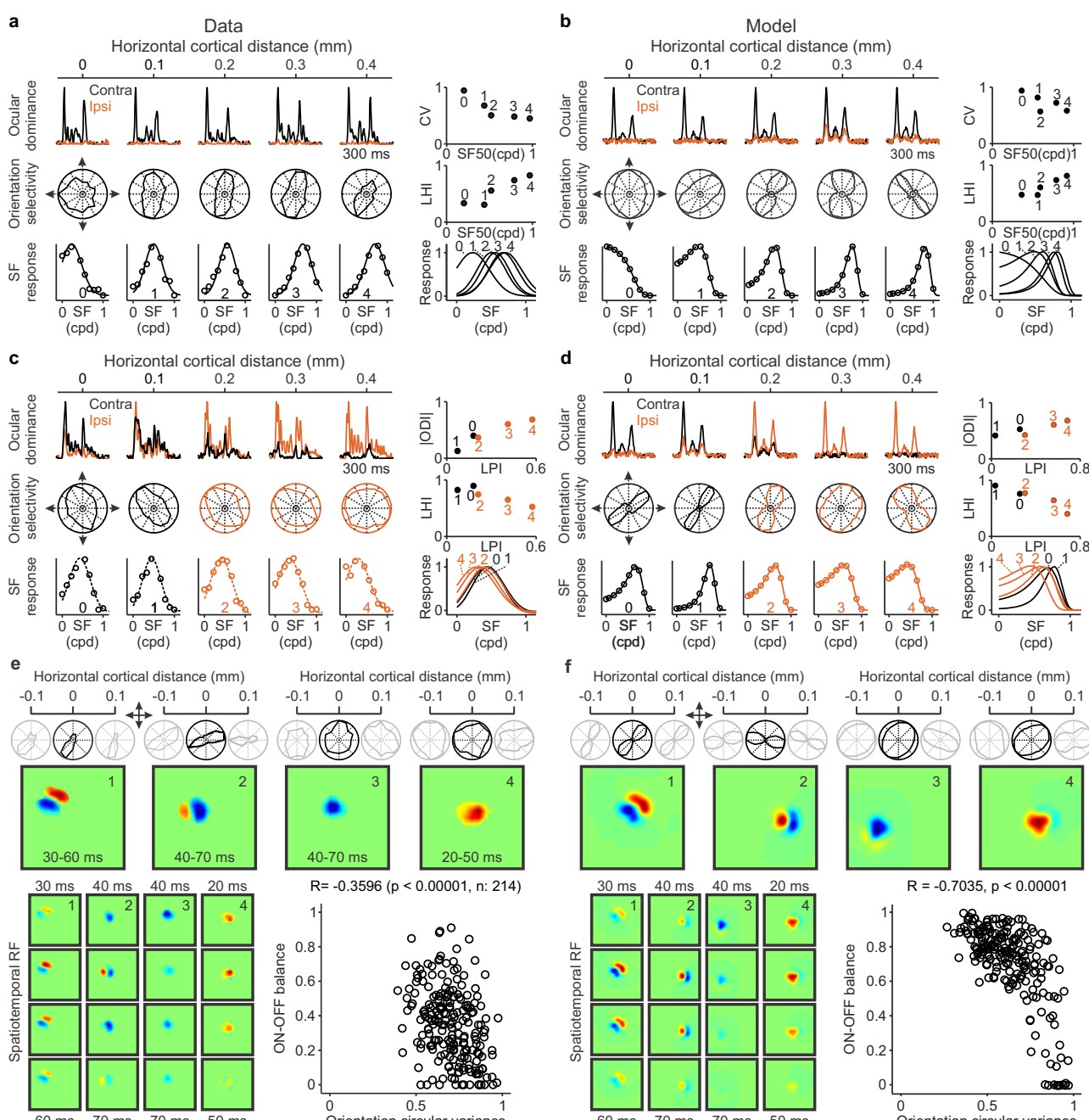

**Fig. 8 Measurements of multiple stimulus dimensions along horizontal tracks of cat visual cortex and model simulations. a** Example recording from a horizontal track parallel to an ocular dominance band. From top to bottom, the left panel shows responses to static gratings presented through the contralateral (black) or ipsilateral eyes (orange), orientation tuning, and spatial frequency tuning. The spatial resolution (highest spatial frequency generating half-maximum response, SF50) increases from cortical site 0 to 4, and this increase is associated with a decrease in orientation selectivity measured as circular variance (CV, right top) and an increase in orientation clustering (right middle) measured as a local homogeneity index (LHI). The right bottom panel shows the five spatial frequency curves from the left panel superimposed to facilitate their comparison. **b** The model replicates the data illustrated in (**a**). **c** Example recording from a horizontal track crossing an ocular dominance border (same organization as in **a**). The response strength to low spatial frequencies (LPI: low pass index) increases from cortical sites 0 to 4, and this LPI increase is associated with an increase in ocular dominance (right top, ODI: ocular dominance index) and a decrease in LHI (right middle). **d** The model replicates the data illustrated in **c**. **e** Top. Orientation tuning of four cortical sites (black) and their adjacent sites 100-microns apart (gray). Middle. Receptive fields with greater ON-OFF response balance (left) have higher orientation selectivity and local homogeneity index than those that are ON or OFF dominated (right). Bottom left. Receptive fields shown at different time delays from the stimulus. Bottom right. Significant correlation between orientation selectivity and ON-OFF balance (rank correlation with Matlab 'corr' function, R: correlation coefficient, p: probability of no correlation). **f** The model replicates the experimental measures in **e**. Source data are provided as a Source Data file.

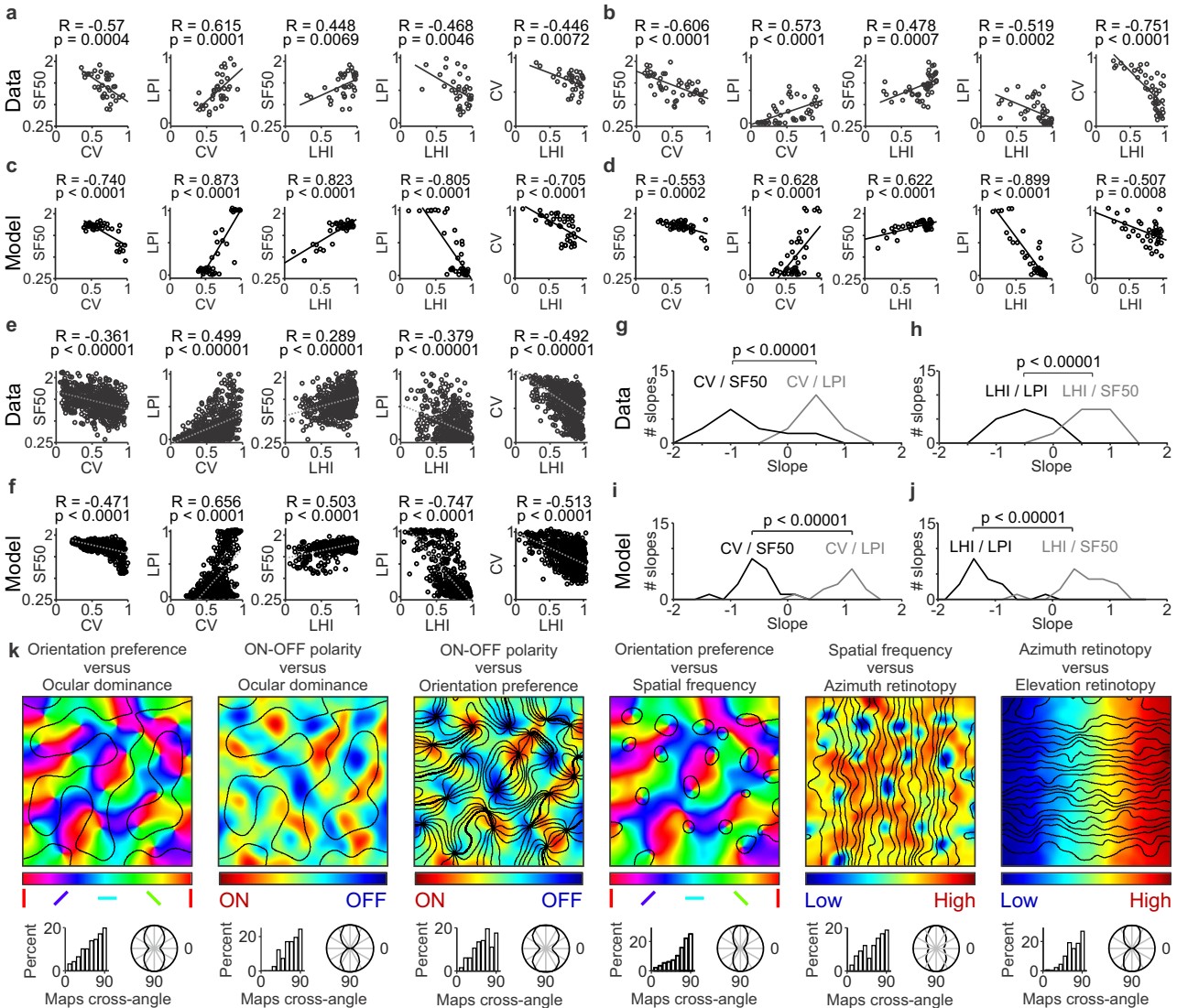

**Fig. 9 Correlations among multiple stimulus parameters represented in the visual cortical map. a** Example correlations between multiple pairs of stimulus parameters measured in a single horizontal track of cat visual cortex ($n = 35$), including orientation selectivity measured as circular variance (CV), spatial resolution measured as the highest spatial frequency that generates half-maximum response (SF50), response strength to low spatial frequencies measured as low-pass index (LPI), and cortical clustering of orientation preference measured as a local homogeneity index (LHI). From left to right, the figure shows significant correlations between pairs of stimulus parameters. In all panels, r and p values were calculated with rank correlation (Matlab 'corr' function). **b** Example single-track correlations measured in a different animal ($n = 47$). **c–d** The model replicates single-track correlations as illustrated in two different simulations (**c** and **d**). The number of data points in each plot equals the average number of data points illustrated in (**a–b**). **e** Correlations from all single-track recordings obtained in multiple animals ($n = 8$ animals, 17 single tracks, 633 data points). **f** The model replicates the experimental correlations illustrated in (**e**), $n = 20$ single tracks, 800 data points. **g** Distribution of correlation slopes measured in cat visual cortex between CV and SF50 (black line), and CV and LPI (gray line). The CV/SF50 slopes were significantly more negative than the CV/LPI slopes. **h** Distribution of correlation slopes measured in cat visual cortex between LHI and LPI (black line) and between LHI and SF50 (gray line). The LHI/LPI slopes were significantly more negative than the LHI/SF50 slopes ($n = 8$ animals, 17 single tracks for all slope measures in **g–h**). **i–j** The model replicates the differences in slopes measured experimentally ($n = 20$ single tracks). Two-sided Wilcoxon tests for all slope comparisons. **k** Model simulations of the relations between paired combinations of maps. For each map combination, the top panel shows the pair of superimposed maps (color scale for the map listed first and contour plot for the second). The bottom panel shows the intersection angles between the two maps in a histogram (left) and a polar plot (right). Source data are provided as a Source Data file.

performed these simulations with and without afferent sorting for ocular dominance and ON-OFF contrast polarity. If the criteria for track selection required to reproduce just a few weak correlations, many tracks passed the criteria even when the afferents were not sorted. However, as the criteria for track selection became stricter and required to reproduce a larger number of strong correlations with the correct sign and range, the model could not reproduce any of the tracks without afferent sorting (Supplementary Table 2).

Based on these results, we conclude that afferent sorting is needed to explain the combined correlated topography of multiple stimulus dimensions measured in cat visual cortex.

**Main theory propositions.** Taken together, our experimental and modeling results provide support for a theory that we briefly summarize in three propositions.

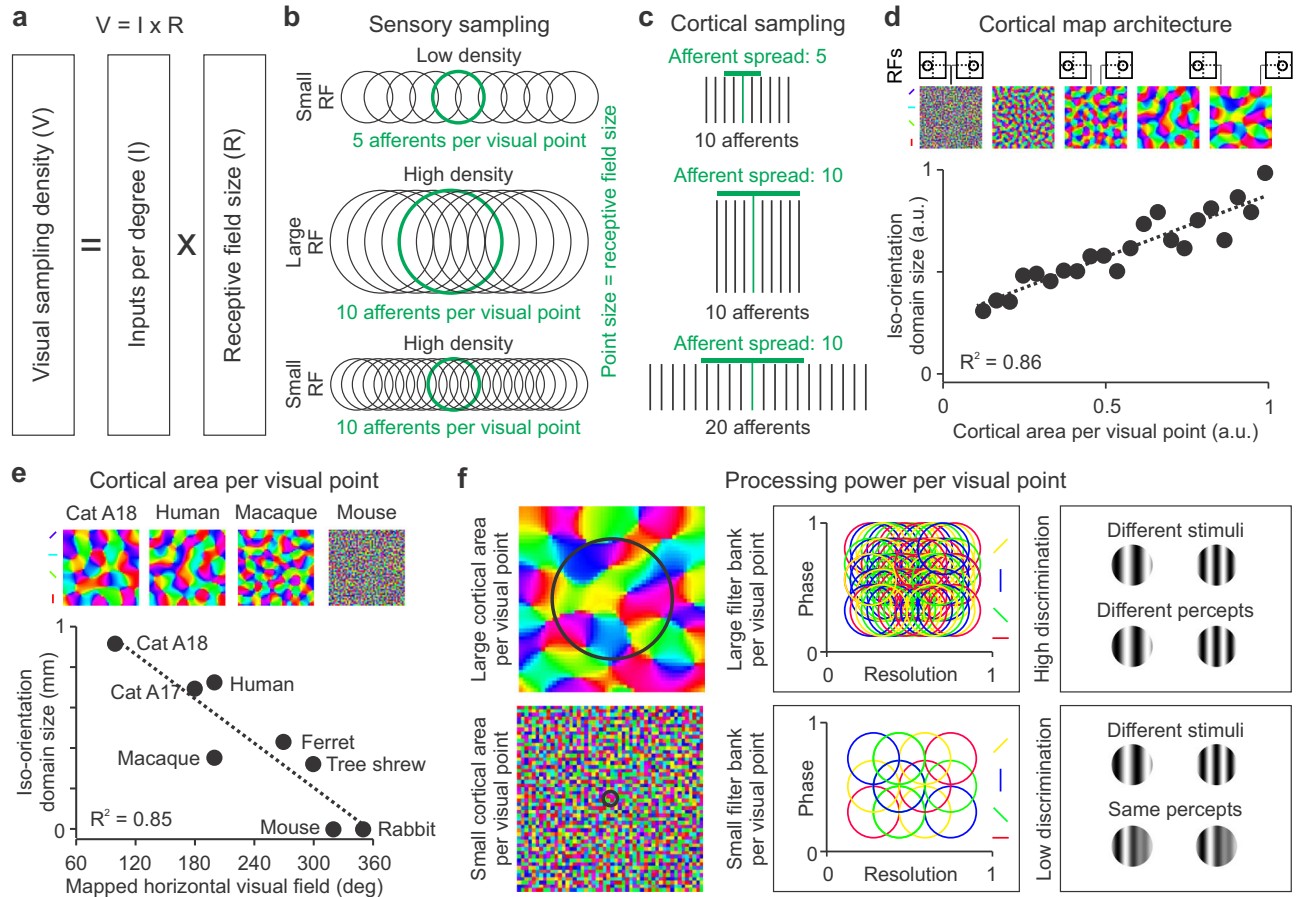

**Fig. 10 Sensory cortical maps maximize the balance between spatial resolution and cortical processing power. a** Cartoon illustrating the definition of visual sampling density (V) as the multiplication of receptive field size (R) times input afferents per degree (I). **b** Cartoon illustrating a horizontal rectangle of visual space sampled by 5 afferents with small receptive fields (top), 10 afferents with large receptive fields (middle) and 20 afferents with small receptive fields (bottom). The central receptive field (in green) illustrates the size of the visual point (point size = receptive field size) and the number of afferents per visual point. **c** Cartoon illustrating the spread of the afferent axonal arbor for each of the three examples shown in (**a**). Because the axon arbor spreads to cortical locations with overlapping retinotopy, the spread is larger when there is an increase in receptive field size (compare top and middle) or afferent density (compare top and bottom). Therefore, the size of the cortical region sampled by each afferent (axon arbor spread) depends on both receptive field size and input density. **d** In the model, the size of the iso-orientation domains increases with the size of the iso-retinotopic domains (cortical area representing the same visual point), as shown in a scatter plot (bottom, a.u.: arbitrary units) and example orientation maps (top). The size of the cortical area representing the same visual point is illustrated by two vertical lines separating regions with non-overlapping receptive fields (RFs). **e** The model predicts that the size of iso-orientation domains should decrease when the cortex has to map a large visual field and devote smaller areas to represent each visual point (top). Size measurements of iso-orientation domains and horizontal visual fields mapped in different cortical areas confirm this prediction. **f** In the model, larger iso-orientation domains (left) accommodate larger filter banks per visual point that sample spatial-frequency/phase combinations more finely (middle). As a consequence, the model predicts that cortices with larger iso-orientation domains should allow higher stimulus discrimination per visual point. Source data are provided as a Source Data file.

1) Visual sampling density. The diversity of visual cortical maps across species originates from variations in the thalamic afferent density sampling visual space, which we define as the number of inputs (afferents) per degree times the receptive-field size (Fig. 10a). This proposition indicates that an evolutionary increase in visual acuity requires not only a reduction in receptive field size but also a massive increase in number of afferents to keep sampling density high (Fig. 10b, c).

2) Afferent sorting. As afferent sampling density increases, the size of visual cortex representing the same visual point also increases to accommodate the larger number of afferents with overlapping receptive fields. In turn, the larger cortical region allows afferents and cortical targets with the same retinotopy to segregate by other stimulus dimensions that are not just spatial position (Fig. 10d).

3) Afferent convergence and synaptic competition. Afferents with similar retinotopy converge at the same cortical point, but

the thalamocortical convergence is limited by synaptic competition. As afferent density increases, the convergence becomes constrained by more stimulus dimensions that are not just retinotopy and, as a consequence, each axon arbor occupies a smaller portion of the cortical area representing each visual point.

## Discussion

We propose a theory of cortical map formation that explains the natural diversity of visual cortical maps based on variations in afferent sampling density. The theory is illustrated with an afferent-density model that follows very closely the main developmental stages of the early visual pathway including the afferent sorting. Across the animal kingdom, primary sensory cortical areas sort their thalamic afferents by their dominant sensory modality that can include retinotopy[22,25], somatotopy[26], sound frequency[57], gustotopy[58] or electroreception[59]. Moreover, in

primary visual cortex, mammals with extensive binocular vision sort afferents by multiple stimulus dimensions that include not only retinotopy but also eye input[60] and ON-OFF polarity[61–64]. Therefore, afferent sorting is likely to play a major role in cortical map development and the wiring of thalamocortical circuits[23–26,57].

The theory proposes that cortical map diversity originates from variations in visual sampling density (V), which we define as the number of inputs per degree (I) times their receptive field size (R), V = I x R (Fig. 10a). As with the Ohm's law, the density of visual sampling acts as a driving force (voltage) proportional to input density (current) and receptive field size (resistance). In the theory of cortical map formation, the effect of the driving force is to enlarge the cortex and make neurons segregate by multiple stimulus dimensions. The driving force can be made stronger by increasing the afferent density and/or making the receptive field size larger (Fig. 10b). The increase in afferent sampling density is associated with an enlargement of the afferent axon arbor and the cortical iso-retinotopic region representing the same visual point (Fig. 10c). In turn, as the iso-retinotopic region becomes larger, the multiple afferents and cortical neurons with the same retinotopy segregate by multiple stimulus dimensions such as eye input, ONOFF polarity, and orientation, while reducing the size of their axon arbors through synaptic competition. The theory predicts that the size of the iso-retinotopic domains and/or afferent clusters should determine the size of the iso-orientation domains (Fig. 10d). Consistent with this prediction, iso-orientation domains become smaller as the cortical space per visual point decreases to accommodate a larger visual field (Fig. 10e). Also consistent with this prediction, iso-orientation domains increase in size with the number of afferents (e.g. from macaque to human) or receptive field size (e.g. from cat area 17 to cat area 18).

The theory also predicts that animals with the largest iso-orientation domains should have the largest cortical areas representing the same retinotopy and, therefore, the greatest processing power per visual point. We define processing power as the number of cortical neurons available to discriminate stimuli centered at the same visual point, a term that is equivalent to filter bank size in signal processing (Fig. 10f). Because the size of the perceived visual point is associated with the afferent receptive field size, processing power can be increased by changing afferent density and/or receptive field size. This flexibility of design allows each species to reach a different compromise between processing power and spatial resolution. For example, the large iso-orientation domains and afferent receptive-fields of cat area 18 can maximize processing power per visual point at the expense of reducing spatial resolution. This may be a good compromise for an area specialized in visual motion that requires discriminating fine movements of relatively large targets[65]. Conversely, macaque area V1 can afford losing some processing power per visual point (smaller iso-orientation domains than cat area 18) to achieve a much higher spatial resolution (many afferents with much smaller receptive fields). This may be a good compromise for an area specialized in visual detail. Human area V1 can maintain the high spatial resolution of macaques while approaching the processing power of cat area 18 by dramatically increasing afferent density[17] (Fig. 10e). Therefore, our theory of cortical map formation not only provides insight about the organization of visual maps from different species but also about the diversity of cortical map functions. The theory also predicts that the multiple map gradients for different stimulus dimensions should be closely related and, therefore, the map of one-dimension can be used as input to reproduce the maps for other dimensions. If ON-OFF retinal mosaics become available through imaging or other methods in the future[33,66], the theory provides a path towards

reconstructing the multidimensional map of an individual brain to guide the implantation of cortical prosthesis[67].

## Methods

**General structure of the computational model**. We organize the model in three stages: retinal development, development of the cortical subplate, and development of the visual cortex. At the stage of retinal development, we simulate the mosaics of ON and OFF retinal ganglion cells from the two eyes, which are ipsilateral and contralateral to the cortical subplate from one brain hemisphere. The ON and OFF retinal mosaics are then fed into the dorsal Lateral Geniculate Nucleus (LGN) of the thalamus and, in turn, the thalamic afferents are fed into the cortical subplate. At the stage of cortical subplate development, we sort the thalamic afferents by retinotopy, eye input and ON-OFF polarity. At the last stage of visual cortical development, we spread the axonal arbors of the thalamic afferents in the developing cortex and adjust their synaptic weights based on visual experience. Each stage of the model aims to reproduce as closely as possible the experimental data that is currently available, including the new measurements that we report in this paper.

**Model stage 1. Retinal development**. The retinal simulations replicate the ON and OFF retinal mosaics from the retinal ganglion cell type that dominates the projection to the LGN, which is the midget cell in macaques and beta cell in cats. The model can incorporate other cell types (e.g. parasol cells in macaques and alpha cells in cats) or multiple cell types. However, for simplicity, it assumes that the cortical topography for retinotopy, eye input and ON-OFF polarity is determined by the inputs of the cell type that dominates the retino-geniculate-cortical projection in number and strength. In the rest of the paper, we use the generic terms ON and OFF retinal mosaics to refer to this dominant retinal ganglion cell type in each species.

The simulation of ON and OFF retinal mosaics reproduces as close as possible the ON and OFF retinal arrays experimentally measured in different species. Consistently with the experimental data, the simulated ON and OFF retinal mosaics are independent, regularly spaced, randomly jittered, and do not allow two neurons to occupy the same spatial position[33,42,68,69]. The algorithm that simulates the retinal mosaics is the same for the two eyes but each eye mosaic is different because the model randomly jitters the position of each retinal ganglion cell. We start generating ON and OFF retinal mosaics by simulating a grid that samples retinal space at a fixed inter-cell distance. The fixed inter-cell distance (dx, dy) is a variable that changes the average retinal ganglion cell density separately for ON and OFF cells and separately along the azimuth (**RX**) and elevation (**RY**) retinal axis. For example, if (dx, dy) values are smaller for OFF than ON cells, the cell density is higher for OFF than ON cells. If dx values are smaller than dy values, the cell density is higher along the azimuth than elevation axes.

After the cell grid is created, the model jitters the position of each retinal ganglion cell with a random number taken from a uniform distribution whose range can be specified separately for the azimuth (±rjx) and elevation axis (±rjy). The absolute number of cells of each type along the azimuth (nx) and elevation axes (ny) can also be changed together with the inter-cell distance to simulate retinas of different sizes and cell densities. We define the original position of each retinal ganglion cell, **Ret**$_{op}$, as the (rx, ry) grid position within the azimuth (**RX**) and elevation (**RY**) retinal axes times the inter-cell distance (dx, dy) plus the position jitter (jx, jy), as shown in Eq. 1. The grid position is given by two integer numbers (rx, ry), each one ranging from one to the total number of cells from each type, ON or OFF, along the azimuth or elevation axes. The integer number increases from left to right within the azimuth retinal axis (**RX**) and from bottom to top within the elevation axis (**RY**). The inter-cell distance (dx, dy) can take any value in pixels to match the retinal cell density that we want to simulate. For example, to simulate the cat retina, we use an inter-cell distance of five pixels that corresponds to 110 microns (22 microns/retinal pixel).

$$\mathbf{Ret_{op}}(rx, ry) = [rx \times dx + jx, ry \times dy + jy] \begin{cases} 1 \le rx \le nx \\ 1 \le ry \le ny \\ -rjx \times dx \le jx \le rjx \times dx \\ -rjy \times dy \le jy \le rjy \times dy \end{cases} \quad (1)$$

In the final stage of retinal development, the model simulates interactions between neighboring ON and OFF retinal ganglion cells to adjust their final position. This process prevents ON and OFF retinal ganglion cells from being at the same retinal position. It also allows the model to match the statistics of the retinal mosaic that we want to simulate (e.g. average and standard deviation of the distance distribution between pairs of ON and OFF cells). We define an ONOFF pair (p) as the pair of ON and OFF cells separated by the shortest distance within each retinal location. We simulate ON-OFF retinal interactions by scaling the distance between each ON-OFF cell pair. The retinal scaling changes the array maximum and minimum ON-OFF distance (Maxd, Mind) to match the maximum and minimum distance that we want to replicate (Maxdr, Mindr). We calculate the new ON-OFF scaled distance for each cell pair, ONOFFdnew (p), by adjusting the

ON-OFF distance before scaling, ONOFFd, as shown in Eq. 2.

$$\mathbf{ONOFFdnew}(p) = (\mathbf{ONOFFd}(p) - \text{Mindr})\frac{\text{Maxdr} - \text{Mindr}}{\text{Maxd} - \text{Mind}} + \text{Mindr} \quad (2)$$

The model calculates the final [rx, ry] position of each retinal ganglion cell by adding the original [rx, ry] position (**Ret**$_{op}$) to the adjustment of [rx, ry] position resulting from the ON-OFF interactions. The position adjustment is done along the angle ($\Theta$) between the line connecting the cell pair and the azimuth axis (Eq. 3). Because the interactions move ON and OFF cells within each pair in opposite directions, the adjustments are opposite in sign (sign) for ON and OFF cells.

$$\mathbf{Ret}(\text{rx}, p) = \mathbf{Ret_{op}}(\text{rx}) + \text{sign} \times \tfrac{\mathbf{ONOFFdnew}(p) - \mathbf{ONOFFd}(p)}{2} \times \cos\theta \left\{ \begin{array}{l} \text{sign} = 1 \text{ for ON} \\ \text{sign} = -1 \text{ for OFF} \end{array} \right\}$$

$$\mathbf{Ret}(\text{ry}, p) = \mathbf{Ret_{op}}(\text{ry}) + \text{sign} \times \tfrac{\mathbf{ONOFFdnew}(p) - \mathbf{ONOFFd}(p)}{2} \times \sin\theta \left\{ \begin{array}{l} \text{sign} = 1 \text{ for ON} \\ \text{sign} = -1 \text{ for OFF} \end{array} \right\}$$

$$(3)$$

Retinal receptive fields (**RRF**) are modeled as two-dimensional Gaussian functions with a center at retinal position (rx, ry) and a size equal to one standard deviation ($\sigma$), as shown in Eq. 4. For simplicity, all simulations of cat visual cortex use a constant retinal receptive field size with a standard deviation of 2–5 retinal pixels (44–110 microns, 22 microns/retinal pixel, 0.2–0.4 degrees for cat central retina with 250 microns/degree).

$$\mathbf{RRF}(\text{rx}, \text{ry}, \sigma) = \frac{1}{2\pi\sigma^2} \exp\left(-\frac{(\mathbf{RX} - \text{rx})^2 + (\mathbf{RY} - \text{ry})^2}{2\sigma^2}\right) \quad (4)$$

In cats and primates, the receptive field of each LGN cell closely resembles the receptive field of a dominant retinal input. Therefore, the model uses a neuronal divergence equal to one and makes the receptive field of each thalamic cell (**TRF**) identical to the receptive field of its retinal input (**RRF**), as shown in Eq. 5.

$$\mathbf{TRF}(\text{rx}, \text{ry}, \sigma) = \mathbf{RRF}(\text{rx}, \text{ry}, \sigma) \quad (5)$$

Future versions of the model could increase the number of thalamic afferents with overlapping receptive fields by using larger values of neuronal divergence or incorporating additional retinal mosaics. The neuronal divergence could be implemented by interpolating the receptive field array of the simulated ON-OFF retinal mosaic. For example, if the neuronal divergence is two, half of the thalamic afferents could receive only one input and have identical receptive fields to the retinal mosaic. In addition, the other half could receive two inputs and have receptive fields calculated as a weighted average of two input receptive fields. For example, if two neighboring ON retinal ganglion cells make connection with one ON thalamic neuron, the model can first calculate the thalamic receptive field as the receptive-field sum of the two retinal inputs (RFsum = RRF1 + RRF2). Then, it can calculate the weight of each input (w1, w2) based on the overlap between the receptive field from each input and the receptive-field sum. Then, the final thalamic receptive field can be computed as a weighted sum of the receptive fields from the retinal inputs (TRF = w1 x RRF1 + w2 x RRF2). The receptive field overlap can be measured by normalizing each receptive field to an absolute maximum of 1 (−1 for OFF and +1 for ON) and then calculating the dot product between the receptive fields averaged over all pixels and rectifying the average so that negative weights equal zero. The model assumes that the strongest retino-geniculo-cortical connections dominate the development of cortical topography for retinotopy, eye input and ON-OFF polarity. Therefore, a neuronal divergence equal to one also provides a simple approach to build cortical topography purely based on the strongest retinogeniculate connections.

**Model stage 2. Development of cortical subplate (thalamic afferent sorting by retinotopy).** The model divides the cortical subplate into equally spaced square locations and accommodates one thalamic afferent in each location. The number of locations within the cortical subplate equals the number of thalamic afferents. In turn, the number of thalamic afferents equals the number of ganglion cells from the two eyes projecting to one LGN times the neuronal divergence, which is one if each retinal ganglion cell connects only to one thalamic afferent. The separation between two square locations within the cortical subplate is constant and equal to 50 microns. This constant separation makes the size of the cortical subplate directly related to the number of thalamic afferents. For example, a patch of the cortical subplate that receives 180,000 ON and 180,000 OFF thalamic afferents has $600 \times 600 = 360,000$ square locations and measures $30,000 \times 30,000$ square microns. In contrast, a patch of the cortical subplate that receives 100 times less afferents (1800 ON and 1800 OFF) is 10 times smaller ($60 \times 60 = 3600$ cortical locations) and measures only $3000 \times 3000$ square microns. The model assumes that the most important factor in generating cortical orientation maps is the number of thalamic afferents sampling the same point of visual space. If this number is small, the cortex can only devote a small region to represent each visual point. However, if the cortical area per visual point is large, the number of afferents with overlapping receptive fields is also large and the cortex can sort afferents by eye input and ON-OFF polarity in addition to retinotopy.

The model sorts the afferents in three sequential stages. At the retinotopy stage, it sorts afferents by retinotopy only. At the eye-input stage, it sorts afferents with similar retinotopy by eye input. At the ON-OFF polarity stage, it sorts afferents with similar retinotopy and eye input by ON-OFF polarity. This sequence of

afferent sorting (retinotopy → eye input → ON-OFF) has been demonstrated in the ferret visual thalamus[5,39] and the model assumes that it is preserved in the retino-thalamo-cortical pathway of all mammals. There is evidence that retino-thalamo-cortical connections segregate by retinotopy before eye input in cats, ferrets, and macaques[35,39,70]. There is also evidence that the segregation by eye input can be prevented at very early stages of development by ablation of cortical subplate[34,37,38]. There is also evidence that ON-OFF segregation is a slow process that can last several months after birth in the cat retina[71] and that occurs after the segregation by retinotopy and eye input in the ferret thalamus[39].

The model simulates the retinotopy sorting by assigning a [x, y] position within the cortical subplate (**CP**) to each afferent (i) based on its [rx, ry] retinotopy. Afferents with the lowest values of [rx, ry] retinotopy are assigned the lowest [x, y] values of cortical position (the top left corner of the cortical subplate) and those with the highest [rx, ry] values are assigned the largest [x, y] values (bottom right corner of the cortical subplate, Eq. 6). The maximum index is equal to the total number of afferents (nx x ny) along the azimuth and elevation axes. Because afferents from the two eyes can have overlapping receptive fields and the same retinotopy, the order of afferent assignment is randomized and afferents reaching the cortex earlier get the best retinotopic match. After the retinotopic sorting is complete, afferents with nearby receptive field positions are located in nearby cortical locations and those with receptive fields far apart are located in cortical locations that are also far apart.

$$\mathbf{CP}_i(x, y, \text{rx}, \text{ry}) = \mathbf{TRF}_i(\text{rx}, \text{ry}) \left\{ \begin{array}{l} 1 \leq i \leq \text{nx} \times \text{ny} \\ 1 \leq x \leq \text{nx} \\ 1 \leq y \leq \text{ny} \end{array} \right. \quad (6)$$

**Model stage 2. Development of cortical subplate (thalamic afferent sorting by eye input).** After finishing the retinotopy sorting, the model starts sorting afferents by eye input using a variation of an algorithm developed by Swindale[13,29]. As in many models of cortical maps, the algorithm performs a convolution between a difference of Gaussians and a matrix of cortical values. However, unlike previous models, our convolution is only used to simulate an afferent relocation within the cortical subplate through a movement in one of eight possible directions (e.g. one pixel to the left if the convolution value is largest in the left region of the afferent). The afferent movement can be interpreted as a growth cone changing positions within the cortical subplate or as the pruning of an axon arbor at later developmental stages that effectively changes its central position in cortex. The goal of the model is not to simulate how afferents segregate in the cortex but to simulate the contribution of afferent segregation to the construction of cortical maps.

The model segregates afferents within the cortical subplate (**CP**) as follows. It starts by assigning one of two possible values to each afferent based on their eye of origin (eye), 1 for the contralateral eye and −1 for the ipsilateral eye. Then, the binary cortical subplate is convolved with an afferent sorting filter (**ASF**) that has a center-surround organization. The filter center attracts afferents of the same type while the surround repels afferents of different type making afferents of the same type to become close together. The center of the filter is always circular and its radius is defined by a single standard deviation ($\sigma_c$). The filter surround can be circular or elongated and the radii are described by two different standard deviations ($\sigma_{sx}$, $\sigma_{sy}$). The surround along the filter longest axis is two times larger than the center size (e.g. [center, x-surround, y-surround] = [0.5, 1, 1] mm for simulations of cat visual cortex and [0.5, 0.5, 1] mm for simulations of macaque visual cortex). The afferent sorting filter is modeled as a difference of two multivariate Gaussian functions (Eq. 7). Each of the functions is described by the [x, y] position of the selected afferent within the antero-posterior (**AP**) and medio-lateral (**ML**) axes of cortical space, two vectors that describe a local subset of x (**X** vector) and y positions (**Y** vector) influencing the afferent sorting, and the standard deviations of the filter center ($\sigma_c$) and surround ($\sigma_{sx}$, $\sigma_{sy}$). The filter angle ($\rho$) is described as the dot product of the **X** and **Y** vectors divided by their magnitudes (i.e. the square root of the sum of the squared elements of each vector). Circular filters segregate afferents in beaded regions resembling the ocular dominance segregation of cats. Elongated filters segregate afferents in stripes resembling the ocular dominance segregation of macaques and humans. Filters with large diameter segregate afferents in larger ocular dominance domains than filters with smaller diameter[29].

$$\mathbf{ASF}(x, y, \sigma_c, \sigma_{sx}, \sigma_{sy}) = \mathbf{f_{center}}(x, y, \sigma_c) - \mathbf{f_{surround}}(x, y, \sigma_{sx}, \sigma_{sy})$$

$$\mathbf{f}(x, y, \sigma_X, \sigma_Y) = \frac{1}{2\pi\sigma_X\sigma_Y\sqrt{1-\rho^2}} \exp\left(-\left(\frac{1}{2(1-\rho^2)}\right)\left[\frac{(\mathbf{AP}-x)^2}{\sigma_X^2} + \frac{(\mathbf{ML}-y)^2}{\sigma_Y^2} - \frac{2\rho(\mathbf{AP}-x)(\mathbf{ML}-y)}{\sigma_X\sigma_Y}\right]\right) \quad (7)$$

$$\rho = \frac{\mathbf{X} \cdot \mathbf{Y}}{||\mathbf{X}||||\mathbf{Y}||}$$

The model performs the sorting convolution (**SC**) between the afferent sorting filter and the cortical subplate in multiple developmental steps (s), each step centering the filter on a different afferent position. For each selected afferent (i) at a given cortical position [x, y], the model calculates the convolution when the afferent is at its original position (xp = x, yp = y) and when it moves to one of eight adjacent positions (xp = x + p, yp = x + p) within a radius of 1 pixel (p = [−1 0 1]),

as shown in Eq. 8.

$$\mathbf{SC}_i(\mathrm{xp}, \mathrm{yp}, s) = \mathbf{ASF}(x, y, \sigma_c, \sigma_{sx}, \sigma_{sy})\mathbf{CP}_i(\mathrm{xp}, \mathrm{yp}, s) \begin{cases} \mathrm{xp} = x + p \\ \mathrm{yp} = y + p \\ p = [-1\ 0\ 1] \\ 1 \le s \le 10 \end{cases} \quad (8)$$

After performing all the convolutions, the afferent moves to the position that generates the maximum convolution absolute value (xpMax, ypMax), as shown in Eq. 9. In turn, the adjacent afferent that was at position (xpMax, ypMax) moves to the original position of the selected afferent $[x, y]$.

$$\mathbf{CP}_i(x, y, s+1) = \mathbf{CP}_i(\mathrm{xpMax}, \mathrm{ypMax}, s) \{1 < s < \mathrm{sl}\} \quad (9)$$

For example, if the cortical position with the largest convolution value is one pixel to the right of the original position, the selected afferent moves to position $[x + 1]$ and the afferent that was at position $[x + 1]$ moves to position $[x]$. The full set of sorting convolutions is repeated for ten sequential developmental steps ($s$) and the segregation remains nearly identical in subsequent steps[29]. At the last developmental step (sl = 10), afferents with the same retinotopy and eye input are neighbors within the cortical subplate.

**Model stage 2. Development of cortical subplate (thalamic afferent sorting by ON-OFF polarity).** After the model finishes sorting the afferents by retinotopy and eye input, it starts sorting them by ON-OFF polarity using a similar algorithm as for eye input. The algorithm assigns values of 1 and −1 to afferents of ON and OFF contrast polarity. Then, it calculates the convolution between the cortical subplate (**CP**) and the afferent sorting filter (**ASF**). The afferent sorting filter for ONOFF polarity is identical in shape and size to the afferent sorting filter for ocular dominance. However, in species with different sampling density along azimuth and elevation axes (e.g. macaques), the filters are elongated and the angles for ocular dominance and ONOFF sorting are orthogonal. As for ocular dominance segregation, the ONOFF sorting sequentially centers the filter at each afferent (Eq. 8) and calculates convolutions at the afferent original position $[\mathrm{xp} = x, \mathrm{yp} = y]$ and adjacent positions $[\mathrm{xp} = x + p, \mathrm{yp} = y + p]$. The afferent moves to the position with the largest convolution value (xpMax, ypMax, Eq. 9) that receives input from the same eye.

At the last developmental step (sl = 10), all afferents from the cortical subplate are organized in maps for retinotopy, eye dominance and ON-OFF polarity. In these maps, afferents of the same type (i.e. same retinotopy, eye input and ON-OFF polarity) are close together in the cortical subplate forming afferent clusters that are very similar to those measured in the visual cortex of minks, ferrets, cats and mice[15,38,62,64]. The model assumes that the similarity in stimulus preferences of neighboring afferents help them drive common cortical targets more effectively through synchronous activation[19]. In turn, their synchronous activation reinforces their synaptic weights at neighboring cortical regions, making their cortical targets also similar in retinotopy, eye dominance and ON-OFF polarity.

**Model stage 3. Development of visual cortex (spread of thalamic axon arbors).** After finishing the afferent sorting, the model makes each afferent grow into an immature cortex and start spreading its axonal arbor. In real brains, the size of the axonal arbors varies across cell types (e.g. larger for Y than X thalamic afferents in cats) and species (e.g. larger for X afferents in cats than Parvocellular afferents in macaques). We assume that the most important factor in shaping cortical topography is the average thalamic arbor size at each cortical location, which is roughly constant throughout the cortex[5]. Therefore, in the model, the size of the thalamic axonal arbor represents the average LGN arbor size in each species.

The thalamic axonal arbor (TAA) is modeled as a two-dimensional Gaussian function (Eq. 10) with a standard deviation ($\sigma$) set to 3–4 cortical pixels in all simulations of cat visual cortex reported in the paper (50 microns per cortical pixel, $\sigma$: 150–200 microns). The axon arbor spreads from its central cortical position [xc, yc] to other cortical pixels within the anteroposterior (**AP**) and mediolateral (**ML**) cortical axes. The synaptic weight at a given cortical location ($w_i$) is determined by the distance of the afferent synapse from the axonal arbor center, the afferent type (e.g. dominant or non-dominant within an afferent cluster), and the afferent receptive field position. The synaptic weight is maximum at the axonal arbor center and decays to zero at the arbor edges (radius: 10–15 cortical pixels, 500–750 microns). Among afferents at the same cortical location, the synaptic weight is maximum for afferents with the dominant receptive-field polarity (e.g. OFF afferent in OFF afferent cluster) and minimum for those with non-dominant polarity (e.g. ON afferent in OFF afferent cluster).

$$\mathbf{TAA}_i(x, y, \sigma) = w_i(x, y)\frac{1}{2\pi\sigma^2}\exp\left(-\frac{(\mathbf{AP} - \mathrm{xc})^2 + (\mathbf{ML} - \mathrm{yc})^2}{2\sigma^2}\right) \quad (10)$$

As a consequence of the axon arbor spread, each cortical location receives inputs from multiple afferents that compete for cortical space. This synaptic competition is required to replicate the range of spatial-frequency preferences experimentally measured at nearby locations in cat visual cortex. To simulate the synaptic competition, the model calculates the dominant thalamic receptive field of each cortical pixel, separately for each eye. It selects the central afferent of the cortical pixel ($c$; the afferent reaching its cortical pixel first, before arbor spread).

Then, it calculates the weighted receptive field average of all neighboring afferents converging at that pixel that have the same polarity as the central afferent (**DTRF**$k$). The receptive field from each neighboring afferent is multiplied by the weight of the axon arbor from the central afferent ($\mathbf{w}_c$) at the main cortical pixel $[x, y]$ of the neighboring afferent. The weighted receptive fields are then summed and the sum normalized by its maximum value to obtain a matrix of dominant weights associated with the central afferent (**DW**$_c$). The weights are then applied to all converging neighboring afferents with non-dominant polarity if their retinotopic positions overlap a circular area centered on the dominant thalamic receptive field. The diameter of this circular area equals the average cortical receptive field at that cortical location, which is ~1 degree in all simulations of cat visual cortex. In cortical regions with large receptive fields, the dominant receptive field is sharpened with a power of 2 to allow afferents with non-dominant polarity to be more competitive. When simulating cortical maps with ON and OFF afferent clusters and ON-OFF competition, the final weight of each afferent ($\mathbf{wf}_i$) is calculated as 1 minus the weight extracted from the weight matrix (Eq. 11). When simulating cortical maps without ON and OFF afferent clusters and only retinotopic competition, the final weight is extracted directly from the weight matrix.

$$\mathbf{DW}_c(\mathrm{rx}, \mathrm{ry}) = \frac{\sum_{k=1}^{n}\mathbf{w}_c(x, y) \times \mathbf{DTRF}_k(x, y, \mathrm{rx}, \mathrm{ry})}{\max(\sum_{k=1}^{n}\mathbf{w}_c(x, y) \times \mathbf{DTRF}_k(x, y, \mathrm{rx}, \mathrm{ry}))}$$
$$\mathbf{wf}_i(x, y, \mathrm{rx}, \mathrm{ry}) = 1 - \mathbf{DW}_c(\mathrm{rx}, \mathrm{ry}) \quad (11)$$

Through this synaptic competition, afferents with non-dominant polarity have their weights decreased to zero if their receptive fields completely overlap the dominant receptive field and increased as the receptive-field overlap decreases. The synaptic competition between ON and OFF afferents has a correlate in real brains. Neighboring afferents of the same contrast polarity have a competitive advantage over afferents of different polarity because they are more frequently co-activated by retinal waves or visual experience[19,72]. In contrast, neighboring afferents of different contrast polarity are co-activated less frequently and only when they have non-overlapping receptive fields that can be simultaneously stimulated (e.g. vertical grating for ON and OFF afferents with horizontally displaced receptive fields). The model assumes that the afferent groups most frequently co-activated are also most effective at driving their common cortical targets and strengthening their connections.

After the initial afferent weights are adjusted, the model simulates the population cortical receptive field (**CRF**) at each cortical location $[x, y]$ as the weighted sum of the receptive fields from all thalamic afferents at that location (Eq. 12).

$$\mathbf{CRF}(x, y) = \sum_{k=1}^{n} \mathbf{w}_k \times \mathbf{TRF}_k(x, y) \quad (12)$$

Then, the model measures the cortical response to different stimulus orientations (**CR$\theta$**) by performing a two-dimensional Fast Fourier transform (FFT) of the cortical receptive field. The FFT space is divided into 16 equal orientation sectors (11.25 degrees per sector) and the cortical response to each orientation ($\theta$) is calculated as the sum across all spatial frequencies ($sf$) within the preferred orientation sector (Eq. 13). The model calculates the preferred orientation of the cortical receptive field (**CRF**p$\theta$) as the orientation that generates the maximum response.

$$\mathbf{CR}\theta(x, y, \theta) = \sum_{sf=1}^{nsf} \mathbf{FFT}(\mathbf{CRF}(x, y)) \quad (13)$$
$$\mathbf{CRF}p\theta(x, y) = \max(\mathbf{CR}\theta(x, y, \theta))$$

The model uses the values of orientation preference at each cortical location to generate a primordial orientation map (POM) in the cortical subplate before birth (Eq. 14).

$$\mathbf{POM}(x, y) = \mathbf{CRF}p\theta(x, y) \quad (14)$$

In this primordial orientation map, most cortical locations receive input from both eyes but with different eye dominance. Therefore, the model generates three variations of the primordial orientation map, one for the contralateral eye, another for the ipsilateral eye, and another for the dominant eye at each cortical location. The primordial orientation maps from contralateral and ipsilateral eyes share many features in common but are not accurately matched in orientation and ON-OFF receptive field structure, consistently with experimental data[48–50].

**Model stage 3. Development of visual cortex (the mature visual map).** In the final developmental stage (i.e. after birth), the model adjusts the synaptic weights of the thalamic afferents based on visual experience. If all orientations and both eyes drive the cortex with equal strength, the model maximizes the cortical coverage of all orientations within each afferent cluster. However, if there is a bias in visual experience towards one eye and/or one orientation, the model makes the cortical map also biased.

The model simulates monocular deprivation by shrinking the axon arbors of the deprived eye. The axon arbor shrinkage is simulated by elevating the afferent weight ($w$) to a power of 5 and then normalizing the weight to the original value. As in real brains, monocular deprivation in the model has a pronounced effect on the ocular dominance map but the orientation map from the non-deprived eye remains normal. The model also simulates the effect of biased visual experience towards a dominant orientation. It does this by adjusting the synaptic weights of

the afferents at a percentage of randomly selected cortical locations (pcl) to shift their preference towards the dominant orientation (pcl = 20% to generate the subtle biases described in experimental observations[73]). For simplicity, the description below assumes that both eyes receive equal stimulation from all possible stimulus orientations during cortical development.

The model starts the maturation of the primordial orientation map by maximizing the coverage of all orientations within each afferent cluster. This maximization is important to allow each afferent type to signal the maximum number of orientations possible (e.g. allow OFF contralateral-eye afferents to signal all possible orientations of a dark stimulus projected on the contralateral retina at position [rx, ry]). To achieve this maximization, the model convolves the primordial orientation map of each afferent cluster ($POM_c$) with four orientation-coverage filters ($OCF$). The orientation-coverage filter is mathematically the same as the afferent sorting filter (Eq. 7) but is 25% smaller. The surround of the orientation-coverage filter is also elongated (ratio between longest and shortest axes = 2), and its angle can take four different values (0, 45, 90, 135 degrees). To perform the convolution, the model first transforms the patch of the primordial orientation map into complex space. Then, it convolves the real components of the patch, cos ($POM_c$), with four orientation-coverage filters ($\alpha$ = 0, 45, 90 or 135 degrees). Then, for each convolution with a real component, it performs another convolution with the imaginary map components, sin ($POM_c$), using orientation-coverage filters with all angles except the one used for the real component (e.g. $\beta$ = 45, 90 and 135 degrees for $\alpha$ = 0 degrees). The 12 convolutions generate 12 different cortical patches. The model measures the orientation coverage (cov) in each of the 12 patches and selects the one with maximum orientation coverage (maxcov), which is the patch that has its pinwheel center closest to the center of the afferent cluster (Eq. 15).

$$POM_c(x, y, \text{cov}) = \arctan \frac{\cos(POM_c(x,y))OCF(x,y,\alpha)}{\sin(POM_c(x,y))OCF(x,y,\beta)} \left\{ \begin{array}{l} \alpha = [0, 45, 90, 135] \\ \beta = [\alpha + 45, \alpha + 90, \alpha + 135] \end{array} \right\}$$
$$OM(x, y) = POM_c(x, y, \text{maxcov})$$
(15)

This maximization process allows each afferent cluster to cover a complete orientation cycle and makes the orientation map (OM) periodic. In real brains, this maximization process is likely mediated by visual experience, which co-activates afferents with multiple stimulus orientations at each position of visual space over the time period following eye opening that is known as the critical period. Visual stimuli should co-activate adjacent afferents more frequently than distant afferents because adjacent afferents have more similar stimulus preferences (i.e. similar retinotopy, eye input and ONOFF polarity). In turn, because adjacent afferents make more connections with common cortical targets, their co-activation should help drive their targets more effectively and reinforce their synaptic weights. Afferents with the same retinotopy (i.e. overlapping receptive fields) should be more strongly co-activated than afferents with different retinotopy (i.e. non-overlapping receptive fields). Therefore, through afferent co-activation, species with a large number of afferents with overlapping receptive fields should devote large cortical areas to represent the same retinotopy (i.e. large iso-retinotopic cortical domains). Conversely, species with a limited number of afferents with overlapping receptive fields can only devote limited areas of cortex to represent the same retinotopy. The model makes cortices with large cortical iso-retinotopy domains to accommodate multiple orientation cycles per visual point. Conversely, it makes cortices with small cortical iso-retinotopy domains to accommodate fewer orientation cycles per visual point (e.g. tree shrews[53]) or distribute orientation preference nearly randomly (e.g. rodents[15]). The model also makes the shape of orientation domains to be closely associated with the shape of the iso-retinotopic domains or afferent clusters. For example, large elongated iso-retinotopic domains generate large elongated orientation domains and the longest axes of both domains are orthogonal, consistently with experimental measurements in tree shrew and cat area V1[3,74].

In real brains, orientation maps and cortical receptive fields mature at the same time. To simplify and speed up the computation process, the model matures the orientation map first and then uses this map to guide the maturation of the cortical receptive fields from both eyes. At each cortical location, the model starts by adjusting the synaptic weights of the afferents with non-dominant-polarity (e.g. ON afferents in an OFF afferent cluster) until the preferred orientation of their receptive fields matches the preferred orientation of the mature cortical orientation map at that location. This synaptic adjustment is done by convolving the receptive field at each cortical location with the same receptive field rotated to match the orientation of the map and then using the values of this convolution to adjust the synaptic weights of the afferents. This process simulates the synaptic co-activation and reinforcement of afferents with a population receptive fields matching the dominant orientation at each cortical region. The model also linearly decreases the weights of the afferents with non-dominant-polarity as a function of their cortical distance from the border of the afferent cluster and pinwheel centers. The weights of non-dominant polarity afferents at the border of an afferent cluster are multiplied by one, those at pinwheel centers by zero and those in between pinwheel centers and afferent-cluster borders by 0.5. If the changes in synaptic weight of non-dominant polarity afferents are not enough to match the orientation preference of the mature cortical map, the model adjusts the synaptic weights of the afferents with dominant polarity. After the first iteration of synaptic adjustments is complete, the model generates the orientation maps for the contralateral and ipsilateral eyes by extracting the orientation preference of the cortical

receptive fields, as previously described. At this developmental stage, 95% of the cortical locations from contralateral- and ipsilateral-eye maps have the same orientation preference. In a second and last iteration of synaptic adjustments, the model minimizes the binocular differences in orientation preference, spatial frequency and dominant polarity at each cortical location of the visual maps from the two eyes.

**Measurements of stimulus topography in the mature cortical map**. The mature cortical map generated by the model contains a combined representation of multiple stimulus parameters such as spatial position, eye input, dark-light polarity, orientation preference, orientation selectivity, spatial resolution, low-pass filtering, and different receptive field structures. The model generates multiple maps for individual or combined stimulus properties, population receptive fields, afferent axon arbors, and correlations of neuronal stimulus preferences across the cortex. All maps are interpolated four times to minimize correlation noise. In the orientation maps, the interpolations are done separately for real and imaginary map components and the components are then combined. The cortical retinotopic map (**RM**) describes the central position of the population receptive field at each cortical location for each eye or both eyes (Eq. 16). The ocular dominance map (**ODM**) describes the difference between the cortical receptive fields from the contralateral (**CRFC**) and ipsilateral (**CRFI**) eyes. The contrast-polarity map (**CPM**) describes the difference between ON dominated (**CRFON**) and OFF dominated (**CRFOFF**) cortical receptive fields. The orientation map (**OM**) describes the receptive-field preferred orientation at each cortical region (**CRFpθ**). The orientation-selectivity map (**OSM**) describes the average orientation tuning of each cortical region measured as circular variance (**CRFcv**). The spatial-frequency map (**SFM**) describes the receptive-field preferred spatial-frequency (**CRFsf**) at each cortical location (i.e. maximum spatial-frequency value in the FFT measured at the preferred orientation).

$$\begin{aligned} RM(x, y) &= CRF(x, y) \\ ODM(x, y) &= CRFC(x, y) - CRFI(x, y) \\ CPM(x, y) &= CRFON(x, y) - CRFOFF(x, y) \\ OM(x, y) &= CRFp\theta(x, y) \\ OSM(x, y) &= CRFcv(x, y) \\ SFM(x, y) &= CRFsf(x, y) \end{aligned}$$
(16)

The model also generates an orientation homogeneity map (**OHM**) that describes the local orientation gradients measured with a local homogeneity index (**LHI**), as described in Eq. 17. The LHI of each cortical location[75,76] is calculated with an exponential function that includes all preferred orientations ($\theta_j$) and distances ($d_j$) of regions surrounding the selected cortical location within a circular radius ($\sigma$) of 100 microns. The function has a normalization factor ($k$) that makes the maximum LHI value equal to one (Eq. 17). LHI reaches the maximum value of one in cortical regions with complete orientation homogeneity and the minimum value of zero in cortical regions with the lowest orientation homogeneity. The model also calculates the orientation tuning associated with the cortical receptive field (**CRFcv**) as the circular variance of the cortical responses (**CRθ**) to the different stimulus orientations ($o$), as shown in Eq. 17.

$$OHM(x, y) = LHI(x, y)$$
$$LHI = k \sum e^{-\frac{d_j^2}{2\sigma^2}} e^{2i\theta_j}$$
$$CRFcv = 1 - \left| \frac{\sum_{o=1}^{16} CR\theta_o e^{i2\theta}}{\sum_{o=1}^{16} CR\theta_o} \right|$$
(17)

The model also outputs paired correlations for different combinations of stimulus parameters such as orientation, circular variance, spatial resolution, ocular dominance, ON-OFF dominance, retinotopy, and low-pass spatial frequency. The spatial resolution is defined as the highest spatial frequency that generates 50% of the maximum response (SF50). The low-pass spatial frequency index (LPI) is defined as the ratio between the response at zero spatial frequency and the maximum response (i.e. one for low-pass filters and zero for band-pass filters). The model generates cortical maps, receptive fields and stimulus response tuning both under binocular or monocular stimulation. To simulate monocular stimulation, the model multiplies the cortical responses by an ocular dominance index. The ocular dominance index is calculated as the afferent average weight from the dominant eye divided by the afferent average weight from the non-dominant eye.

**Quantification of model error in simulations of species-specific cortical maps**. We quantified the model error at generating species-specific cortical maps for macaque, cat and tree shrew by calculating the difference between simulated and experimentally-measured maps of the same species. For comparison, a control error was calculated as the difference between experimentally-measured maps from two species, choosing the two species with most similar brain sizes (macaque minus cat for map simulations of macaque and cat; cat minus tree shrew for map simulations of tree shrew). The errors for all properties except pattern similarity and gradient orthogonality were measured by randomly selecting cortical patches of 2 × 2 millimeters and then measuring pinwheel density (in pinwheels per square millimeter), same-sign and different-sign pinwheel distance (in microns), and distance between the peak or valley of orientation domains (in microns). The error measurements for pinwheel distance were obtained from 100 cortical patches and

those from orientation-domain distance with just 40 patches. The errors in pattern similarity were also measured by randomly selecting 40 cortical patches of $2 \times 2$ millimeters and then calculating one minus the correlation between the Fast Fourier Transform of the two cortical patches under comparison. The error in pattern similarity is zero if the two patches are identical and one if the correlation of the Fast Fourier Transforms is zero. The error in gradient orthogonality was measured by calculating the frequency of different angle intersections between the map gradients for orientation preference and ocular dominance (only in macaque and cat because tree shrews do not have ocular dominance bands along the cortical surface). The angles were measured between iso-orientation lines plotted for 10 different orientations and ocular dominance bands plotted as contour plots at 25, 50 and 75% of the absolute maximum value. The gradient orthogonality was measured as the ratio between the frequencies of 90 and 0 degree angles (90 angle frequency divided by 0 angle frequency) and the ratio subtracted from a value of 6 for macaque and 5 for cat. In the measurements of gradient orthogonality, the model error was obtained from cortical maps simulated with afferent sorting filters matched in size to the iso-retinotopic domain of each species (i.e. the diameter of the filter center was equal to the average width of the ocular dominance bands) whereas the control error was obtained by using afferent sorting filters 50% smaller than the iso-retinotopic domain.

**Quantification of model simulations for correlated map gradients.** We quantified the model performance at replicating correlated map gradients as follows. We simulated 1000 recording tracks and measured all possible correlations across all cortical pixels within the track between pairs of stimulus-map dimensions. The stimulus-map dimensions were orientation selectivity, spatial resolution, low-pass index, and local orientation homogeneity. The simulated recording tracks sampled the cortical map at 40 cortical pixels per track. For statistical comparison, we run the simulations with and without sorting the afferents by eye input and ON-OFF polarity. For each simulation type, we counted the tracks that passed specific criteria of correlation strength, sign and dimension range. We counted tracks that had 3–5 correlations with a strength equal or larger than 0.15, 0.2, 0.3, 0.4 or 0.5 and tracks that also had correlations with the same sign as those measured experimentally. We also counted tracks that had a value range for each correlated stimulus dimension equal or larger than 0.2 times the range experimentally measured. We compared the counts obtained in the two simulation types with a Chi squared test.

**Surgical procedures and data collection in electrophysiological recordings.** We tested several main predictions of the computational model with electrophysiological recordings from cat primary visual cortex. Adult male cats ($n = 16$, 3–4 Kg, 0.8–1.2 years old) were tranquilized with an intramuscular injection of acepromazine (0.2 mg/kg) and anesthetized with an intramuscular injection of ketamine (10 mg/kg). Continuous infusions of propofol ($5–6 \ mg \cdot kg-1 \cdot h-1$), sufentanil ($10–20 \ ng \cdot kg-1 \cdot h-1$), vecuronium bromide ($0.2 \ mg \cdot kg-1 \cdot h-1$), and saline (1–3 ml/h) were administered to maintain anesthesia and eliminate eye movements during the recordings. All vital signs were carefully monitored and maintained within normal physiological limits. Silicon probes with 32 linearly-arranged recording sites (100-micron inter-electrode distance) were inserted tangentially into the primary visual cortex to maximize the number of recordings sampling cortical layer 4. We performed combined measurements of orientation and spatial frequency tuning in 9 cats and measurements of ON/OFF receptive field organization in 16 cats (some of the data for ON/OFF receptive field analysis was collected over many years and included in previous publications). All surgical and experimental procedures were performed in accordance with the guidelines of the U.S. Department of Agriculture and were approved by the Institutional Animal Care and Use Committee at the State University of New York, State College of Optometry.

**Visual stimuli and data analysis in electrophysiological recordings.** Visual stimuli were generated with Matlab Psychtoolbox (Matlab versions: 2010 to 2014; Psychtoolbox versions: 3.0.9 to 3.0.11) and displayed on a CRT monitor in early experiments (refresh rate = 120 Hz, mean luminance = 61 cd/m2) and a ViewPixx monitor in later experiments (refresh rate = 120 Hz, mean luminance = 112 cd/m2). Cortical orientation and spatial frequency tuning were measured with a sequence of flashed sinusoidal gratings that had different orientations, phases and spatial frequencies. The gratings included all parameter combinations of 8 equally spaced orientations, 4 equally spaced phases, and 10-11 spatial frequencies. The spatial frequencies ranged from 0.03 to 1 cycles per degree in earlier experiments and also 2 cycles per degree in later experiments. Each grating combination of orientation, spatial frequency and phase was presented 20 times in random order under monocular stimulation, first through the contralateral eye and then through the ipsilateral eye. The orientation/direction tuning was also measured with light and dark bars presented on a mid-gray background sweeping at 8 different orientations and 16 directions of movement. Cortical receptive fields were mapped with sparse noise stimuli made of white squares on black backgrounds (to measure ON receptive field subregions) or black squares on white backgrounds (to measure OFF subregions). The square stimuli were presented for 33 ms, were 2.8 degrees/side in size, and sampled visual space at positions separated by 1.4 degrees.

Responses to flashed sinusoidal gratings were measured within a temporal integration window of 200 ms, starting 20 ms after the stimulus onset. All responses

were averaged across grating phases. The orientation tuning was measured from responses to different orientations at the preferred spatial frequency of the cell group sampled within a recording site of the multielectrode array. The orientation tuning curve was fit with a von Mises function and the fit used to measure the preferred orientation and orientation tuning of cell group measured with each cortical recording site. The orientation tuning was measured as the circular variance (CV) of the responses to all the orientations, which ranges from one (broadest tuning) to zero (narrowest tuning). We also measured the orientation selectivity index as 1 – (NP / PO), where PO is the response to the preferred orientation and NPO the response to the non-preferred orientation (orthogonal to the preferred one). The orientation selectivity is 0 when the responses to the preferred and non-preferred orientations are equal and 1 when the response to the non-preferred orientation is zero. The orientation preferences from multiple contiguous recording sites within the multielectrode array were used to compute the local homogeneity index (LHI)[75,76]. LHI measurements were calculated within a 300 microns linear distance from the reference recording site (i.e. the 6 closest electrodes in the linear multielectrode array).

The spatial frequency tuning was measured on a logarithmic scale for spatial frequency and fit with a difference-of-Gaussians function. The fits were then used to extract the low-pass index (LPI) and spatial resolution of the function. The LPI was defined as the ratio between the response to zero spatial frequency (y-intercept) and the maximum response. The spatial resolution was defined as the highest spatial frequency that generated half of the maximum response (SF50). The range of spatial resolution varied greatly across different penetrations and animals due to variations in the visual eccentricity of the recordings. Therefore, to make comparisons of SF50 across animals and multielectrode penetrations, we converted each SF50 to a relative SF50 (SF50r) by dividing each SF50 value by the average SF50 across the simultaneously recorded neurons. Only recordings with a reasonable signal to noise ratio (snr ≥ 3.5) were included in the analysis of orientation and spatial frequency tuning. The snr was computed as the ratio between the maximum response and the baseline firing rate. The baseline firing rate was defined as the average firing rate integrated within the first 20 msec and last 40 msec of the peri-stimulus time histogram.

ON and OFF receptive fields were calculated in time windows of 10 ms following the stimulus onset and normalized by their maximum response. They were subtracted from each other at each time window to generate a sequence of ON-OFF receptive field subtractions spanning 24 temporal frames, from 0–10 ms to 230–240 ms. To reduce receptive-field noise, we summed the absolute value of the receptive field across time, normalized the sum by the maximum response, and set any response below 20% of the maximum to zero. To obtain accurate correlation values between ON-OFF receptive field dominance and orientation tuning, we selected the receptive fields and orientation measures with highest signal to noise (RFsnr > 12, ORsnr > 6, ORresp > 10 spikes). RFsnr was calculated as the ratio between the maximum value within the receptive field and the average of all values below 80% of the maximum. ORsnr was calculated as the ratio between the maximum number of spikes within a 50 msec bin and the average across all bins (16 directions x 35 bins, 1.75 sec per bar sweep). ORresp was calculated as the maximum number of spikes at the preferred orientation within a 50 msec bin. We also selected receptive fields with pronounced ON-OFF segregation that matched the preferred orientation measured with moving bars (ORdiff < 30 degrees). ORdiff was calculated as the difference between the orientation preference measured with moving bars and the predicted from the ON-OFF structure of the receptive field (calculated with a fast Fourier transform, FFT). The ON-OFF response balance was measured as 1- abs (ON-OFF) / (ON + OFF), where ON and OFF are respectively the maximum ON and OFF values within the receptive field. When ON and OFF responses are identical in strength, the ON-OFF balance reaches its maximum value of one. When the receptive field is completely dominated by ON or OFF, the ON-OFF balance is zero. We used four different approaches to calculate the ON-OFF response balance: mean, maximum, mean average, and maximum average. The mean ON-OFF balance is the mean of all values of ON-OFF balance measured at all the time windows of the receptive field (illustrated in Fig. 3e). The maximum ON-OFF balance is the maximum value across all time windows. The mean-average ON-OFF balance is the ON-OFF balance calculated after averaging the receptive fields from all time windows and then taking the ON mean and OFF mean from the receptive field average instead of the ON and OFF maximum values. The maximum-average ON-OFF balance is the ON-OFF balance calculated by averaging the receptive fields from all different time windows and then taking the ON and OFF maximum values from the receptive field average. All four approaches revealed a significant positive correlation between ON-OFF response balance and orientation tuning (r = 0.3596 for mean ON-OFF balance, r = 0.2886 for maximum ON-OFF balance, r = 0.3825 for maximum-average ON-OFF balance, and r = 0.3226 for mean-average ON-OFF balance, $p < 0.00001$ for all).

**Reporting summary.** Further information on research design is available in the Nature Research Reporting Summary linked to this article.

## Data availability

All the electrophysiological measurements and computer simulations from this study are available from source data provided with this paper, from a repository in Zenodo[40], and upon request from the correspondence author (jalonso@sunyopt.edu).

## Code availability

Code to run customized simulations and generate the figures and tables reported in this study are available from a repository in Zenodo[40], and upon request from the correspondence author (jalonso@sunyopt.edu).

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

## Acknowledgements
This work was supported by RO1 EY005253 and RO1 EY027361 from NIH/NEI (J.M.A.) and by DFG Emmy-Noether grant KR 4062/4–1 (J.K.).

## Author contributions
S.N., E.K. and J.M.A. developed the computational model. J.J., E.K., S.N., H.R., J.K., and J.M.A. performed the electrophysiological measurements. S.N., E.K., K.L.T., J.J., H.R., Q.Z., J.K., and J.M.A. performed analysis of electrophysiological measurements and/or model simulations. The paper was written by S.N. and J.M.A. and edited by all the authors.

## Competing interests
The authors declare no competing interests.
