## [Peer review file · Nature Communications]

REVIEWER COMMENTS

Reviewer #1 (Remarks to the Author):

In this manuscript, the authors propose a theoretical model in which a possible developmental mechanism of various cortical maps is suggested. The authors developed a computational model, based on the idea of “retinal origin of cortical architectures” previously suggested, but also advanced the idea by adding a new component in the model — Sorting of thalamic afferent by type. They also provided additional comparison analyses between simulated results and observations from animal experiments, to show that various cortical map features can be reproduced by the suggested model.

Overall, the manuscript provides an intriguing perspective regarding the origin of cortical functional circuits organized into systematic patterns that have inspired neuroscientists for a long time. While the main idea of the model is interesting and the overall direction of the study is sound, a large portion of analyses and arguments need to be presented with much greater clarity, to make the main result more convincing. Particularly, the current form of the manuscript seems to be prepared for a brief letter format, and I believe that the manuscript can be much strengthened if the authors reorganize the materials so that they can provide a more detailed description of the model itself including its rationale, novelty of the idea compared to other recent approaches, and new core predictions that can explicitly test the model, instead of just reproducing some tendencies observed in experimental observations.

Once these issues are properly addressed, I think the combination of theory and analysis provided in this study will make the article of broad interest to a wide audience in neuroscience. Here my suggestions and concerns are summarized as several major issues and a few minor questions.

Major issues

1. Title, summary, and introduction: What do you mean by the term “theory”? You need to clarify this in the abstract and maybe in the early part of the main text. Currently, it is not clear what is the main idea of your theory, why it is important, what is your rationale behind the idea, and what kind of new insight can be provided by it. Please provide more precise and specific descriptions of these issues in the abstract and introduction.

2. Model: The core material of this study is the theoretical model itself. However, the current form of the manuscript does not spend enough effort to precisely describe it. I think you can make a revised figure 1 (merging with some parts of the Extended Data Figures) to provide information on your model design initially. Considering the character of this study, “See methods” is not a good way of introducing your model.

3. Model details:

A. “The sorting of thalamic inputs by type”: Maybe this is one of the most important components of the model, thus it needs to be more carefully explained. What is the definition of this in the model (i.e. How you implement biological observations in the model)? What is the main reason that you designed this

component and why this is important? As mentioned in the discussion “We propose a theory of cortical map formation that relies on afferent sorting”, this seems to play a crucial role in the model, but it is not fully described.

B. There are several different stages of the sorting (by retinotopy, by eye input, by ON-OFF polarity) and they may play different roles, respectively. Currently, it is only described as “Maps cannot form without each sorting” (Page 5 Line 1), without further analysis or reasoning. Obviously, this is the most important part of the model and you must show what is the mechanism that enables this sorting to organize maps. You also need to add analyses with controls in which sorting in each stage, or some different combinations of them, are removed or weakened — What can you find under these conditions? Does it validate the idea that sorting is required to form cortical maps?

C. “Consistently with experimental evidence” (Page 5 Line 3): Is there any direct evidence that retinotopic and other types of sorting cause to form cortical maps? (You have to distinguish correlation and causality).

D. In several places, there are sentences “The model predicts ...”, but it is not clear why and how that result is predicted (If you simply run the model and get this result, this is not a prediction). For each prediction, please provide a brief description of reasoning.

E. Variations across species: There are available RGC data from rodent species. Why don't you perform simulations using these animal data? Can your model simulation produce a salt-and-pepper type (but not completely random, Ringach et al. 2016) cortical organization in rodents? For this issue, what is different from other theoretical approaches? (e.g. Meng et al. 2012; Weigand et al. 2017; Jang, Song & Paik 2020). You need to introduce several other ideas and compare.

F. Page 7 Line 2: “The model predicts that an increase in ON-OFF response balance should be associated with an increase in ...”. Is this an “empirical” prediction from your computational simulations, or the one from mathematical toy model analysis?

G. “... the model to match the statistics of the retinal mosaic ...” (Methods, Page 3; Extended Data Figure 1): This only matches the very first order statistics of nearest neighbors. How can you match higher-order statistics of RGC mosaics observed experimentally? (e.g. Eglén et al. 2008)

H. “... (CP) is convolved with an afferent sorting filter (ASF) that has a center-surround organization (Figure S1c)”: I am not sure if this is an acceptable and biologically plausible design. First, this type of band-pass filter can always generate a certain frequency of periodic pattern (Royer & Schwartz 1990), even from random noise, and the result is completely determined by filter parameters, not by signals filtered. Second, this model predicts a Mexican-hat type correlation in local afferents (Miller 1994), but this is not observed in experiments (Ohshiro & Weliky 2006). What is your rationale for this model assumption?

Minor issues

“... ON-OFF geometry, and orientation preference, closely replicating experimental measures” : This sounds not precise. Describe which measures should be replicated, and what statistical test you can show to validate them.

- Page 6

In tree shrews, orientation map is not as uniform as that of primates but organizes into strips and blobs (Kaschube et al. 2010). Thus, pinwheel density and distance significantly vary across the area. What is a determinant of these local patterns in your model, and how can you make a precise estimation of the pinwheel statistics under this condition?

- Page 9

“All these relations measured experimentally could be closely replicated in the model simulations” : How do you validate your model here? i.e. How can you prove that model results are “equivalent” to that of data, instead of just saying that both results show comparable trends or similar statistics?

Reviewer #2 (Remarks to the Author):

The authors introduce a new model of cortical development that accounts for several experimental relationships. The model consists of several stages: the simulation of retinal ganglion cell mosaics, the sorting of afferents at the cortical plate in a precise sequence (retinotopy, eye of origin and ON/OFF clusters), the innervation and ramification of afferents into the cortex, and a vision-dependent learning refinement of receptive fields and maps.

In general, understanding the details of the model was difficult. Although one can get the gist of the different elements from the introduction, all the details are relegated to a Methods section that is not easy to follow and is hampered by poor mathematical notation.

It was not clear which elements of the model are critical to establishing the observed relationships. This is critical because the authors claim the sorting of the afferents is a key contributor to this outcome, but this is not really demonstrated by the analyses offered.

While the model shows great promise, it was felt more detailed analyses are needed to support the claims. An increase in the clarity in the presentation and description of the model would also strengthen this study substantially.

My comments, questions and suggestions for improvement follow.

Major comments:

A key ingredient and the novel aspect of the model is the sorting of afferent inputs. What happens in the model if one forgoes such sorting? Can a subsequent refinement due to learning still generate similar maps, receptive fields, and the associated relationships? Without such verification, it is not clear what sorting is exactly contributing, as it may still be possible to explain the observed relationships without invoking the need for afferent sorting.

In fact, one may suspect sorting can mess things up, as it is necessarily accompanied by a deformation of the retinotopic map. The cartoon on the right shows receptive fields along a line in the retina (just one eye) and their sorting as they go into the cortex (just ON/OFF domains). In such re-organization the receptive fields of the afferents do not change at all. The sorting brings together ON RFs with disparate centers to an ON domain; the same is true for an OFF domain. The local magnification (mm/deg) of ON inputs is *decreased* at the centers of ON domains (and the same happens with OFF domains). Thus, it becomes more likely for cells to sample from ON inputs with rather different locations on the visual field. While this may potentially be corrected by means of synaptic learning, it is not uncommon to see the model report such structures as shown in one of the simulation runs below, which are rarely observed in data.

Of course, the cartoon represents an extreme version of sorting. It would be helpful to visualize the initial and final locations of the afferents after sorting in the model, along with the size of the arbors used. If the axonal arbor is large compared to the movement of the afferents (see cartoon above), then a cortical cell could still “see” the same inputs and synaptic learning could presumably yield similar receptive fields without the sorting. If the arbors are smaller or about the same size of the average afferent displacement (cartoon above) then a cortical cell will necessarily see an increase in local retinotopic scatter at the input and be susceptible to generate the kind of structures displayed above.

The contribution of synaptic learning is also not immediately obvious. Are some of the same relationships observed (Fig 4) present in the “primordial” map and receptive fields from the input? Is synaptic learning enhancing such relationships or completely establishing them? If the observed relationships are generated by synaptic learning, then what is the value of sorting? If the relationships are observed in the “primordial” map, are they also there to some degree without invoking sorting?

Finally, it would also be helpful to visualize the location of the afferents a mature cortical receptive field is pooling from and, importantly, determine how many of those afferents make up a cortical, simple receptive field. Another important number is the density of afferents per cortical hypercolumn, as well as the ratio between the width of a simple RF and the center size of the afferent inputs (see Alonso & Reid) – are those comparable? The authors could estimate these directly from their simulations.

It was difficult to follow the Methods section and understand exactly what the model entails. The math notation needs to be cleaned up and the model presented in a succinct way. Having GUI to was extremely helpful to get an intuition about the model, although the user has no access to the model parameters. I would encourage the authors to make available those parameters as well.

Minor comments

In lack of a unifying model of cortical map topography, it has been frequently argued that different species use different mechanisms to generate cortical maps, receptive fields and stimulus selectivity. Against this belief, here we introduce a general theory of cortical map formation that replicates the cortical topography and receptive field properties of different species, and their combined topography for multiple stimulus dimensions.

There have been numerous prior attempts to come up with a universal mechanism. Work by Fred Wolf and colleagues on self-organizing maps, Swindale, Goodhill, Miller, Ringach & Paik, Sur and colleagues, and wiring minimization such as those from Chklovskii and colleagues.

Finally, the co-activation of the convergent afferents creates a primordial map for stimulus orientation that is later refined by visual experience (Figure 1e-f).

I could not find a mathematical description of what this co-activation entails. The Methods includes a section that begins with --

The model assumes that afferent co-activation plays a major role in the synaptic adjustment transforming the primordial orientation map into a mature visual map.

And it goes on to provide a verbal description of what is expected. Is the model implementing this? How? Also, it is stated that “*because each afferent can only maintain a maximum synaptic weight throughout the cortex, strengthening the synaptic weight at the center of the axon arbor should weaken the weight at the arbor edges.*” I know of no evidence this is the case. Such a mechanism would mean synaptic learning is not local anymore. Can the authors provide a citation for this statement?

Consistently with experimental evidence, visual retinotopic maps do not form if the thalamic afferents are not sorted by retinotopy, ocular dominance maps do not form if the afferents are not sorted by eye input, and ON-OFF maps do not form if the afferents are not sorted by ON-OFF polarity.

Sounds a bit like circular reasoning here. Can you clarify what you mean?

Moreover, maps for other stimulus dimensions such as orientation do not form either if the afferents are not sorted by ON-OFF polarity because the size of the iso-orientation domains is determined by the size of the ON and OFF afferent clusters.

There are plenty of models where afferents are not sorted, yet maps are generated just fine. Is this a statement about experiments? If so, provide a citation.

SF50 – Please define somewhere. I could not find the definition.

This afferent sorting plays a major role in cortical map development and minimizes the wiring of thalamocortical circuits that link neurons with similar response preferences.

Not immediately obvious how afferent sorting leads to wiring minimization of the cortex.

The theory of cortical map formation that we propose predicts that the number of afferents sampling each retinal point should determine the number of stimulus dimensions segregated in the cortical map.

How exactly? What is a stimulus dimension? What is “processing power”?

For example, if three neighboring ON retinal ganglion cells make connection with one ON LGN neuron, the LGN receptive field is calculated as the weighted sum of the three inputs and the strength of each input is determined by the overlap between its receptive field and the three-input receptive-field sum.

To compute the LGN RF you need to know the weights. Then you state that you compute the weights by the overlap between the RF and the RGC receptive field. But you don't have the RF. This is totally circular. And then, the next sentence you state you are going to have a one-to-one connection between RGC and LGN neurons anyway.

This sequence of afferent sorting ... is consistent with experimental measurements in the visual pathway [8,9].

In several places, the authors remark that sorting of afferents at the cortical plate, along with the sequence they propose, is an established experimental fact. It would help to discuss the experimental evidence for each of the sorting steps and provide clear supporting evidence for each. The sentence above, for example, one of the studies refers to the segregation in the LGN, not the cortex – why is it relevant?

Recall that during development many cells in the ferret LGN have receptive fields that are similar to those of simple-cells in V1 (Tavazoie & Reid, 2000). These appear to result for the convergence of multiple ON/OFF RGC RFs onto single geniculate targets. Why couldn't a similar wiring arise in the cortex without invoking afferent sorting?

Methods Eqn (6). Is TRF defined anywhere?

“algorithm that we recently developed....”

This is really a modified version of a Kohonen self-organizing map.

In the GUI, are the squares representing the RGC distributions and the cortical maps covering the same part of the visual field? That is, is the RGC distribution projecting to the entire cortical area we see simulated? If not, can arrange these in such a way that they match?

How much is the retinotopy distorted after you sort eye and on/off? If you make a grid on the cortex and ask where the centers of the RFs at those locations are in the visual field, can you please plot the resulting grid? Is the amount of deformation consistent with what we see experimentally (eg, Campbell & Blasdel in primate or Fitzpatrick et al in ferrets or tree shrews)?

I could not find the reference to the video files referenced in the text.

Where is the periodicity of the orientation map coming from in the model?

Retinas are simulated on a square grid + uniform (x,y) noise. Most other modeling studies have used hexagonal arrays and isotropic noise. It is not immediately obvious this has no effect on the results.

It would be helpful if you could label IPSI/CONTRA eye and ON/OFF dominance in the leftmost two panels in the GUI (under primordial maps tab).

The relationship between orientation tuning and the local balance of ON/OFF inputs has been emerges in a prior model (see Ringach 2007, PLoS One, Fig 5c), and the same for spatial frequency (Fig 6 and see Jang & Paik 2017). These types of models demonstrate that many of the observed relationships can, in principle, arise without the need to invoke afferent sorting.

Reviewer #1 (Remarks to the Author):

In this manuscript, the authors propose a theoretical model in which a possible developmental mechanism of various cortical maps is suggested. The authors developed a computational model, based on the idea of “retinal origin of cortical architectures” previously suggested, but also advanced the idea by adding a new component in the model — Sorting of thalamic afferent by type. They also provided additional comparison analyses between simulated results and observations from animal experiments, to show that various cortical map features can be reproduced by the suggested model.

Overall, the manuscript provides an intriguing perspective regarding the origin of cortical functional circuits organized into systematic patterns that have inspired neuroscientists for a long time. While the main idea of the model is interesting and the overall direction of the study is sound, a large portion of analyses and arguments need to be presented with much greater clarity, to make the main result more convincing. Particularly, the current form of the manuscript seems to be prepared for a brief letter format, and I believe that the manuscript can be much strengthened if the authors reorganize the materials so that they can provide a more detailed description of the model itself including its rationale, novelty of the idea compared to other recent approaches, and new core predictions that can explicitly test the model, instead of just reproducing some tendencies observed in experimental observations.

Once these issues are properly addressed, I think the combination of theory and analysis provided in this study will make the article of broad interest to a wide audience in neuroscience. Here my suggestions and concerns are summarized as several major issues and a few minor questions.

Thank you for giving us the opportunity to explain the model in greater detail and perform additional quantitative comparisons between experimental data and model simulations. The reviewer is correct. The paper was originally formatted as a brief letter and transferred to Nature Communications without changing the format. We now use the additional space that Nature Communications provides to explain and illustrate the model in greater detail.

The paper now starts by briefly describing the theory and the main stages of the model. It then describes each stage in more detail together with the associated main theory predictions. We also emphasize more effectively the main parameters driving the differences in cortical topography across species and the differences with previous models. Unlike previous models, the most important parameter in our model is the afferent sampling density. We now make this clear from the start of the paper. We also provide additional simulations and controls to simulate cortical map differences across species and additional metrics to statistically compare experimental measurements with model stimulations.

Major issues

1. Title, summary, and introduction: What do you mean by the term “theory”? You need to clarify this in the abstract and maybe in the early part of the main text. Currently, it is not clear what is the main idea of your theory, why it is important, what is your rationale behind the idea, and what kind of new insight can be provided by it. Please provide more precise and specific descriptions of these issues in the abstract and introduction.

We have extensively modified all sections of the paper including the abstract and introduction to describe in more detail the theory, its main predictions and general significance. We hope that

the expansion of the manuscript length, number of figures, and GUI to perform customized simulations will address this criticism.

2. Model: The core material of this study is the theoretical model itself. However, the current form of the manuscript does not spend enough effort to precisely describe it. I think you can make a revised figure 1 (merging with some parts of the Extended Data Figures) to provide information on your model design initially. Considering the character of this study, “See methods” is not a good way of introducing your model.

We are very grateful for this criticism that helped us explain the model and theory more effectively. We have followed the recommendation from the reviewer and the first section of the paper is now used to describe and illustrate the main stages of the model. We also have a new figure 1 that explains in simple terms the main ideas behind the theory and its main predictions.

3. Model details:

A. “The sorting of thalamic inputs by type”: Maybe this is one of the most important components of the model, thus it needs to be more carefully explained. What is the definition of this in the model (i.e. How you implement biological observations in the model)? What is the main reason that you designed this component and why this is important? As mentioned in the discussion “We propose a theory of cortical map formation that relies on afferent sorting”, this seems to play a crucial role in the model, but it is not fully described.

Our apologies for the confusion, which was caused in great part by our attempt to condense the work in a short paper format. We have been working in this theory and model for more than five years and have been making continuous changes to make the model replicate an increasingly larger number of experimental measures. We reached the point at which the model was replicating all the experimental measurements that was designed for and we got really excited. However, we misjudged the effort that was needed to effectively communicate how the model works. We hope that this new version of the manuscript is more effective at explaining the most important components of the model, which are afferent sampling density, afferent sorting and synaptic competition. We have also made an effort at explaining in greater detail why these three components are important to simulate realistic models and how they can be interpreted in terms of biological developmental processes.

We believe that the model that we describe is the only currently available model that can accurately replicate the cortical inter-related topography from multiple stimulus dimensions that include retinotopy, ocular dominance, ON-OFF polarity, orientation, spatial resolution, low-pass filtering, local orientation homogeneity, the correlations among multiple stimulus dimensions that we measured for first time, and the diversity of receptive field structure and orientation tuning. It is also the only model that can closely replicate the orientation maps of different species using as input the gradient from a single stimulus dimension such as ocular dominance and retinotopy (see Figure 7). In this new version of the paper, we also explain in more detail the biological evidence supporting the main developmental stages of the model.

B. There are several different stages of the sorting (by retinotopy, by eye input, by ON-OFF polarity) and they may play different roles, respectively. Currently, it is only described as “Maps cannot form without each sorting” (Page 5 Line 1), without further analysis or reasoning. Obviously, this is the most important part of the model and you must show what is the mechanism that enables this sorting to organize maps. You also need to add analyses with controls in which

sorting in each stage, or some different combinations of them, are removed or weakened — What can you find under these conditions? Does it validate the idea that sorting is required to form cortical maps?

We now provide more simulations to illustrate more effectively the role of each stage of the model in cortical map formation. We also explain in greater detail how the model simulations relate with available experimental data from different species. We also provide new statistical analyses to evaluate the contribution of afferent sorting to the model performance.

C. “Consistently with experimental evidence” (Page 5 Line 3): Is there any direct evidence that retinotopic and other types of sorting cause to form cortical maps? (You have to distinguish correlation and causality).

We now describe in greater detail the experimental evidence supporting each stage of the model. We also make a greater effort at using simple and consistent terminology that helps avoid confusion. It is well accepted that retinotopic and ocular dominance maps originate from the sorting of thalamic afferents by retinotopy and eye input respectively. The sorting mechanisms (synaptic pruning, molecular cues), not the afferent sorting, remain a matter of debate.

There is no experimental evidence that retinotopic maps cause the development of maps for ocular dominance and orientation. This is one of the main novel ideas behind our theory. However, in support of the theory, the organization of retinotopic maps is known to be closely associated with the organization of ocular dominance maps¹ and orientation maps²⁻⁵. In this new version of the manuscript, we make these points clearer.

D. In several places, there are sentences “The model predicts ...”, but it is not clear why and how that result is predicted (If you simply run the model and get this result, this is not a prediction). For each prediction, please provide a brief description of reasoning.

We now introduce early in the paper the theory and its main predictions (new figure 1, also described in ⁶). We then introduce the computational model to test the theory with specific simulations that can be quantitatively compared with experimental measures. The theory predicts that the dense sampling of visual space by a large number of afferents with overlapping receptive fields should make the afferents to be sorted in cortex by eye input and ON-OFF polarity. Moreover, in animals with extensive binocular vision, the theory predicts that the afferents should become arranged in an eye-polarity grid that cause inter-related gradients for multiple stimulus dimensions including retinotopy, ocular dominance, ONOFF dominance, orientation preference/selectivity, spatial resolution, spatial frequency selectivity, and receptive field structure. The main manuscript now describes these points in much greater detail.

E. Variations across species: There are available RGC data from rodent species. Why don't you perform simulations using these animal data? Can your model simulation produce a salt-and-pepper type (but not completely random, Ringach et al. 2016) cortical organization in rodents? For this issue, what is different from other theoretical approaches? (e.g. Meng et al. 2012; Weigand et al. 2017; Jang, Song & Paik 2020). You need to introduce several other ideas and compare.

We are very grateful for this comment. We now make a greater effort at distinguishing our model from other 'ONOFF retinal models'. Unlike other models, ours does not require any specific ON-OFF geometry to generate orientation maps. Orientation maps can emerge with any ONOFF

retinal geometry as long as the afferent sampling density is high. The ONOFF retina geometry is important to reproduce individual variations in cortical topography but not to generate cortical orientation maps. For example, our model can generate a cortical orientation map with a rodent retina as long as we use the thalamic afferent density of the cat. It can also generate salt-and-pepper orientation maps that are not completely random^{7,8} if there is a subtle inhomogeneity in the ON and OFF sampling by the retina⁸ or if the number of ON and OFF afferents with overlapping receptive fields is large enough to be sorted in small cortical clusters.

F. Page 7 Line 2: “The model predicts that an increase in ON-OFF response balance should be associated with an increase in ...”. Is this an “empirical” prediction from your computational simulations, or the one from mathematical toy model analysis?

Our apologies for the confusion. It is a prediction based on the changes in ON-OFF response balance resulting from the afferent sorting. We proposed this before² but failed to explain it here. We hope that the new figure 1 and the more detailed description at the start of the paper makes this point clearer.

G. “... the model to match the statistics of the retinal mosaic ...” (Methods, Page 3; Extended Data Figure 1): This only matches the very first order statistics of nearest neighbors. How can you match higher-order statistics of RGC mosaics observed experimentally? (e.g. Eglen et al. 2008)

Great point. Thank you. We now make it clear that the model can simulate and/or use any measured retinal array as input (see new supplementary figure 1). However, as discussed above, the statistics of the retinal mosaics do not determine the emergence of orientation maps in our theory; they are only needed to reproduce individual cortical maps. In our theory, the most important parameter to reproduce differences in orientation maps across species is the density of thalamic afferents representing the same visual point.

H. “... (CP) is convolved with an afferent sorting filter (ASF) that has a center-surround organization (Figure S1c)”: I am not sure if this is an acceptable and biologically plausible design. First, this type of band-pass filter can always generate a certain frequency of periodic pattern (Rojer & Schwartz 1990), even from random noise, and the result is completely determined by filter parameters, not by signals filtered. Second, this model predicts a Mexican-hat type correlation in local afferents (Miller 1994), but this is not observed in experiments (Ohshiro & Weliky 2006). What is your rationale for this model assumption?

Our apologies again for the confusion. The afferent sorting filter is only used to sort afferents, not to generate orientation maps. The afferent sorting filter makes an afferent move within the cortical subplate in different directions (e.g. up, down, to the left or to the right). The convolution is only used to identify the direction that makes the afferent join afferents of the same type (e.g. same eye input or same ONOFF polarity). The model can use any other sorting algorithms to generate afferent clusters with the same retinotopy, eye input and ONOFF polarity. It does not assume any specific mechanism (e.g. correlated activity as in Miller, 1994). The sorting of thalamic afferents, by itself, does not make the orientation map periodic as in Rojer & Schwartz 1990. The afferent sorting filter allows us to accurately replicate the biological patterns of ocular dominance maps that we want to simulate¹ and their differences across species. It allows us to easily generate ocular dominance blobs or stripes of different sizes and reduce their size to simulate their absence in some monkeys¹. We now provide a more detailed explanation of each stage of the model. This

detailed explanation should make it clearer that orientation maps only emerge after the afferents spread their axon arbors and converge in visual cortex, not during afferent sorting.

Minor issues

- Page 5 "... ON-OFF geometry, and orientation preference, closely replicating experimental measures" : This sounds not precise. Describe which measures should be replicated, and what statistical test you can show to validate them.

Thank you for asking us to provide more statistical comparisons between experimental measures and model. We now include several new metrics to statistically compare biological and simulated cortical maps (see Supplementary Tables 1 and 2).

- Page 6. In tree shrews, orientation map is not as uniform as that of primates but organizes into strips and blobs (Kaschube et al. 2010). Thus, pinwheel density and distance significantly vary across the area. What is a determinant of these local patterns in your model, and how can you make a precise estimation of the pinwheel statistics under this condition?

We are very grateful for this question that helped us explain better how our model generates elongated orientation domains. We now explain in greater detail how the model replicates the orientation maps of tree shrews (Figure 7) and how the elongated orientation domains originate from elongated iso-retinotopic cortical patches (i.e. cortical patches representing the same point in visual space that are elongated). Briefly, the theory predicts that, when cortical iso-retinotopic patches are elongated (e.g. at 17/18 border in tree shrews and cats), the orientation domains also become elongated to maximize the coverage of orientation preference. Conversely, when the iso-retinotopic patch is more circular, the iso-orientation domains are also circular. This is one of the main predictions from our theory: the organization of the retinotopic map drives the organization of maps for all other stimulus dimensions such as ocular dominance, ONOFF polarity, orientation and spatial frequency.

- Page 9. "All these relations measured experimentally could be closely replicated in the model simulations" : How do you validate your model here? i.e. How can you prove that model results are "equivalent" to that of data, instead of just saying that both results show comparable trends or similar statistics?

We now provide new metrics and statistical comparisons to support our claim that we are accurately simulating realistic biological maps (see Supplementary Tables 1 and 2).

Reviewer #2 (Remarks to the Author):

The authors introduce a new model of cortical development that accounts for several experimental relationships. The model consists of several stages: the simulation of retinal ganglion cell mosaics, the sorting of afferents at the cortical plate in a precise sequence (retinotopy, eye of origin and ON/OFF clusters), the innervation and ramification of afferents into the cortex, and a vision-dependent learning refinement of receptive fields and maps.

In general, understanding the details of the model was difficult. Although one can get the gist of the different elements from the introduction, all the details are relegated to a Methods section that is not easy to follow and is hampered by poor mathematical notation. It was not clear which elements of the model are critical to establishing the observed relationships. This is critical because the authors claim the sorting of the afferents is a key contributor to this outcome, but this is not really demonstrated by the analyses offered. While the model shows great promise, it was felt more detailed analyses are needed to support the claims. An increase in the clarity in the presentation and description of the model would also strengthen this study substantially. My comments, questions and suggestions for improvement follow.

Thank you for the criticisms. In this new version of the paper, we have greatly expanded the length of the manuscript and number of figures. We describe the model in greater detail, provide more model simulations and data-model comparisons, and have simplified the description of the model in the methods section. We hope that these extensive changes help us to communicate more clearly the main stages of the model and their contribution to generating cortical topography.

Major comments:

A key ingredient and the novel aspect of the model is the sorting of afferent inputs. What happens in the model if one forgoes such sorting? Can a subsequent refinement due to learning still generate similar maps, receptive fields, and the associated relationships? Without such verification, it is not clear what sorting is exactly contributing, as it may still be possible to explain the observed relationships without invoking the need for afferent sorting.

This is a great point raised by both reviewers. The most important parameter of the model is the afferent density per visual point. As we now illustrate in Figure 1, if there are only four afferents per point (ON and OFF from contralateral and ipsilateral eyes), there is no need for afferent sorting. However, if there are 100 afferents, then the model sorts them by eye input and ON-OFF polarity to reproduce the ocular dominance bands and ON-OFF domains demonstrated in carnivores and primates. We now document in greater detail the contribution of afferent sorting to cortical topography in the paper, main figures and supplementary table 2.

It is clear that retinotopic maps cannot form without sorting afferents by retinotopy, at least in animals with high afferent sampling density. For example, if a cortical plate receives 100 afferents with the same retinotopy, the afferents have to be accommodated in different cortical points either through a random process or afferent sorting. Evidence from cats, macaques and humans indicate that afferents with the same retinotopy but different eye input are sorted by type. That is, they segregate in ocular dominance bands in visual cortex. There is also strong evidence that segregation by eye input in both thalamus and cortex requires sorting of thalamic afferents by a combination of molecular cues and neuronal activity⁹⁻¹¹.

We now make clear that the afferent sorting from the model can be done through multiple mechanisms. It can be driven by molecular cues, axon arbor pruning, or a combination of both. Our model does not require any specific sorting mechanism. It only requires that afferents are sorted by retinotopy, eye input and ON/OFF polarity. There is strong evidence that afferents are sorted by retinotopy in the visual cortex of all mammals. There is also strong evidence that afferents are sorted by eye input in the visual cortex of many mammals with extensive binocular vision. There is increasingly stronger evidence that afferents are sorted by ON/OFF polarity in the thalamus and cortex of different mammals including the mink, ferret, cat and mouse (although the sorting in the mouse may not be needed if the afferent density is low⁸).

In fact, one may suspect sorting can mess things up, as it is necessarily accompanied by a deformation of the retinotopic map. The cartoon on the right shows receptive fields along a line in the retina (just one eye) and their sorting as they go into the cortex (just ON/OFF domains). In such re-organization the receptive fields of the afferents do not change at all. The sorting brings together ON RFs with disparate centers to an ON domain; the same is true for an OFF domain. The local magnification (mm/deg) of ON inputs is *decreased* at the centers of ON domains (and the same happens with OFF domains). Thus, it becomes more likely for cells to sample from ON inputs with rather different locations on the visual field. While this may potentially be corrected by means of synaptic learning, it is not uncommon to see the model report such structures as shown in one of the simulation runs below, which are rarely observed in data.

Thank you for helping us clarify this important point about the afferent sorting. The afferent sorting is restricted to the cortical region representing the same retinotopic point. Within this iso-retinotopic cortical region, all cortical receptive fields are overlapped in visual space. The model adjusts the size and shape of this cortical region based on the size and shape of the ocular dominance bands in each individual animal or species. Retinotopy jumps increase if the values of the model parameters are not biologically possible. However, when the values of afferent number, afferent receptive field size and size of iso-retinotopic cortical patch are consistent, the changes in cortical retinotopy replicate very closely the biological measures (e.g. see movie 6e).

The average separation between afferents converging at the same cortical location in the model is ~2.5 geniculate centers, closely matching the experimental measures (Jin et al., 2011) and the average cortical receptive field size in many different species including macaques, cats, ferrets and mice (~2.5 geniculate centers diameter). In some cortical regions, there is sometimes a competition between the orientation preference imposed by the arrangement of ON-OFF subregions and ON-ON subregions. The GUI figure that the reviewer provides is an example of this, where ON-OFF arrangement causes the orientation preference to be at 150 degrees whereas the ON-ON arrangement causes a weaker bias towards 45 degrees (0 degrees is left-right, 90 degrees is up-down, the polar plots show responses to movement directions orthogonal

to the preferred orientation). We do have experimental measurements similar to the one illustrated by the reviewer in cat visual cortex around cortical regions close to orientation fractures (rapid changes in orientation preference). It is important to remember that each cortical pixel in the model does not reproduce a single cortical cell but a population of cortical cells integrated over 50 x 50 cortical microns.

Of course, the cartoon represents an extreme version of sorting. It would be helpful to visualize the initial and final locations of the afferents after sorting in the model, along with the size of the arbors used. If the axonal arbor is large compared to the movement of the afferents (see cartoon above), then a cortical cell could still “see” the same inputs and synaptic learning could presumably yield similar receptive fields without the sorting. If the arbors are smaller or about the same size of the average afferent displacement (cartoon above) then a cortical cell will necessarily see an increase in local retinotopic scatter at the input and be susceptible to generate the kind of structures displayed above.

Excellent suggestion. Thank you. In the previous version of the manuscript, we had a figure showing the final cortical locations and weights of the afferents converging at the same cortical pixel, but this was not clear enough. We now provide more detailed explanations and illustrations of axon arbors and cortical retinotopy after the afferent sorting is complete. We also make clear that the afferent axon arbor cannot project to a cortical region with a retinotopy that does not overlap with the afferent receptive field. This simple principle limits the retinotopy jumps that the reviewer is concerned about. We hope that the new illustrations and explanations make this clear.

The contribution of synaptic learning is also not immediately obvious. Are some of the same relationships observed (Fig 4) present in the “primordial” map and receptive fields from the input? Is synaptic learning enhancing such relationships or completely establishing them? If the observed relationships are generated by synaptic learning, then what is the value of sorting? If the relationships are observed in the “primordial” map, are they also there to some degree without invoking sorting?

Thank you again. This is an important question that we now address in greater detail. Both developmental stages are needed, the afferent sorting and the synaptic competition. For example, without ONOFF synaptic competition, the changes in spatial resolution within a local cortical distance of 400 microns are limited to 0.25 cpd, a value almost four times smaller than the one obtained in experimental measures. ONOFF synaptic competition segregates convergent afferents in visual space and expands the spatial frequency range. Afferent sorting is also needed. Without clusters of afferents with the same retinotopy, eye input and ONOFF polarity, we cannot reproduce all the correlations among multiple stimulus dimensions measured in our experiments. We now provide simulations and quantification of the model performance without afferent sorting.

Finally, it would also be helpful to visualize the location of the afferents a mature cortical receptive field is pooling from and, importantly, determine how many of those afferents make up a cortical, simple receptive field. Another important number is the density of afferents per cortical hypercolumn, as well as the ratio between the width of a simple RF and the center size of the afferent inputs (see Alonso & Reid) – are those comparable? The authors could estimate these directly from their simulations.

In the previous version of the paper, the location of the afferents in cortical space was illustrated in a supplementary figure. In this new version, it is emphasized and illustrated in the main figures

of the paper. It is important to make clear that the cortical receptive fields at each cortical pixel of the modeled maps represents the average population receptive field from multiple cortical neurons. Therefore, the biological correlate of the cortical receptive fields that we simulate need to be measured with multiunit activity (as in Jin et al., 2011 or Kremkow et al. 2016^{2,12}) and not with single neurons as in Alonso, Usrey and Reid, 2001¹³. The density of afferents per cortical pixel is consistent with the values measured by Jin et al., 2011 and with anatomical estimates¹⁴. However, these numbers are difficult to compare because the model counts an afferent as convergent if it has one synaptic weight in a cortical pixel even if the synaptic weight is small and its contribution to the cortical receptive field (and the ability to measure it experimentally) negligible. The model can count only afferents with a synaptic weight larger than a specific value but choosing this threshold value is quite arbitrary. Therefore, a comparison of afferent convergence between model and biology is not very reliable. We can only say that, in both model and biology, there are hundreds of thalamic afferents converging at the same cortical point¹⁴ (e.g. see Figure 5c).

It was difficult to follow the Methods section and understand exactly what the model entails. The math notation needs to be cleaned up and the model presented in a succinct way. Having GUI to was extremely helpful to get an intuition about the model, although the user has no access to the model parameters. I would encourage the authors to make available those parameters as well.

Thank you for this comment. We agree that the methods section was difficult to follow and the GUI would be much more helpful if it was expanded. We now made an effort at making the methods section much easier to follow and added more parameters and illustrations in the GUI. We also made an effort at using simpler and more consistent terminology in the GUI and paper.

Minor comments

In lack of a unifying model of cortical map topography, it has been frequently argued that different species use different mechanisms to generate cortical maps, receptive fields and stimulus selectivity. Against this belief, here we introduce a general theory of cortical map formation that replicates the cortical topography and receptive field properties of different species, and their combined topography for multiple stimulus dimensions.

There have been numerous prior attempts to come up with a universal mechanism. Work by Fred Wolf and colleagues on self-organizing maps, Swindale, Goodhill, Miller, Ringach & Paik, Sur and colleagues, and wiring minimization such as those from Chklovskii and colleagues.

We agree. We have removed this sentence from the paper.

Finally, the co-activation of the convergent afferents creates a primordial map for stimulus orientation that is later refined by visual experience (Figure 1e-f). I could not find a mathematical description of what this co-activation entails. The Methods includes a section that begins with -- *The model assumes that afferent co-activation plays a major role in the synaptic adjustment transforming the primordial orientation map into a mature visual map.* And it goes on to provide a verbal description of what is expected. Is the model implementing this? How? Also, it is stated that *“because each afferent can only maintain a maximum synaptic weight throughout the cortex, strengthening the synaptic weight at the center of the axon arbor should weaken the weight at the arbor edges.”* I know of no evidence this is the case. Such a mechanism would mean synaptic learning is not local anymore. Can the authors provide a citation for this statement?

We have completely re-written this section of the paper. The model does not simulate the afferent co-activation. It assumes that there is afferent co-activation during visual experience and this co-activation helps adjust the synaptic weights and maximize the orientation coverage at each afferent cluster.

Consistently with experimental evidence, visual retinotopic maps do not form if the thalamic afferents are not sorted by retinotopy, ocular dominance maps do not form if the afferents are not sorted by eye input, and ON-OFF maps do not form if the afferents are not sorted by ON/OFF polarity. Sounds a bit like circular reasoning here. Can you clarify what you mean?

We now describe in more detail the role of afferent sorting in generating maps for retinotopy, ocular dominance, ON/OFF and orientation. We also provide simulations to support each of these statements. We hope that the illustrations together with a more detailed explanation help to make this point clearer.

To further clarify this point, it is helpful to simulate the binocular visual cortex of a cat or a ferret. To simulate the cortical topography of these species, the model has to reproduce the afferent clusters in cortical space. These afferent clusters contain numerous afferents with similar retinotopy, the same eye input and same ON/OFF polarity¹⁵⁻¹⁷. If there are hundreds of afferents converging at the same cortical point¹⁴, it is clear that there has to be some mechanism that sorts the afferents by type to generate these clusters. The question is not whether the afferents are sorted or not. The question is how the afferents are sorted. The sorting could take place in the cortical subplate before the afferents grow in the cortex. It could also take place in the visual cortex by axon arbor pruning. The model does not require any specific sorting mechanism. It just requires that afferents are clustered in visual cortex by type.

Moreover, maps for other stimulus dimensions such as orientation do not form either if the afferents are not sorted by ON-OFF polarity because the size of the iso-orientation domains is determined by the size of the ON and OFF afferent clusters. There are plenty of models where afferents are not sorted, yet maps are generated just fine. Is this a statement about experiments? If so, provide a citation.

This statement refers specifically to our model. We now describe the model in more detail and provide simulations without afferent sorting. We also make a greater effort at explaining more clearly what we call a 'cortical map'. We say early in the paper that the pattern of an orientation map can be simulated with many different models or simply by filtering noise. The point that we are trying to make is that the pattern of an orientation map is not a cortical map. A cortical map has to describe all the stimulus dimensions associated with each cortical point, which include retinotopy, ocular dominance, ON/OFF polarity, orientation preference/selectivity, spatial frequency preference/selectivity, and receptive field structure. It has to provide all these parameters for each eye, for the cortical neurons and for the hundreds of thalamic afferents that converge at the same cortical point. It has also to provide the values of the correlations among all parameters across cortical space.

There is a large number of approaches that can be used to reproduce the pattern of an orientation map. However, the number of approaches decreases dramatically if the requirement is not only to reproduce the cortical gradient for orientation but also for other multiple stimulus dimensions and their relations (e.g. see Supplementary Table 2). We have been working on this model over

the past five years and kept modifying it to replicate an increasingly larger number of experimental observations. This approach provided an unusually large number of biological constraints that helped us to reveal what we believe are some important general principles of cortical map formation.

SF50 – Please define somewhere. I could not find the definition.

Our apologies. SF50 was previously defined in the methods section under ‘Visual stimuli and data analysis’ and under ‘Measurements of stimulus topography in the mature cortical map’. It was also defined in the legend of Figure 4. However, this is an important parameter that should be easily found when reading the paper. We now define SF50 also in the main text.

This afferent sorting plays a major role in cortical map development and minimizes the wiring of thalamocortical circuits that link neurons with similar response preferences. Not immediately obvious how afferent sorting leads to wiring minimization of the cortex.

We agree. We have re-written this section and reduced the emphasis on wiring economy.

The theory of cortical map formation that we propose predicts that the number of afferents sampling each retinal point should determine the number of stimulus dimensions segregated in the cortical map. How exactly? What is a stimulus dimension? What is “processing power”?

We now explain these terms and ideas in greater detail. We call stimulus dimension to any stimulus property represented in a cortical map such as retinotopy, ocular dominance, ON-OFF polarity, orientation and spatial frequency. We define processing power as the number of V1 cortical neurons available to process each visual point.

For example, if three neighboring ON retinal ganglion cells make connection with one ON LGN neuron, the LGN receptive field is calculated as the weighted sum of the three inputs and the strength of each input is determined by the overlap between its receptive field and the three-input receptive-field sum. To compute the LGN RF you need to know the weights. Then you state that you compute the weights by the overlap between the RF and the RGC receptive field. But you don't have the RF. This is totally circular. And then, the next sentence you state you are going to have a one-to-one connection between RGC and LGN neurons anyway.

We apologize for the confusion. The model has been evolving over the past five years and we are struggling to incorporate all the features that we would like to have. Retinogeniculate convergence is one of the features that we really would like to add but did not have time to implement. We now describe it in greater detail in the methods section as a possible future addition.

This sequence of afferent sorting ... is consistent with experimental measurements in the visual pathway [8,9]. In several places, the authors remark that sorting of afferents at the cortical plate, along with the sequence they propose, is an established experimental fact. It would help to discuss the experimental evidence for each of the sorting steps and provide clear supporting evidence for each. The sentence above, for example, one of the studies refers to the segregation in the LGN, not the cortex – why is it relevant?

Thank you for this criticism. We now explain this in greater detail in the methods section. There is evidence that ON and OFF afferents segregate in the LGN and cortex of ferrets^{16,18}. There is

also evidence that afferents segregate first by retinotopy, then by eye input and then by ON/OFF polarity in ferret LGN¹⁸. Therefore, it is reasonable to assume that the afferent segregation follows the same sequence in ferret visual cortex. In cats, there is also evidence that eye input segregates in cortex before ON and OFF pathways segregate in the retina^{11,19}. There is also evidence that ON and OFF afferents segregate in the visual cortex of both cats and ferrets and there is some limited evidence in macaques^{2,15,16}. Although a considerable amount of data is still needed to understand the extent of ON and OFF afferent segregation, the available evidence is consistent with the notion that afferents segregate first by retinotopy, then by eye input and then by ON/OFF polarity. However, even if this sequence was not correct, our model still works as long as the afferents segregate by these three stimulus dimensions and form clusters in cortex. There is strong evidence that this is the case in four different species^{8,15-17}.

Recall that during development many cells in the ferret LGN have receptive fields that are similar to those of simple-cells in V1 (Tavazoie & Reid, 2000). These appear to result from the convergence of multiple ON/OFF RGC RFs onto single geniculate targets. Why couldn't a similar wiring arise in the cortex without invoking afferent sorting?

The afferent sorting could be done through different mechanisms and the model that we propose does not require any specific one. Our afferent sorting filter is very helpful to replicate the specific patterns of ocular dominance found in different species but the model could use any other sorting algorithm that can generate afferent clusters for eye input and ON-OFF polarity. In real brains, the afferent sorting may be done through molecular cues, afferent pruning or a combination of both. The pruning of retinogeniculate convergence is a good example of afferent sorting by eye input¹⁰ and a similar mechanism is probably responsible for afferent sorting by ON/OFF polarity¹⁸. The work that the reviewer cites is a good example of afferent sorting in thalamus. Early in development, each LGN cell receives convergent input from multiple retinal afferents with different retinotopy, eye input and ON-OFF polarity. However, as retinogeniculate connections mature, most retinogeniculate connections are pruned and the ones remaining are matched in retinotopy, eye input and ON-OFF polarity. This pruning process segregates and sorts afferents by eye input and, in some animals, by ON-OFF polarity. This is precisely what we call afferent sorting.

Methods Eqn (6). Is TRF defined anywhere?

In the previous version, it was defined in equation 5. However, the definition was probably confusing because it was defined separately for ON, OFF, contra and ipsi pathways (e.g. TRFONC for ON contra pathways). We have now simplified the math and definitions to make the description of the model more succinct and easier to follow.

“algorithm that we recently developed....” This is really a modified version of a Kohonen self-organizing map.

Our algorithm is more related to the work of Swindale than Kohonen. Therefore, we now describe it as a modified version of Swindale's algorithm as we did in our J Neuroscience paper (Swindale, 1991). All these algorithms generate clusters of particles that can be afferents, orientation preferences or retinotopies. What is different about our algorithm is that it does not change the weights of anything. It simply relocates afferents and make them become clustered by type. As Kohonen and Swindale, we take measurements of the local neighborhood of a pixel (afferent) and measure whether the pixel is surrounded by pixels of the same type or different type. In the previous version of the algorithm, we changed the polarity of the pixel if it was surrounded by

afferents of different type. In the new algorithm from this paper, we move the afferent towards the region that has afferents of the same type.

The advantage of our sorting method is that it can easily replicate any specific pattern of ocular dominance map and match the shape and size of the ocular dominance bands from an individual animal. It is also important to estimate the size of the iso-retinotopic cortical region. This feature of the model is also very helpful when using ocular dominance maps as inputs to generate cortical orientation maps (e.g. Figure 7).

In the GUI, are the squares representing the RGC distributions and the cortical maps covering the same part of the visual field? That is, is the RGC distribution projecting to the entire cortical area we see simulated? If not, can arrange these in such a way that they match?

Our apologies. The figures of the old GUI probably made this reviewer believe that the model causes a pronounced distortion of retinotopy. We have now changed the figures of the paper and the GUI to avoid this confusion. It is not easy to generate a simple figure that illustrates a square cortical patch and the corresponding square retinal patches from the two eyes projecting to the cortical patch. The reason is that a square cortical patch can only accommodate a square number of afferents originating from both eyes, which is a subset of the illustrated retinal arrays. We tried several approaches to illustrate this in a way that is simple, clear and avoids confusion. If we only show the subset of retinal ganglion cells feeding the thalamic afferents of the cortical patch, the reader cannot appreciate the ONOFF geometry of the retinal array. If we show the entire retinal array, the reader can be confused and believe that the receptive fields of the thalamic afferents are widely separated in visual space. To avoid this problem, we decided to show the two square ONOFF retinal arrays that include all the inputs to the cortical patch to allow the readers to appreciate the ONOFF geometry of the retinal arrays. Then, in a separate figure panel, we show the receptive fields of the subset of afferents that project to the square cortical patch, 64 (8 x 8) in the new figure 1.

How much is the retinotopy distorted after you sort eye and on/off? If you make a grid on the cortex and ask where the centers of the RFs at those locations are in the visual field, can you please plot the resulting grid? Is the amount of deformation consistent with what we see experimentally (eg, Campbell & Blasdel in primate or Fitzpatrick et al in ferrets or tree shrews)?

The simulated changes in retinotopy are very consistent with experimental measures. Retinotopy changes smoothly when measured across a few hundreds of cortical microns^{20,21} and more randomly when measured at shorter distances of ≤ 100 microns^{2,5,22}. We now illustrate the retinotopy after afferent sorting by eye input and ONOFF polarity (new Figure 4c, right). We also illustrate smooth changes in retinotopy (and the comparison with experimental data) in Figure 6d, S4c and Figure/movies 6e-f.

I could not find the reference to the video files referenced in the text.

Our apologies. We now make sure that all figure panels are cited in the main text. However, notice that some of the video files are part of a supplementary figure and are then cited in the supplementary figure legend.

Where is the periodicity of the orientation map coming from in the model?

The map becomes periodic when the orientation coverage is maximized in each afferent cluster. We now make this clear when describing the model in both the main text and methods section.

Retinas are simulated on a square grid + uniform (x,y) noise. Most other modeling studies have used hexagonal arrays and isotropic noise. It is not immediately obvious this has no effect on the results.

We now explain in greater detail the role of ONOFF geometry in the simulations of cortical topography. In the model, differences in ONOFF geometry reproduce individual variations in cortical topography but not species differences. Instead, what causes the variation in orientation maps across species is a variation in afferent sampling density. The paper explains now in more detail the relative contribution of ONOFF geometry and afferent sampling density in the model.

In single retinal arrays, the hexagonal tiling helps to sample visual space as homogeneously and densely as possible. However, the visual cortex receives ON and OFF afferents from multiple retinal arrays that originate in two different eyes and there are several types of retinal ganglion cells projecting to LGN. Each retinal ganglion cell also makes connections with multiple geniculate neurons²³, and this retinogeniculate divergence/convergence transforms each ONOFF retinal array in multiple ONOFF thalamic arrays. When all thalamic mosaics are included, hundreds of thalamic afferents project to the same cortical point¹⁴. In our theory and model, the sorting of these thalamic afferents by retinotopy, eye input and ONOFF polarity becomes a more important factor in shaping cortical topography than the ONOFF geometry of a specific retinal array (e.g. beta cells from the contralateral eye). However, that being said, consistent with previous ONOFF retinal models^{7,8,24-27}, the ONOFF retinotopy is also extremely important in our model and, without it, it is not possible to reconstruct the exact cortical topography of an individual animal.

It would be helpful if you could label IPSI/CONTRA eye and ON/OFF dominance in the left most two panels in the GUI (under primordial maps tab).

Thank you. We have completely revised and expanded the GUI to allow the reader perform a larger number of custom simulations and visualize the cortical gradients for multiple stimulus dimensions, receptive fields and thalamic axon arbors.

The relationship between orientation tuning and the local balance of ON/OFF inputs has been emerges in a prior model (see Ringach 2007, PLoS One, Fig 5c), and the same for spatial frequency (Fig 6 and see Jang & Paik 2017). These types of models demonstrate that many of the observed relationships can, in principle, arise without the need to invoke afferent sorting.

Thank you for bringing up this important point. It is true that previous models are able to replicate certain map patterns and map relations. However, our model and theory aims to replicate a much larger number of experimental observations and, by doing so, provide a theoretical framework that is much more biologically constrained than in the past. As the number of constraints increase, it becomes more difficult to replicate experimental observations as we now illustrate in supplementary table 2. This has been a long journey for us that took us through multiple back-and-forth routes between experiment and model. Our goal from the start was to have a model that could replicate all the experimental measures that we have been collecting over the years in cat visual cortex. The model had to be modified multiple times to incorporate additional biological constraints and became increasingly more ambitious over the years. We realize that the model still falls short from replicating all the experimental evidence available and there are several

features that we would like to add in the future. However, in our opinion, the approach of challenging the model to reproduce an increasingly larger body of experimental data has revealed several basic principles that we think are important. We would be very surprised if they turned out to be wrong. Time will tell.

- 1 Najafian, S., Jin, J. & Alonso, J. M. Diversity of Ocular Dominance Patterns in Visual Cortex
Originates from Variations in Local Cortical Retinotopy. *J Neurosci* **39**, 9145-9163,
doi:10.1523/JNEUROSCI.1151-19.2019 (2019).
- 2 Kremkow, J., Jin, J., Wang, Y. & Alonso, J. M. Principles underlying sensory map topography in
primary visual cortex. *Nature* **533**, 52-57, doi:10.1038/nature17936 (2016).
- 3 Lee, K. S., Huang, X. & Fitzpatrick, D. Topology of ON and OFF inputs in visual cortex enables an
invariant columnar architecture. *Nature* **533**, 90-94, doi:10.1038/nature17941 (2016).
- 4 Yu, H., Farley, B. J., Jin, D. Z. & Sur, M. The coordinated mapping of visual space and response
features in visual cortex. *Neuron* **47**, 267-280, doi:10.1016/j.neuron.2005.06.011 (2005).
- 5 Das, A. & Gilbert, C. D. Distortions of visuotopic map match orientation singularities in primary
visual cortex. *Nature* **387**, 594-598, doi:10.1038/42461 (1997).
- 6 Kremkow, J. & Alonso, J. M. Thalamocortical Circuits and Functional Architecture. *Annu Rev Vis
Sci* **4**, 263-285, doi:10.1146/annurev-vision-091517-034122 (2018).
- 7 Ringach, D. L. *et al.* Spatial clustering of tuning in mouse primary visual cortex. *Nat Commun* **7**,
12270, doi:10.1038/ncomms12270 (2016).
- 8 Tring, E., Duan, K. & Ringach, D. L. ON/OFF domains shape receptive fields in mouse visual cortex.
bioRxiv preprint doi: <https://doi.org/10.1101/2021.09.09.459679> (2021).
- 9 Huberman, A. D., Murray, K. D., Warland, D. K., Feldheim, D. A. & Chapman, B. Ephrin-As mediate
targeting of eye-specific projections to the lateral geniculate nucleus. *Nat Neurosci* **8**, 1013-1021,
doi:10.1038/nn1505 (2005).
- 10 Shatz, C. J. The prenatal development of the cat's retinogeniculate pathway. *J Neurosci* **3**, 482-499
(1983).
- 11 LeVay, S., Stryker, M. P. & Shatz, C. J. Ocular dominance columns and their development in layer
IV of the cat's visual cortex: a quantitative study. *J Comp Neurol* **179**, 223-244,
doi:10.1002/cne.901790113 (1978).
- 12 Jin, J., Wang, Y., Swadlow, H. A. & Alonso, J. M. Population receptive fields of ON and OFF thalamic
inputs to an orientation column in visual cortex. *Nat Neurosci* **14**, 232-238, doi:10.1038/nn.2729
(2011).
- 13 Alonso, J. M., Usrey, W. M. & Reid, R. C. Rules of connectivity between geniculate cells and simple
cells in cat primary visual cortex. *J Neurosci* **21**, 4002-4015 (2001).
- 14 Peters, A. & Payne, B. R. Numerical relationships between geniculocortical afferents and
pyramidal cell modules in cat primary visual cortex. *Cereb Cortex* **3**, 69-78,
doi:10.1093/cercor/3.1.69 (1993).
- 15 Jin, J. Z. *et al.* On and off domains of geniculate afferents in cat primary visual cortex. *Nat Neurosci*
11, 88-94, doi:10.1038/nn2029 (2008).
- 16 Zahs, K. R. & Stryker, M. P. Segregation of ON and OFF afferents to ferret visual cortex. *J
Neurophysiol* **59**, 1410-1429, doi:10.1152/jn.1988.59.5.1410 (1988).
- 17 McConnell, S. K. & LeVay, S. Segregation of on- and off-center afferents in mink visual cortex. *Proc
Natl Acad Sci U S A* **81**, 1590-1593, doi:10.1073/pnas.81.5.1590 (1984).
- 18 Speer, C. M., Mikula, S., Huberman, A. D. & Chapman, B. The developmental remodeling of eye-
specific afferents in the ferret dorsal lateral geniculate nucleus. *Anat Rec (Hoboken)* **293**, 1-24,
doi:10.1002/ar.21001 (2010).
- 19 Bodnarenko, S. R., Jeyarasasingam, G. & Chalupa, L. M. Development and regulation of dendritic
stratification in retinal ganglion cells by glutamate-mediated afferent activity. *J Neurosci* **15**, 7037-
7045 (1995).
- 20 Blasdel, G. & Campbell, D. Functional retinotopy of monkey visual cortex. *J Neurosci* **21**, 8286-
8301 (2001).

- 21 Bosking, W. H., Crowley, J. C. & Fitzpatrick, D. Spatial coding of position and orientation in primary visual cortex. *Nat Neurosci* **5**, 874-882, doi:10.1038/nn908 (2002).
- 22 Wang, Y. *et al.* Columnar organization of spatial phase in visual cortex. *Nat Neurosci* **18**, 97-103, doi:10.1038/nn.3878 (2015).
- 23 Hamos, J. E., Van Horn, S. C., Raczkowski, D. & Sherman, S. M. Synaptic circuits involving an individual retinogeniculate axon in the cat. *J Comp Neurol* **259**, 165-192, doi:10.1002/cne.902590202 (1987).
- 24 Paik, S. B. & Ringach, D. L. Retinal origin of orientation maps in visual cortex. *Nat Neurosci* **14**, 919-925, doi:10.1038/nn.2824 (2011).
- 25 Paik, S. B. & Ringach, D. L. Link between orientation and retinotopic maps in primary visual cortex. *Proc Natl Acad Sci U S A* **109**, 7091-7096, doi:10.1073/pnas.1118926109 (2012).
- 26 Jang, J. & Paik, S. B. Interlayer Repulsion of Retinal Ganglion Cell Mosaics Regulates Spatial Organization of Functional Maps in the Visual Cortex. *J Neurosci* **37**, 12141-12152, doi:10.1523/JNEUROSCI.1873-17.2017 (2017).
- 27 Song, M., Jang, J., Kim, G. & Paik, S. B. Projection of Orthogonal Tiling from the Retina to the Visual Cortex. *Cell Rep* **34**, 108581, doi:10.1016/j.celrep.2020.108581 (2021).

REVIEWER COMMENTS

Reviewer #1 (Remarks to the Author):

I appreciate the authors' effort devoted to the manuscript, because the computational idea introduced here may provide an important approach of unraveling the hidden relationship between cortical map formation and the structure of thalamocortical circuitry. I also thank the authors for their hard work in the revision. Particularly, it was good to use the first section of the manuscript to describe the model, a core material of the paper. Now the model is explained better with greater details, thus is ready for a full-fledged argument of its main story.

While the manuscript is improved in its organization and demonstration of the model, now it becomes clear where the authors stopped short of studying models published previously. This raises the key problem — with the current framing of the manuscript, the authors' argument becomes massively misleading, particularly about the difference between the current model and other models suggested previously.

Apart from a number of detailed issues, let me point out a couple of major issues that may seriously affect the novelty of the current manuscript. All the other issues may need to be discussed, only when the authors can properly address the following issues.

1. The main idea, hypothesis and the main findings of the current study are not new. The current model should be considered an extended version of already existing model recently published.

Unfortunately, the most important part of the current study, the working mechanism of the model, is almost identical to that in a recently published study, but the authors do not realize this. Please find the paper below.

Retino-Cortical Mapping Ratio Predicts Columnar and Salt-and-Pepper Organization in Mammalian Visual Cortex, Jaeson Jang, Min Song, and Se-Bum Paik, Cell Reports 30(10), 3270-3279 (2020)

As the authors mentioned in the manuscript as “Here we introduce a general theory of cortical map formation, whose main proposal is that cortical map diversity originates from differences in the afferent sampling density of sensory space across species.”, “The close relation between afferent sampling density and cortical size is the main pillar of our theory. The theory proposes that, as afferent sampling density increases, the cortex becomes larger and segregates afferents...”, this is the exactly same hypothesis and same finding in the above study.

I was surprised that the authors did not cite or discuss this paper (I find that several papers from the above group were already cited. Then the most relevant one is missing??), which raises a serious issue for the novelty of the model presented here.

Strictly speaking, it is true that two models are not completely identical in details, as the current model considers more various factors of retinotopy, ocular dominance and etc. However, it is also clear that we cannot say this model is novel because it is different from other models to some degree.

Rather, a correct and reasonable claim is that the current model is an extended version of the previous model, Jang et al 2020, and may provide a more detailed repertoire of model analysis. I believe this is how the current manuscript should be framed. In this case, the authors also should consider that a large portion of the current results is already reported in the previous two papers (Jang, Song and Paik, Cell Reports 2020 and Song, Jang, Kim and Paik, Cell Reports 2021: currently Ref 12). Thus the authors should carefully study previous literatures to find which part of the current model is indeed new and worth publication.

2. There are incorrect explanations and interpretations, particularly when the current model is compared with previous ones, regarding novelty and distinction.

This also happens in the previous version of the manuscript (see Minor comments #1 from Reviewer 2), and is still ongoing issue in the revised manuscript. This issue may also seriously affect the reliability of the current study unless the manuscript is thoroughly corrected.

For example, the authors mentioned that “Unlike previous models, the theory predicts that nearly all ON-OFF mosaic geometries should be able to generate an orientation map as long as the afferent sampling density is high. Conversely, it predicts that orientation maps will not form if the afferent sampling density is low, regardless of the ON-OFF mosaic geometry”. Please read the main result (Figs 2 and 3 and Fig S1) of the Jang, Song and Paik 2020 above. The idea, hypothesis and even most of the main results are identical (They showed that the same cat mosaics, or same model mosaics, can generate both columnar architecture and salt-and-pepper organization). Thus, the current result should not be claimed as a totally new one, but may be considered a nice validation of the previous model.

Authors may claim that the current model can generate maps not only with any RGC mosaics data but also with arbitrary mosaics that are mathematically generated as described in the manuscript. This is another place that the authors are ignoring an important model condition previously reported.

First, maps can emerge from various ON-OFF mosaics, as long as they are in some noisy, periodic lattice pattern. See Ringach, Haphazard wiring of simple receptive fields and orientation columns in visual cortex. *J. Neurophysiol.* 92, 468–476 (2004) and Ringach, On the origin of the functional architecture of the cortex. *PLoS ONE* 2, e251 (2007), for various model conditions of RGC mosaics.

Second, this is because these periodic ON and OFF lattices can always generate an interference pattern that determines the local organization of ONs and OFFs and eventually constrains the columnar map periodicity. See Amidror, I. *The Theory of the Moiré Phenomenon* (Kluwer Academic, Norwell, Massachusetts, USA, 2000). Thus, please find that even your periodic RGC model should generate a Moiré interference pattern, and this is why you can get a map when sampling density is sufficiently high.

Third, however, there is an important model factor that is not arbitrary but should be validated by animal data — The structure of the RGC mosaics can vary topographic organization of the emerged map, which is already analyzed in animal data. For example, hexagonal RGCs lattice pattern will generate

hexagonal periodicity of the cortical map, while rectangular RGCs lattice will generate different, rectangular-like map periodicity. Then, the topographic pattern of the maps is already known as hexagonal. See Paik and Ringach, Retinal origin of orientation maps in visual cortex, *Nature Neuroscience* 14, 919-925 (2011). This is an important part of the model to be validated.

Overall, my point is that the authors should have a thorough understanding of other models to prove that their model has sufficient novelty and thus worth publication.

All things considered, my primary concern is that the authors are overstating the novelty of the current model with incomplete and inexact understanding of previous studies. Thus, I am not strongly convinced that the main idea of the current model is novel, and the authors' attempts to convince me of this were not compelling with the current version of the manuscript and the rebuttal.

However, I still think the manuscript is worth publishing if properly revised. Then, why do I support this study, and ultimately hope it is accepted? As I stated previously, the basis of this manuscript is a good one; an important question. The manuscript provides an intriguing perspective regarding the origin of cortical functional circuits organized into systematic patterns that have inspired neuroscientists for a long time.

Thus, I recommend that the authors put efforts to clarify that the model is based on the same idea with a model previously published, but the authors try to extend it to make a broader and more detailed model. I hope the authors will do the work properly and to finally make it.

Reviewer #2 (Remarks to the Author):

The authors have thoroughly revised the manuscript improving description of the model and have adequately addressed all my questions/concerns.

This is an unusual response to an unusual review. The main purpose of our response is to strongly express our disagreement with the reviewer assessment of our work. Our model is very different to any of the models published by the Paik lab. Our main ideas, hypotheses and results are ours (not Paik's) and are very different from those previously published. The only similarity between our model and previous Paik's models is that they both use an array of ON and OFF retinal ganglion cells as many other models have used going back to the work of Linsker. Below, there is a detailed response to each point raised by the reviewer (all changes in the main manuscript are marked in red).

Reviewer #1 (Remarks to the Author):

I appreciate the authors' effort devoted to the manuscript, because the computational idea introduced here may provide an important approach of unraveling the hidden relationship between cortical map formation and the structure of thalamocortical circuitry. I also thank the authors for their hard work in the revision. Particularly, it was good to use the first section of the manuscript to describe the model, a core material of the paper. Now the model is explained better with greater details, thus is ready for a full-fledged argument of its main story.

Thank you for acknowledging our 6-month effort at addressing all the comments from the two reviewers.

While the manuscript is improved in its organization and demonstration of the model, now it becomes clear where the authors stopped short of studying models published previously. This raises the key problem — with the current framing of the manuscript, the authors' argument becomes massively misleading, particularly about the difference between the current model and other models suggested previously. Apart from a number of detailed issues, let me point out a couple of major issues that may seriously affect the novelty of the current manuscript. All the other issues may need to be discussed, only when the authors can properly address the following issues.

Below, we make an effort at addressing all the major criticisms. However, we would have appreciated if the reviewer listed all the criticisms that he/she currently has, as is common practice in peer review. Our understanding of this paragraph is that we are only being asked to address what the reviewer calls 'a couple of major issues', listed as 1 and 2.

1. The main idea, hypothesis and the main findings of the current study are not new. The current model should be considered an extended version of already existing model recently published. Unfortunately, the most important part of the current study, the working mechanism of the model, is almost identical to that in a recently published study, but the authors do not realize this. Please find the paper below.

Retino-Cortical Mapping Ratio Predicts Columnar and Salt-and-Pepper Organization in Mammalian Visual Cortex, Jaeson Jang, Min Song, and Se-Bum Paik, Cell Reports 30(10), 3270-3279 (2020)

As the authors mentioned in the manuscript as "Here we introduce a general theory of cortical map formation, whose main proposal is that cortical map diversity originates from differences in the afferent sampling density of sensory space across species.", "The close relation between afferent sampling density and cortical size is the main pillar of our theory. The theory proposes that, as afferent sampling density increases, the cortex becomes larger and segregates afferents...", this is the exactly same hypothesis and same finding in the above study. I was surprised that the authors did not cite or discuss this paper (I find that several papers from the above group were already cited. Then the most relevant one is missing??), which raises a serious issue for the novelty of the model presented here.

We respectfully disagree with the reviewer. Our main idea, hypothesis and main findings did not originate in Paik's work. Our main hypothesis was proposed in a paper that we published in 2017 (Mazade and Alonso, 2017, Visual Neuroscience), which is cited in Jang, Song, and Paik (2020), and was further elaborated in a model that we published in 2019 (Najafian et al., 2019, Journal of Neuroscience) and a review that we published in 2018 (Kremkow and Alonso, 2018, Annual Review of Vision Science).

Our main hypothesis is that an increase in **thalamic** afferent sampling density leads to **thalamic afferent sorting**, which drives the emergence of cortical clustering for retinotopy, ocular dominance, ONOFF polarity, and orientation.

The hypothesis from Jang, Song, and Paik (2020) is that cortical clustering arises from the 'feedforward mapping **ratio between RGC and V1** neurons ($F = NV/NR = \text{number of V1 neurons per RGC}$)'.

The two papers use the term '**sampling**' to explain the emergence of orientation maps. But the use of a common term does not imply that the ideas, models and hypotheses are the same. **In Jang, Song and Paik (2020), the cortex samples the ONOFF retinal mosaic geometry** and orientation maps emerge as the cortical size (and number of cortical neurons) increase the cortex/retinal ratio beyond the Nyquist limit. **In our paper, the thalamic afferents sample each point of visual space** and orientation maps emerge as the increase in the number of thalamic afferents expands the representation of each visual point in cortex. Therefore, in Jang, Song and Paik (2020) orientation maps originate as a consequence of **how cortical targets sample the retinal ONOFF mosaic geometry**. In our model, orientation maps originate as a consequence of **how thalamic afferents sample visual space**.

In fact, we could argue that the hypothesis from Jang, Song, and Paik (2020) is closely related to the hypothesis that we proposed in our 2017 paper. The similarity of the

ideas between our 2017 paper and the Jang, Song, and Paik (2020) paper is best illustrated with a figure below (Figure 1). The left panel reproduces our figure from 2017 and the right panel a figure from Jang, Song, and Paik (2020). **In our paper of 2017, we proposed that, as the number of thalamic afferents increases, the area of cortex representing the same visual point expands** increasing cortical visual acuity. The larger cortical area per visual point allows the afferents to be sorted by their properties (retinotopy, ocular dominance and ON/OFF polarity), which drives the emergence of cortical orientation maps. **In Jang, Song, and Paik (2020), they also proposed that the visual cortex representing the ON/OFF retinal mosaic expands.** However, their idea diverges from us by proposing that orientation maps emerge because the cortical expansion increases the cortical sampling of the retinal mosaic (cortical/retinal ratio). **Therefore, both papers (ours in 2017, Jang, Song and Paik in 2020) propose that orientation maps emerge from an expansion in**

Figure 1 The panel on the left reproduces a figure from Mazade and Alonso (2017). The panel on the right reproduces a figure from Jang, Song, and Paik (2020).

cortical size.

The hypothesis from Jang, Song, and Paik (2020) is that orientation maps emerge when the **cortical sampling of the ON/OFF retinal mosaic** exceeds the Nyquist limit. Consistently with their hypothesis, orientation maps emerge in animals that have large cortical/retinal ratios. Our hypothesis is that orientation maps emerge from an increase in **thalamic sampling of visual space** that expands the cortical area representing the same visual point, which is the same idea that we proposed in 2017 (Mazade and Alonso, 2017) and in 2019 (Najafian, 2019).

A problem with the cortical/retinal ratio from Jang, Song, and Paik (2020) is that it is calculated using the total number of retinal ganglion cells. In animals without

orientation maps, many retinal ganglion cells do not project to V1. Therefore, their ratio between retina and cortex includes many retinal neurons that are not inputs relayed through the thalamus to the cortex.

Jang, Song, and Paik (2020) propose that animals with a low cortical/retinal ratio do not have orientation maps because the cortical sampling of the ONOFF retinal array geometry is low. Our model proposes that they do not have orientation maps because only a small percentage of the retinal ganglion cells are relayed through thalamic afferents to the cortex and, therefore, the cortex is too small (small number of afferents leads to small cortex) to

accommodate multiple orientation domains per cortical point. In the model from Jang, Song, and Paik (2020), the cortex expands to sample better the ONOFF retinal array; in our model, the cortex expands because the number of thalamic afferents increases. **Both models rely on an expansion of visual cortex to explain the emergence of orientation maps (as we proposed in Mazade and Alonso, 2017), but the mechanisms and interpretation of this expansion are different in the two models.**

Figure 2 illustrates how the ideas and hypothesis driving our model evolved over the years before **Jang, Song, and Paik (2020) published their paper.** In 2017, we showed that the distance between afferents with non-overlapping receptive fields (RF) was larger in

Figure 2. a. Figure from Mazade and Alonso (2017). **b.** Figure from Najafian and Alonso (2019). **c.** Figure from our paper under review (Najafian et al., 2021).

animals that have orientation maps than those with salt-and-pepper organization (Figure 2a). We proposed that afferents separated by larger distances had to be accommodated in a larger cortex. The green and orange squares in Figure 2a represent axon arbors with receptive fields separated by one thalamic receptive field center (the numbers on the top left corner are the distance between axon arbors in mm). Notice that already in 2017 we were proposing that the emergence of orientation maps results from the increase in the number of thalamic afferents and size of cortical area where the afferents are sorted. In 2019, we published a model proposing that cortical clusters for ocular dominance emerge as the cortical size representing the same visual point becomes larger (Figure 2b). In figure 2b (as in Figure 2a), the lines attached to the receptive-field cartoons signal cortical regions receiving afferents with receptive fields separated by one receptive field center. These two papers were published before Jang, Song and Paik (2020). In the current paper under review

we extend the same idea of afferent sorting to orientation maps (Figure 2c). **Notice that the common thread of our idea is that thalamic afferents need to be sorted. Paik's model does not use afferent sorting;** it simply projects the retinal arrays into the cortex. **The reviewer is completely ignoring one of the most important features of our model, which is afferent sorting.**

That being said, we are grateful to the reviewer for bringing up this potential source of confusion and we now make clear in the main manuscript that the visual sampling of our afferent-density model is very different from the cortical sampling of the ONOFF retinal mosaic geometry proposed by Jang, Song, and Paik (2020). The relation that Jang, Song and Paik (2020) discovered is interesting although we disagree about the interpretation. Jose Manuel has been always a strong supporter of the papers from Paik and Ringach labs even if he disagreed with their interpretation and conclusions. We hope that we are treated in the same way.

Strictly speaking, it is true that two models are not completely identical in details, as the current model considers more various factors of retinotopy, ocular dominance and etc. However, it is also clear that we cannot say this model is novel because it is different from other models to some degree.

The two models are very different. Our model does not work if the afferents are not sorted by retinotopy, ocular dominance and ONOFF polarity. By comparison, Paik's model assumes that there is no afferent sorting in cortex. **The reviewer may think that afferent sorting is not important but this is not a good reason to ignore a major component of our model and then claim that the two models are the same.**

At least three correlations have values of $r \geq$						
	$r \geq 0.15$	$r \geq 0.25$	$r \geq 0.3$	$r \geq 0.4$	$r \geq 0.5$	p value
Full model	996	989	974	929	827	$< 0.1 \times 10^{-300}$
No afferent sorting	727	524	179	25	2	
All five correlations have values of $r \geq$						
	$r \geq 0.15$	$r \geq 0.25$	$r \geq 0.3$	$r \geq 0.4$	$r \geq 0.5$	p value
Full model	931	908	837	696	489	0.11×10^{-213}
No afferent sorting	122	47	6	0	0	
All five correlations have correct sign and values of $r \geq$						
	$r \geq 0.15$	$r \geq 0.25$	$r \geq 0.3$	$r \geq 0.4$	$r \geq 0.5$	p value
Full model	929	908	837	696	489	0.11×10^{-213}
No afferent sorting	59	29	4	0	0	
All five correlations have correct sign, range > 0.2 , and values of $r \geq$						
	$r \geq 0.15$	$r \geq 0.2$	$r \geq 0.3$	$r \geq 0.4$	$r \geq 0.5$	p value
Full model	927	907	836	695	489	0.11×10^{-213}
No afferent sorting	7	3	0	0	0	

Supplementary Table 2.

We emphasize again that, without afferent sorting, we cannot reproduce the correlations that we measured in cat visual cortex and **we report in this paper for first time.** **The reviewer is completely ignoring the fact that we cannot reproduce the multiple correlations in cortical mapping without afferent sorting** (e.g. see supplementary table 2

from the paper). Only in the extreme case of having just one thalamic afferent per point of visual space, our model generates thalamic afferents with the same organization as the ON

and OFF arrays of retinal ganglion cells, as in Paik's and Ringach's retinal models. However, **in this extreme scenario, our afferent-density model does not generate orientation maps because the afferent density is extremely low.**

Our two models also make very different predictions that can be tested with experiments in the future:

- 1) In Paik's retinal models, the cortical orientation map originates from a Moire interference of ONOFF retinal mosaics. In our afferent-density model, the cortical orientation map originates from afferent sorting and coverage of orientation preference. **The Moire interference pattern of the ONOFF retinal mosaics is completely disrupted by afferent sorting in our afferent-density model.**
- 2) In Paik's retinal models, the cortical orientation map is a projection of a dominant ONOFF retinal mosaic and, therefore, it is possible to predict the location of each pinwheel center within the orientation map based on the ONOFF mosaic arrangement. **In our afferent-density model, the pinwheel location in cortex is determined by afferent sorting and coverage of orientation preference;** it is not related at all to the ONOFF mosaic or any ONOFF interference pattern.
- 3) In Paik's retinal models, the pinwheel centers in the orientation map show the greatest ONOFF receptive field overlap (as a consequence of the Moire interference pattern). **In our afferent-density model, the pinwheel centers can be OFF dominated, ON dominated or ONOFF,** consistent with our experimental observations (see Kremkow et al. 2016).
- 4) In Paik's retinal models, the ocular dominance map is a consequence of receptive field overlap between ON and OFF retinal ganglion cells. In our afferent-density model, it is a consequence of having a large number of afferents from both eyes representing the same visual point, which become sorted in cortex. **Our model does not reproduce the correlation between ON-OFF receptive field distance and ocular dominance that is a central feature of Paik's retinal model (Song et al., 2021).** We do not find this either in the experimental results when we include a reasonable sample size.
- 5) The cortical/retinal sampling ratio from Jang, Song and Paik (2020) predicts that, if the cortex does not change, an increase in the number of retinal ganglion cells should prevent the emergence of orientation maps. **Our model predicts the opposite.** An increase in the number of retinal ganglion cells should make the orientation domains larger. Future experiments may be able to test this hypothesis in genetically modified animals.

- 6) The cortical/retinal sampling ratio from Jang, Song and Paik (2020) predicts that orientation maps in tree shrews should be stronger for the non-dominant eye because the cortical/retinal ratio is larger (smaller number of retinal ganglion cells projecting to the cortex). **Our model predicts the opposite.** The orientation maps should be stronger for the dominant eye because the number of afferents is larger.
- 7) A key component of our model is the axon arbor spread, which makes each thalamic afferent connect to many neighboring cortical neurons. Paik's model uses a much more local connectivity, which is needed to reliably replicate the ONOFF pattern of the retinal mosaic in the cortex.
- 8) There is a long-list of additional differences. The reviewer is interpreting these differences as 'details'. We see them as fundamental differences between both models.

Rather, a correct and reasonable claim is that the current model is an extended version of the previous model, Jang et al 2020, and may provide a more detailed repertoire of model analysis. I believe this is how the current manuscript should be framed. In this case, the authors also should consider that a large portion of the current results is already reported in the previous two papers (Jang, Song and Paik, Cell Reports 2020 and Song, Jang, Kim and Paik, Cell Reports 2021: currently Ref 12). Thus the authors should carefully study previous literatures to find which part of the current model is indeed new and worth publication.

We respectfully disagree with the reviewer's assessment of our work. **We cannot understand how the reviewer can see our model as an extended version of Jang et al., 2020.** The only similarity is that both models use ONOFF mosaics of retinal ganglion cells. **Everything else after the ONOFF mosaic is completely different and the predictions are different (often opposite).** We cannot understand how the reviewer can say that a large portion of the results is reported in the previous two papers from the Paik lab. If the reviewer is referring to the modeling results, the models are completely different. If the reviewer is referring to experimental results from the Alonso's lab analyzed in Song et al. (2021), both the experimental results and analyses are completely different. We are really confused with these statements.

2. There are incorrect explanations and interpretations, particularly when the current model is compared with previous ones, regarding novelty and distinction. This also happens in the previous version of the manuscript (see Minor comments #1 from Reviewer 2), and is still ongoing issue in the revised manuscript. This issue may also seriously affect the reliability of the current study unless the manuscript is thoroughly corrected. For example, the authors mentioned that "Unlike previous models, the theory

predicts that nearly all ON-OFF mosaic geometries should be able to generate an orientation map as long as the afferent sampling density is high. Conversely, it predicts that orientation maps will not form if the afferent sampling density is low, regardless of the ON-OFF mosaic geometry ". Please read the main result (Figs 2 and 3 and Fig S1) of the Jang, Song and Paik 2020 above. The idea, hypothesis and even most of the main results are identical (They showed that the same cat mosaics, or same model mosaics, can generate both columnar architecture and salt-and-pepper organization). Thus, the current result should not be claimed as a totally new one, but may be considered a nice validation of the previous model.

We again disagree with this criticism. **Both models use similar ONOFF retinal mosaics and both models generate orientation maps but the process to generate the orientation maps is completely different.** Jang, Song and Paik (2020) use a projection from the ONOFF retinal mosaics to the cortex and then drive ocular dominance segregation based on ONOFF receptive field overlap. Our model **does not project the ONOFF retinal mosaics in cortex and does not use ONOFF receptive field overlap to drive ocular dominance segregation.** Our model sorts afferents by their properties (retinotopy, eye input, ONOFF polarity) and then generates periodic orientation maps by adjusting orientation coverage with visual experience. **Our results are not identical either.** We have measured a large number of correlations in visual cortex, which are completely novel, involved many years of work and the reviewer is completely ignoring. **Our model reproduces a large number of correlations that could not be reproduced without afferent sorting.**

Authors may claim that the current model can generate maps not only with any RGC mosaics data but also with arbitrary mosaics that are mathematically generated as described in the manuscript. This is another place that the authors are ignoring an important model condition previously reported. First, maps can emerge from various ON-OFF mosaics, as long as they are in some noisy, periodic lattice pattern. See Ringach, Haphazard wiring of simple receptive fields and orientation columns in visual cortex. J. Neurophysiol. 92, 468–476 (2004) and Ringach, On the origin of the functional architecture of the cortex. PLoS ONE 2, e251 (2007), for various model conditions of RGC mosaics. Second, this is because these periodic ON and OFF lattices can always generate an interference pattern that determines the local organization of ONs and OFFs and eventually constrains the columnar map periodicity. See Amidror, I. The Theory of the Moiré Phenomenon (Kluwer Academic, Norwell, Massachusetts, USA, 2000). Thus, please find that even your periodic RGC model should generate a Moiré interference pattern, and this is why you can get a map when sampling density is sufficiently high.

We agree that some noisy ONOFF patterns will create a mosaic interference but **our model does not need this Moire interference to generate orientation maps.** We now make clear in the revised manuscript that, unlike previous retinal models, our model does not rely

on Moire interference patterns to generate orientation maps. Without afferent sorting, our model does not generate periodic orientation maps that match our experimental measures. If we create ONOFF retinal arrays with Moire interference, the maps that emerge from this interference do not match our experimental measures. Please, notice that the number of experimental constrains that we are imposing to our model is much larger than what is currently available in the scientific literature. **The reviewer is completely ignoring the importance of the experimental work that we are reporting to constrain our model, previous models and future models.**

Third, however, there is an important model factor that is not arbitrary but should be validated by animal data — The structure of the RGC mosaics can vary topographic organization of the emerged map, which is already analyzed in animal data. For example, hexagonal RGCs lattice pattern will generate hexagonal periodicity of the cortical map, while rectangular RGCs lattice will generate different, rectangular-like map periodicity. Then, the topographic pattern of the maps is already known as hexagonal. See Paik and Ringach, Retinal origin of orientation maps in visual cortex, Nature Neuroscience 14, 919-925 (2011). This is an important part of the model to be validated.

We would like to emphasize again that our model reproduces with exquisite precision an incredibly large number of experimental measures (see Figures 6, 7, 8, 9; Supplementary Figures 4, 5, 6; and Supplementary Tables 1 and 2). Our model also reproduces the hexagonal periodicity of the cortical map but the hexagonal periodicity does not emerge from a Moire interference in the ONOFF retinal pattern but from high afferent sampling density.

Overall, my point is that the authors should have a thorough understanding of other models to prove that their model has sufficient novelty and thus worth publication. All things considered, my primary concern is that the authors are overstating the novelty of the current model with incomplete and inexact understanding of previous studies. Thus, I am not strongly convinced that the main idea of the current model is novel, and the authors' attempts to convince me of this were not compelling with the current version of the manuscript and the rebuttal.

We really have a hard time understanding how a reviewer could claim that 'The main idea, hypothesis and the main findings of the current study are not new' and 'The current model should be considered an extended version of' [Paik's model]. We completely disagree with this assessment of our work.

However, I still think the manuscript is worth publishing if properly revised. Then, why do I support this study, and ultimately hope it is accepted? As I stated previously, the basis of this manuscript is a good one; an important question. The manuscript provides an intriguing perspective regarding the origin of cortical functional circuits organized into systematic

patterns that have inspired neuroscientists for a long time. Thus, I recommend that the authors put efforts to clarify that the model is based on the same idea with a model previously published, but the authors try to extend it to make a broader and more detailed model. I hope the authors will do the work properly and to finally make it.

We have made a great effort in this report to demonstrate that our afferent-density model is very different from the retinal models of Paik's lab, and from the work of Jang, Song and Paik, Cell Reports 2020. We acknowledge that many properties of cortical maps can emerge simply from ONOFF mosaic interference, as originally proposed by Paik and Ringach labs. **However, we completely disagree with the main request from this very unusual review. The main ideas, hypotheses, findings and model fundamentals are novel and originated from our work, not from the work of Paik's lab.**

REVIEWERS' COMMENTS

Reviewer #1 (Remarks to the Author):

I thank the authors for their detailed answer to my questions.

I particularly thank the authors for having kindly explained me that I've made some confusion about difference between the two models. Please understand that sometimes it is hard to comprehend the bottom-line of a new research within a limited time, especially it is a theoretical model. I'm very sorry for the troubles caused from my incomplete understanding and appreciate your extra time spent on preparing the detailed rebuttal.

It is also very nice for authors to provide a detailed explanation and discussion of comparing the current model with others in the rebuttal. I think it will be a good resource for researchers who are interested in computational models of cortical development.

Now I believe the manuscript is worth publishing and must be of broad interest to the field.